# Development and evaluation of a method to identify potential release areas of snow avalanches based on watershed delineation

Cécile Duvillier[1], Nicolas Eckert[1], Guillaume Evin[1], Michael Deschâtres[1]

[1]Univ. Grenoble Alpes, INRAE, CNRS, IRD, Grenoble INP, IGE, 38000 Grenoble, France

**Correspondence**: Nicolas Eckert (nicolas.eckert@inrae.fr)

**Keywords**: Snow Avalanches, Potential Release Area, Evaluation, Confusion Matrix, Watershed. French Alps.

**Abstract.** Snow avalanches are a prevalent threat in mountain territories. Large-scale mapping of avalanche prone terrain is a prerequisite for land-use planning where historical information about past events is insufficient. To this aim, the most common approach is the identification of Potential Release Areas (PRAs) followed by numerical avalanche simulations. Existing methods for identifying PRAs rely on terrain analysis. Despite their efficiency, they suffer from i) a lack of systematic evaluation on the basis of adapted metrics and past observations over large areas and ii) a limited ability to distinguish PRAs corresponding to individual avalanche paths. The latter may preclude performing numerical simulations corresponding to individual avalanche events, questioning the realism of resulting hazard assessments. In this paper, a method that well identifies individual snow avalanche PRAs based on terrain parameters and watershed delineation is developed, and confusion matrices and different scores are proposed to evaluate it. Comparison to an extensive cadastre of past avalanche limits from different massifs of the French Alps used as ground truth leads to true positive rates (recall) between 80-87% in PRA numbers, and between 92.4% and 94% in PRA areas, which shows the applicability of the method to the French Alps context. A parametric study is performed, highlighting the overall robustness of the approach and the most important steps/choices to maximize PRA detection, among which the important role of watershed delineation to identify the right number of individual PRAs. These results may contribute to better understanding avalanche hazard in the French Alps. Wider outcomes include an in-depth investigation of the issue of evaluating automated PRA detection methods and a large data set that could be used for additional developments, and to benchmark existing and/or new PRA detection methods.

## 1. Introduction

In mountain territories, snow avalanches are a prevalent threat, resulting in casualties and damages to buildings and critical infrastructures (Amman and Bebi, 2000; McClung and Shearer, 2006). Identification of avalanche-prone terrain, hazard/risk mapping and construction of defence structure are, therefore, the most efficient ways to reduce death tolls and costs on the long range for settlements downslope (Arnalds et al., 2004; IRASMOS consortium, 2009; Eckert et al., 2012; Eckert and Giacona, 2022). To this aim, availability of historical information concerning avalanche location, frequency and magnitude is crucial (Giacona et al., 2017). For instance, spatial statistics can be used to interpolate the data available from a sample of paths (Lavigne et al., 2012; 2017) in order to assess avalanche hazard over large areas. However, even in the best-documented areas, historical information is far from exhaustive, and, in many mountain territories, it remains almost absent. The standard approach to

delineate avalanche-prone terrain is then the automated identification of Potential Release Areas (PRAs) on the basis of terrain analysis followed by numerical (Naaim et al., 2004; Bartelt et al.. 2012) and/or statistical-numerical avalanche simulations (Keylock et al., 1999; Eckert et al., 2010; Fischer et al., 2015). In these, PRA identification

supplemented by snow cover information from neighbouring meteorological stations or reanalyses provide the input conditions for avalanche simulations, which defines hazard and risk levels downslope (Gruber and Bartelt, 2007; Barbolini et al., 2011; Bühler et al., 2018; Ortner et al., 2022). Wider benefits can also arise from the systematic detection of PRAs: better understanding of the avalanching process at large scale, identification of areas that need to be reforested to reduce hazard and risk, etc.

Snow avalanche PRA detection methods from terrain analysis belong to the class of susceptibility mapping methods, which are widely used for many mountain hazards (Bertrand et al., 2013; Eckert et al., 2018). Several automated snow avalanche PRA detection methods are now available in the literature, and, since first proposals (Maggioni et al., 2002; Maggioni and Gruber, 2003), different extensions have been implemented (e.g., Sykes et al., 2021). For example, while PRAs were historically assessed independently from snow and weather conditions,

Chueca Cía et al. (2014) developed a multi-criteria analysis for snow avalanche susceptibility mapping that uses wind directions and snowdrift to identify PRAs in a dynamic way. Also, while most of existing approaches are deterministic (e.g., Bühler et al., 2013), Veitinger et al. (2016) apply fuzzy logic to relate past release areas to slope, roughness and a shelter wind index. Similarly, Kumar et al. (2019) detect PRAs in the Lahaul region of Western Himalaya using a probabilistic occurrence ratio and Yariyan et al. (2020) identify more broadly

"avalanche sensitive areas" using different statistical models.

Most of existing PRA detection methods use forest cover and geomorphologic features such as slope, plan curvature, aspect, and distance to ridges as decisive factors. For instance, distance to ridges is generally retained as a useful quantity because it is a proxy for snowdrift, which is known to be an important triggering factor (e.g., Lehning et al., 2000). Also, forests limit avalanches release by anchoring snowpack and, more generally, lower

avalanche hazard and risk downslope (Bebi et al., 2009; Zgheib et al., 2022; in press), so that presence of a dense forest is often considered as sufficient to exclude a given location from PRAs (e.g. Maggioni et al. 2002). Finally, PRA detection methods are primarily oriented towards large avalanches, which are of interest to assess long-term risk for people and settlements downslope, so that a minimal size is generally considered (e.g. Maggioni et al. 2002).

However, there are some disagreements between researchers about i) the exact choice and respective importance of the different factors to be used in PRA detection, and ii) the best parameter values/thresholds to be specified to reach maximal efficiency. For instance, in Maggioni et al. (2002), included factors are slope, aspect, curvature, forest presence, and distance to ridges, whereas Bühler et al. (2013) consider, in addition, information about roughness and flow direction. Also, it is generally admitted that, for slopes steeper than 60°, snow accumulation

is low (e.g., Maggioni and Gruber, 2003). Yet, the range of slopes to be retained in automated PRA detection remains debated, as i) the true range of slopes over which avalanche release is actually possible remains uncertain (e.g., it varies with snow conditions, Schweizer et al., 2003; Naaim et al., 2013), and ii) it is not independent of the chosen Digital Elevation Model (DEM) resolution. Hence, the 28-60° range is selected in Veitinger et al. (2016) using a DEM with a 2m resolution, but 28-55° is preferred in Aydin and Eker (2017) using a DEM with a

resolution of 10 m, and 25-40° is used in Kumar and al., (2019). Similarly, the range of elevations where PRAs are searched depends on the region of the world. Aydin and Eker (2017) consider the 1000-4000m a.s.l. range in

Turkey, while Kumar et al. (2019) adapt this range to higher mountain environments with a minimal elevation fixed to 2800m a.s.l. Finally, dynamic PRA mapping methods consider snow and weather factors. As examples, Chueca Cía et al. (2014) make use of a multi-scale roughness adjusted to snow depth and of a shelter wind index whereas other parameters describing the climatology (snowfall, temperature change, precipitation), lithology and land use are accounted for in Yariyan et al. (2020).

An important characteristics of most of existing PRA detection methods is their limited ability to distinguish PRAs corresponding to individual avalanche paths/events. Indeed, PRA detection methods mostly focus on terrain characteristics at the pixel scale. Hence, they do not easily segment large areas where factors are favourable to avalanche release (suitable slope, roughness, etc.) in different PRAs compatible with the physical processes involved in snow avalanching (Schweizer, 2003). This may lead to unrealistically wide PRAs that, in reality, correspond to different avalanche paths/events. This drawback is critical for using these PRAs for hazard assessment downslope, as it may preclude performing numerical simulations corresponding to realistic individual avalanche events. An exception is, however, the object-based approach of Bühler et al. (2018), where avalanche terrain is segmented into spatially coherent entities using an image classification step.

To evaluate the performance of PRA detection methods, most studies use recorded avalanches (Maggioni and Gruber, 2003; Bühler et al., 2013; Veitinger et al., 2016), with historical information provided either by local observers after each event or by interviewing local people. Evaluation is often qualitative, and compare the detected PRAs with avalanches boundaries obtained from residents (Aydin and Eker, 2017), from aerial or satellite photographs (Bühler et al., 2019) or from avalanche simulations (Nolting et al., 2018). Over the recent years, more quantitative evaluation methods have gained popularity in the snow avalanche field, notably to assess the efficiency of snow avalanche detection from satellite images (e.g., Karas et al., 2021), or of the prediction of snow avalanche occurrences on the basis of snow and weather conditions (Sielenou et al., 2021; Pérez-Guillén et al., 2021; Viallon-Galinier et al., 2022). These evaluation approaches use different metrics to quantify overall performance based on long historical records or large-scale field data taken as ground truth. Considered metrics include probabilistic occurrence ratios (summarized as confusion matrices) and receiver operating characteristic – area under curve (ROC-AUC criteria). To our knowledge, despite the strong development over the last years of different PRA detection methods, these quantitative evaluation methods have only seen limited use so far (Bühler et al., 2018) to evaluate their respective efficiency, advantages and limits in an objective way. This lack is partially attributable to the fact that information about avalanche release areas is even sparser than information about avalanches in general. Indeed, avalanche releases typically occur in remote uninhabited and/or difficult/dangerous-to-access areas, so that defining a reliable sample of "ground truth" PRA is extremely difficult, if not almost impossible.

On this basis, the main objective of the paper is to develop and evaluate an automated PRA detection method adapted to the context of the French Alps. The aim of our approach is to identify all locations where avalanches can occur. There is no notion of frequency, meaning that avalanche releases may occur very frequently in certain PRAs we detect, and extremely rarely in others. Also, our definition of an individual PRA is the maximal extent corresponding to the release of a single avalanche event. Hence, in practice, many avalanche releases may concern a (potentially small) fraction of a single PRA, especially for the largest PRAs we detect. To reach this goal, we build on already existing developments (notably by Maggioni, et al., 2003 and Bühler et al., 2013), so that our method uses topographical parameters (minimal elevation, range of slopes, maximum distance to ridges) and

presence of forests as key factors. In addition to the careful selection of suitable values for these parameters and data sets, two research questions are specifically targeted: the determination of realistic individual PRAs using a watershed delineation algorithm and the evaluation of the method on the basis of scores computed using two metrics, PRA numbers and areas. For the latter, an extensive cadastre of past avalanche limits from different massifs of the French Alps is used as a support to derive a ground truth validation sample, which is shown to be a non-trivial task. In what follows, Sect. 2. introduces datasets and study areas. Sect. 3 describes the proposed PRA detection method and evaluation framework. Sect. 4 details the results for study areas. Sect 5. discusses main findings and pro's and con's of the proposed approach. Sect. 6. concludes and points out outlooks for further applications and developments.

## 2 Data

### 2.1 Study areas

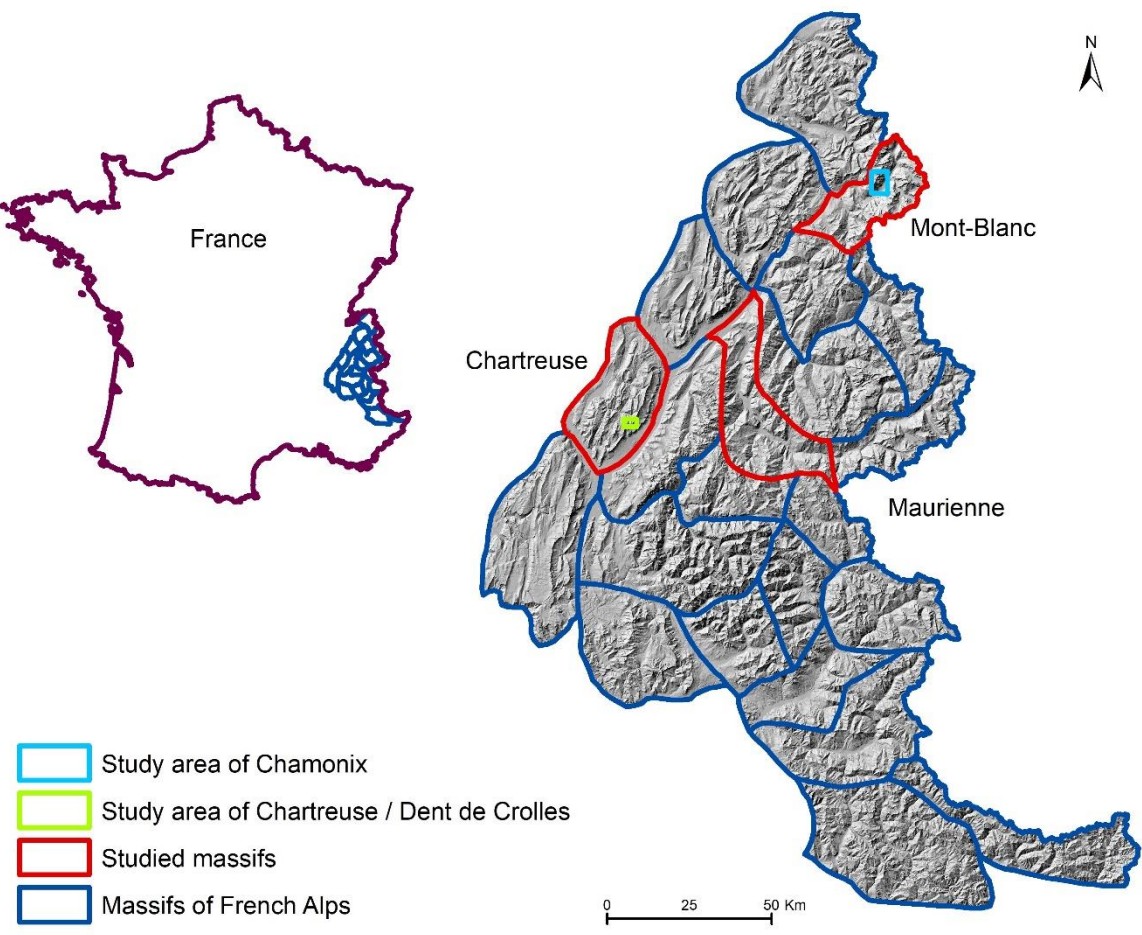

**Figure 1 : Study areas: three entire massifs (within the 23 massifs of the French Alps) and two small highlighted areas, the one of Chamonix and the one of Chartreuse/Dent de Crolles. Digital Elevation Model ©IGN.**

In this paper, we focus on the French Alps and its classical segmentation into 23 massifs for snow-climate reanalyses and operational snow avalanche forecasting (e.g., Durand et al.; 2009a; 2009b; Evin et al., 2021). Despite the high exposure to snow avalanche risk of this territory, no automated PRA detection method was systematically applied in it so far. For this study, three entire massifs with different characteristics are specifically

considered to develop and evaluate the detection method: Mont-Blanc, Chartreuse, and Maurienne (Figure 1). In addition, a focus is made on two smaller test areas, so as to highlight some results and deepen the analyses at a fine spatial scale (Table 1).

The Mont-Blanc massif reaches an elevation of 4,809 m a.s.l. at the Mont-Blanc summit (top of Western Europe), and it is mainly composed of granite and gneiss. The valley of Chamonix is well known for mountaineering, but also to be extremely exposed to snow avalanches. A tragic example is the snow avalanches of Montroc (9 February 1999), which led to the loss of twelve residents and the devastation of fourteen chalets (Ancey et al., 2000). The massif of Chartreuse is a massif of the Prealps, mainly composed of limestone. This massif is less subject to snow avalanches because of its lower elevation (highest point is Chamechaude at 2,082 m a.s.l.), but destructive snow avalanches occurred in it in the past, such as in Saint-Hilaire-du-Touvet (Ancey et al.. 1999). The Maurienne massif has an intermediate elevation (highest peak reaches 3,160 m a.s.l.). Its economy is strongly oriented towards winter sports and several of its large ski areas are threatened by avalanches. The Chamonix area is a 34.3 km² area, which is part of the Mont Blanc massif and includes the municipality of Chamonix Mont Blanc. The Chartreuse / Dent de Crolles area is an even smaller area (7.6 km²) located within the Chartreuse massif and with the Dent de Crolles (2,062 m a.s.l.) in its center (Figure S1 in the SM).

| | Chamonix area | Chartreuse / Dent de Crolles area | Chartreuse massif | Mont Blanc massif | Maurienne massif |
|---|---|---|---|---|---|
| Total area [km²] | 34.3 | 7.6 | 847 | 578 | 917.1 |
| Total area covered by CLPA [km²] | 25.8 | 4.7 | 44 | 354.6 | 382.3 |
| Fraction of area covered by CLPA | 75.3% | 61.7% | 5.2% | 61.3% | 41.7% |
| Total area of PRAs within CLPA extents (validation sample) [km²] | 3.6 | 0.5 | 1.6 | 58.3 | 55.7 |
| Total number of PRAs within CLPA extents (validation sample) | 85 | 28 | 85 | 1522 | 1884 |
| Total area of detected PRAs [km²] | 8.1 | 1.2 | 15.4 | 166.3 | 115.1 |
| Total number of detected PRAs | 210 | 58 | 721 | 3676 | 3638 |
| Total area of detected PRAs within area covered by CLPA [km²] | 5.5 | 0.8 | 2.3 | 90.8 | 71.6 |
| Total number of detected PRAs within areas covered by CLPA | 107 | 39 | 108 | 2003 | 2575 |
| Aerial fraction of detected PRAs within the area | 23.7% | 15.3% | 1.8% | 28.8% | 12.5% |
| Aerial fraction of PRAs within the area covered by CLPA | 21.2% | 17.0% | 5.3% | 25.6% | 18.7% |
| Aerial fraction of PRAs within CLPA extents with regards to total area of PRAs | 67.2% | 68.4% | 15.3% | 54.6% | 62.3% |
| Fraction of PRAs numbers within CLPA extents with regards to total number of PRAs | 40.5% | 48.3% | 11.8% | 41.4% | 51.8% |

**Table 1 : Characteristics of studied massifs and areas. For the PRA detection and the determination of the validation sample, all factors and the DEM resolution are set to their default values (Figures 3-4), and forest cover data is from DB forest IGN.**

**2.2 DEM and forest cover**

155 Topographic information used in the proposed PRA detection method is classically derived from a DEM. We primarily used the reference 25 m resolution DEM from the French National Geographical Institute (IGN) as a good compromise to i) detect the right number and areas of PRAs (even if the detection is then certainly less precise in terms of PRA boundaries that with finer resolutions, e.g. Bühler et al., 2013), and ii) being ultimately applicable over very large areas at reasonable computational costs. Sect. 4.2.3 investigates how using DEMs of 160 finer resolutions (also provided by IGN) affects the results.

In addition to the DEM, our method considers forest cover as input data. For the French Alps, three nation-wide forest cover databases can potentially be used:

- The DB forest (database forest) provided by IGN (French National Geographical Institute) obtained from the interpretation of aerial infrared photographs. 32 classes of vegetation are available (notably, as 165 function of tree species, ADEME and IGN, 2019). This data was created in 2013;

- Corine land cover is a database of land cover based on satellite imagery, which considers five classes of vegetation (artificial surfaces, agricultural areas, forests and semi natural areas, wetlands, and water bodies, Caetano et al., 2009). It is updated every 6 years, and the 2012 version was considered for the sake of comparison with the IGN data;

170 - Theia database is provided by the *Centre d'Expertise Scientifique* (CES), which extracts data about land use, soil humidity and snow cover from Sentinel 2A and 2B satellites (Baghdadi et al., 2021). The 2016 version was selected as the closest from 2013.

Our PRA detection method uses forest presence/absence only. We thus derived this information for the three databases in our study areas. None of them is free of errors. Specifically, visual comparison with aerial photographs 175 shows that, logically, main issues occur when the forest density is low, which makes the limit between forests and non-forest areas difficult to set (Figure 2). However, several comparisons in different areas of the studied massifs suggested that the DB forest was, overall, the most accurate and it was thus primarily selected to be used in our PRA detection method. Sect. 4.2.1. assesses the sensitivity to this choice.

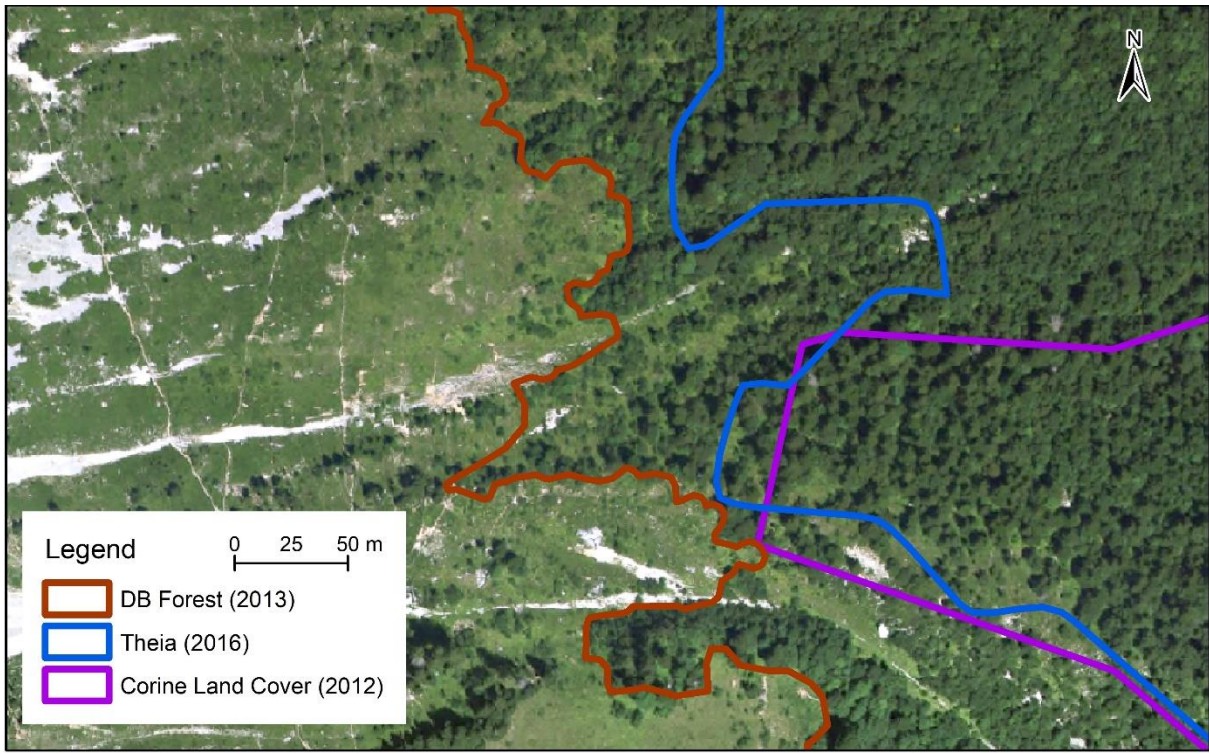

 **Figure 2: Comparison of forest extents from Theia, Corine land cover, and DB Forest from IGN with aerial photographs (©IGN) taken in 2012 within the municipality of Le Sappey-en-Chartreuse, Chartreuse / Dent de Crolles study area.**

**2.3 Avalanche extents from the French avalanche cadastre (CLPA)**

The lack of data concerning historical avalanches contributed to the Val-d'Isère disaster (11 February 1970), where an avalanche led to 39 casualties. Following this tragic event, the French government required the establishment of the CLPA cadastre. Its overall objective is to map the entire avalanche terrain, independently of any frequency consideration. It consists of a collection of maps indicating the maximum extents reached by avalanches in the past. CLPA is obtained using photo-interpretation, terrain observation, historical records, testimonies of residents and mountain professionals such as mountain guides, rescue services and ski resort professionals (Bourova et al., 2016; Naaim-Bouvet and Richard, 2015). CLPA also identifies protection structures (Figure S1 in the SM). The target audience mostly includes mountain professionals (Ancey, 1996). CLPA now covers most of the French Alps, but some areas within the 23 massifs are still completely uncovered (e.g., less than 40% of the Mont Blanc Massif, but nearly 95% of the Chartreuse massif, Table 1). The whole information is freely available at http://www.avalanches.fr (Bonnefoy et al., 2010).

Due to its long history, its regular update by devoted technicians, the continuous financial support of the French ministry of the environment and the consideration in the determination of avalanche terrain of a large amount of different data sources, CLPA is very reliable, meaning that an avalanche extent which is within the CLPA is almost surely a true avalanche extent. By contrast, as all avalanche cadastres, CLPA is not entirely exhaustive. Very rare avalanches may not have occurred since the CLPA exists, and avalanches may have been missed in remote areas. In addition, forest stands that keep the footprints of past events in their landscape forms (e.g., Giacona et al., 2018) are absent in high elevation areas. As a consequence, in areas covered by CLPA, avalanche extents are more exhaustive near human settlements and assets, and less exhaustive in remote areas without stakes and which are difficult to access as well as in high elevation forest-free zones. This is for example the case for high mountain

areas and/or remote valleys, as it is clearly visible in Figure S1 in the SM. Within the same line, CLPA extents are often less exhaustive close to release areas than in runout areas. Yet, CLPA is a very valuable source of information regarding locations where past avalanches occurred, and, among the rare existing cadastres at a spatial scale as large as the entire French Alps (Bourova et al., 2016).

### 3. Proposed PRA detection and evaluation method

The proposed detection of individual PRAs uses topographical information (distance to ridges, slopes, aspect and general curvature) calculated at the pixel scale from the DEM, forest cover extent and a watershed delineation algorithm. Thresholds and parameter values are chosen according to the literature, local peculiarities of the French Alps and a systematic parametric study (Sect. 4.2). Hence, with regards to most of existing methods, main difference is the watershed delineation step, whose underlying idea is similar to the one of the object-based approach of Bühler et al. (2018), namely identifying PRAs corresponding to realistic individual avalanche events. PRAs are detected without any consideration of release frequency, and identified PRAs correspond, for each avalanche path, to a maximal release area.

### 3.1. Calculations at the pixel scale

### 3.1.1 Determination of ridges

To compute the distance to ridges, we use the Geomorphon algorithm of the Grass GIS described in Jasiewicz and Stepinski (2013), which processes the DEM to classify landform elements (ridge, valley) depending on topography. Once ridges have been obtained, the smallest distance to ridges can be evaluated for each pixel of the DEM (it can be equal to 0 when a PRA is in contact with a ridge).

### 3.1.2 Slope, aspect and curvature

Slope is directly obtained as the first derivate of the DEM, and curvature as the second derivate (i.e. first derivate of the slope). Aspect is the maximum slope direction. Concerning curvature, three different quantities can be considered:

- Profile curvature is the curvature of the surface towards the steeper slope;
- Plan curvature is the curvature of the surface transverse to the slope direction;
- General curvature is the curvature of the surface itself. General curvature is positive in convex areas such as ridges, negative in concave areas such as valleys, and null if the plan is horizontal (Zevenbergen and Thorne, 1987). We focus on this last quantity considered as the most relevant for snow avalanching.

Slopes, aspects and general curvatures are obtained using the default option in SAGA GIS (Zevenbergen and Thorne, 1987).

### 3.1.3 Individualization of PRAs using watersheds

To obtain spatial entities which correspond to individual avalanche paths/events, a delineation method is applied as follows (Figure S2 in the SM). First, slopes are calculated for a central pixel and its eight neighbours following Kinner (2003). We then compute the flow direction (Stojkovic et al., 2012). Downward or negative slopes indicate the direction where water flows and provide the flow direction and flow accumulation. The number of accumulated pixels is then obtained for each pixel of the DEM as the sum of all pixels upstream, i.e. which converge in this

direction. Flow accumulation is always non-zero except for pixels located on extremities. Individualized watersheds are obtained from flow accumulation values by identifying the most important flows and by attributing each pixel to one of these flows as detailed in Djokic and Ye (2000). Computations are made with the watershed algorithm of the ARCGIS environment.

### 3.2 The different steps of the proposed PRA detection method

Our detection method is composed of 12 steps (Figure 3). It is based on a DEM with a resolution of 25 m and forest cover extents from IGN DB forest. Following most of existing approaches, it is a binary deterministic classification approach based i) on topographical parameters that do not change with time, and ii) on the presence/absence of forests.

In details, first, we create a layer of points corresponding to filtered pixels of the DEM and for which we evaluate slope, curvature, elevation, aspect, distance to the nearest ridge as stated above, as well as the name of the corresponding massif, latitude, and longitude. To this aim, pixels below the altitude of 1400 m are excluded, as lower elevations receive too little snow in the French Alps under current climate conditions (Durand et al., 2009b) and climate projections clearly indicate a further decrease of snow accumulations in the future for these low elevations (Castebrunet et al., 2014; Verfaillie et al., 2018). Besides, following Maggioni and Gruber (2003), only pixels with a slope between 28° and 60° are kept. Also, only pixels situated at less than 600 m from the closest ridge are further considered. Figure S3 in the SM shows the pdf of the distance to the closest ridge for the study area close to Chamonix, which quickly decreases with distance and is close to zero above 600 m. Hence, even if this latter filter makes sense due to the impact of snowdrift on avalanche release, it generally affects limited areas. Finally, we remove areas covered by a dense forest according to the DB forest of IGN and the pixels kept are converted to a layer of points. The following processing step consists in applying the delineation algorithm to individualize each PRA (Sect. 3.1.3) and only polygons with a minimal planar area of ten pixels (i.e., 6250 m$^2$) are conserved. The resulting polygons are converted to a vector layer. For each polygon corresponding to a PRA, different PRA-scale attributes are stored: distance to the closest ridge, name of the corresponding massif, slope, aspect, elevation, curvature, latitude, longitude, and planar area.

**(a) Calculation of topographic parameters for each pixel**

**(b) Identification of polygons corresponding to individual PRAs**

DEM

*(1) Keep a minimal elevation of 1400 m*

*(2) Choose slope between 28° and 60°*

*(3) Remove forested parts*

*(4) Distance to ridge lower than 600 m for each pixel*

DEM with a minimal elevation of 1400 m, slope between 28° and 60°, without forest part and distance to ridge lower than 600 m

*(5) Calculate curvature and slope*
*(6) Transform curvature into a point layer*
*(7) Calculate distance to ridge for each point*

Point layer with values of slope, curvature and distance to ridge lower than 600 m

*(8) Add values of: slope, curvature, aspect, distance to ridge, elevation, latitutde, longitude, name of massif as attributes*

Point layer with values of: slope, curvature, aspect, distance to ridge, elevation, latitude, longitude, name of massif as attributes

*(9) Delineate watersheds*
*(10) Add values of: distance to ridge and the name of massif (from the point layer at the step 8), slope, aspect, elevation, curvature, latitude, longitude, name of massif as attributes*

Watersheds outside forest, with values of slope, aspect, elevation, curvature, distance to ridge, latitude, longitude, name of massif as attributes

*(11) Calculate area*
*(12) Select watersheds with a minimal area of 10 pixels*

Final layer of polygons (vector) with values of: slope, aspect, elevation, curvature, distance to ridge, area, latitude, longitude, name of massif as attributes

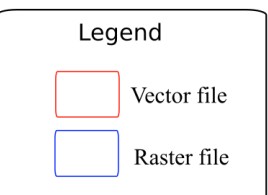

Legend

Vector file

Raster file

265

**Figure 3: The 12 steps of the proposed PRA detection method. Calculations at (a) the pixel scale, (b) at the scale of identified polygons.**

### 3.3 Processing of the CLPA for PRA evaluation

Thanks to the 50-year history of CLPA, the mapped avalanche extents it contains are getting closer and closer to the true maximal extent of avalanche terrain. The CLPA cadastre is therefore a good candidate to be used as ground truth to validate our PRA detection method. However, a direct comparison is meaningless. Indeed, i) there is simply no CLPA at all in some areas of the French Alps (Table 1), ii) path boundaries in CLPAs are not systematically mapped, especially in release areas (Figure S1 of the SM), iii) the CLPA documents the entire maximum extents of observed past avalanches (Figure S1 of the SM), whereas our PRA detection approach only focuses on release areas. Hence, i) our evaluation focuses on the fractions of massifs / study-areas covered by CLPA, and ii) in these, CLPA avalanche extents are processed as follows in order to generate a validation sample that can be compared with our PRAs. First, all boundaries of the CLPA polygons are merged together, leading a single polygon layer representing the maximal extent of past avalanches according to available records, testimonies

and photo-interpretation insights. Second, the same criteria of slope, minimal elevation, distance to ridge, and presence of forest as for the PRA detection are used to filter this polygon. Finally, individual PRAs are identified with the watershed delineation algorithm, and those with a minimal area of 6250 m$^2$ are kept (Figure 4). We use the concept of "validation sample" as it is taken as the "ground truth". However, given that this sample and the overall comparison scheme are not free of discussible limitations (Sect. 5.3), we prefer speaking of an evaluation rather than of a validation approach. Sect. 5.3 discusses its pro's and con's.

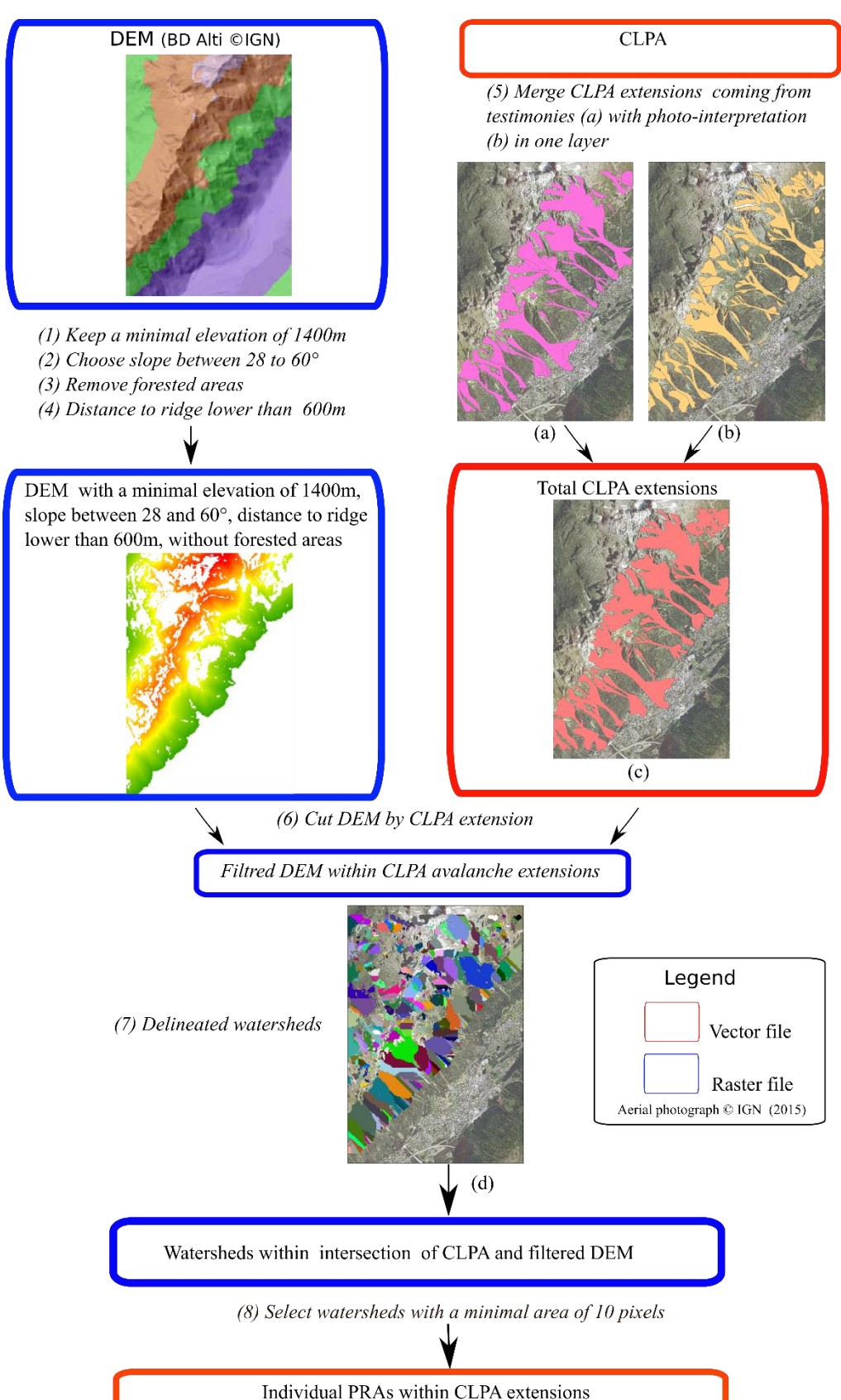

DEM (BD Alti ©IGN)

CLPA

*(5) Merge CLPA extensions coming from testimonies (a) with photo-interpretation (b) in one layer*

*(1) Keep a minimal elevation of 1400m*
*(2) Choose slope between 28 to 60°*
*(3) Remove forested areas*
*(4) Distance to ridge lower than 600m*

(a) (b)

DEM with a minimal elevation of 1400m, slope between 28 and 60°, distance to ridge lower than 600m, without forested areas

Total CLPA extensions

(c)

*(6) Cut DEM by CLPA extension*

*Filtred DEM within CLPA avalanche extensions*

*(7) Delineated watersheds*

Legend

Vector file

Raster file

Aerial photograph © IGN (2015)

(d)

Watersheds within intersection of CLPA and filtered DEM

*(8) Select watersheds with a minimal area of 10 pixels*

Individual PRAs within CLPA extensions

285

**Figure 4: The French avalanche cadaster (CLPA) and the processing steps used to identify individual PRAs within CLPA avalanche extents ("avalanche extensions") as a validation sample for the proposed PRAs detection method. (a) CLPA avalanche extents coming from testimonies, (b) CLPA avalanche extents coming from photo-interpretation, (c) Union of CLPA avalanche extents, (d) Delineation of individual PRAs within CLPA avalanche extents. Aerial photograph ©IGN 2015.**

290

**3.4 Confusion matrices and evaluation scores**

Confusion matrices (Table 2) can be obtained from the comparison between the detected PRAs and the processed CLPA extents (Sect. 3.3), the latter being considered as a reference dataset (ground truth). A confusion matrix includes four numbers (or rates, i.e. standardized numbers): true positives (TP), true negatives (TN), false positives (FP), and false negatives (FN). A true positive means that the prediction and the reference values match, i.e. detected PRAs match processed CLPA extents. A false positive means that a PRA is detected outside the processed CLPA extents. True negatives correspond to areas which are neither detected by our PRA detection method nor included in processed CLPA extents, and false negatives to processed CLPA extents that are not detected by our method.

|  |  | **Detected PRAs (Figure 3)** | |
|---|---|---|---|
|  |  | Yes | No |
| **Processed CLPA extents (Figure 4)** | Yes | True positive (TP) | False negative (FN) |
|  | No | False positive (FP) | True negative (TN) |

**Table 2 : Principle of a confusion matrix and application to our PRA detection method.**

Accuracy and error rates that summarize the confusion matrix are classically computed as follows:

$$Accuracy\ rate = \frac{True\ positive+True\ negative}{True\ positive+True\ negative+False\ positive+False\ negative}\ , \tag{1}$$

$$Error\ rate = \frac{False\ positive+False\ negative}{True\ positive+True\ negative+False\ positive+False\ negative}\ . \tag{2}$$

As the CLPA processing method (Figure 4) applies the same filters than our PRA detection method and because the comparison is restricted to the area covered by CLPA extents (Sect. 3.3), by construction, only true positives can be confidently assessed. Our evaluation therefore focuses on the true positive rate (TPR) also know as recall or sensitivity of the method:

$$TPR\left(recall\right)=\frac{TP}{TP+FN}\ , \tag{3}$$

Finally, to evaluate the ability of our method to detect i) the right number of PRAs and ii) their correct extents, the confusion matrix is computed both in terms of areas (comparison of the areas of the polygons) and numbers (comparison of the number of polygons). In terms of numbers, a detected PRA and a validation polygon match as soon as their intersection is non-zero. The confusion matrix in area is computed by evaluating intersected areas.

**3.5 Robustness of the detection and evaluation method**

In order to tune the different factors involved in the PRA detection method, and, more generally, to assess the robustness of our PRA detection and evaluation approach, our scores (Eqs. 1-3) were computed for a large number of input factors and data sets. In details, we first quantified to which extent the marginal effect of the different factors and data sources involved in our detection method gradually increase evaluation scores. We then more deeply analysed the sensitivity to the most critical factors using a parametric study. In addition, we assessed the impact of the DEM resolution, not only on scores but also on the determination of the validation sample and the subsequent changes in PRA detection. Small study areas were used to understand and showcase detailed results, but the efficiency of the method was primarily analysed in terms of massif-scale scores, which provide a much more systematic assessment free of local effects.

## 4. Results

### 4.1 Results for study areas and massifs

We first illustrate the results of the PRA detection method with the default factor values and data sets (Figures 3-4) in the study area of Chamonix (Figure 5). Avalanche extents from the CLPA, when processed according to Sect. 3.3, lead a validation sample of 85 individual PRAs in this area (PRAs/CLPA, Table 1). For the same area, the automated detection method leads 107 PRAs within the area covered by CLPA, and 103 PRAs outside (PRAs/AUTO outside the area covered by CLPA). The latter are mainly in remote areas not covered by CLPA, the city of Chamonix being surrounded by high mountains, which are difficult to access and document. Among the 107 PRAs detected within the area covered by CLPA, 90 intersect the processed CLPA extents (PRAs/AUTO inside the areas covered by CLPA and matching PRA/CLPAs). By contrast, 17 PRAs detected within the area covered by CLPA do not intersect the processed CLPA extents at all (PRAs/AUTO inside the areas covered by CLPA and not matching PRA/CLPAs).

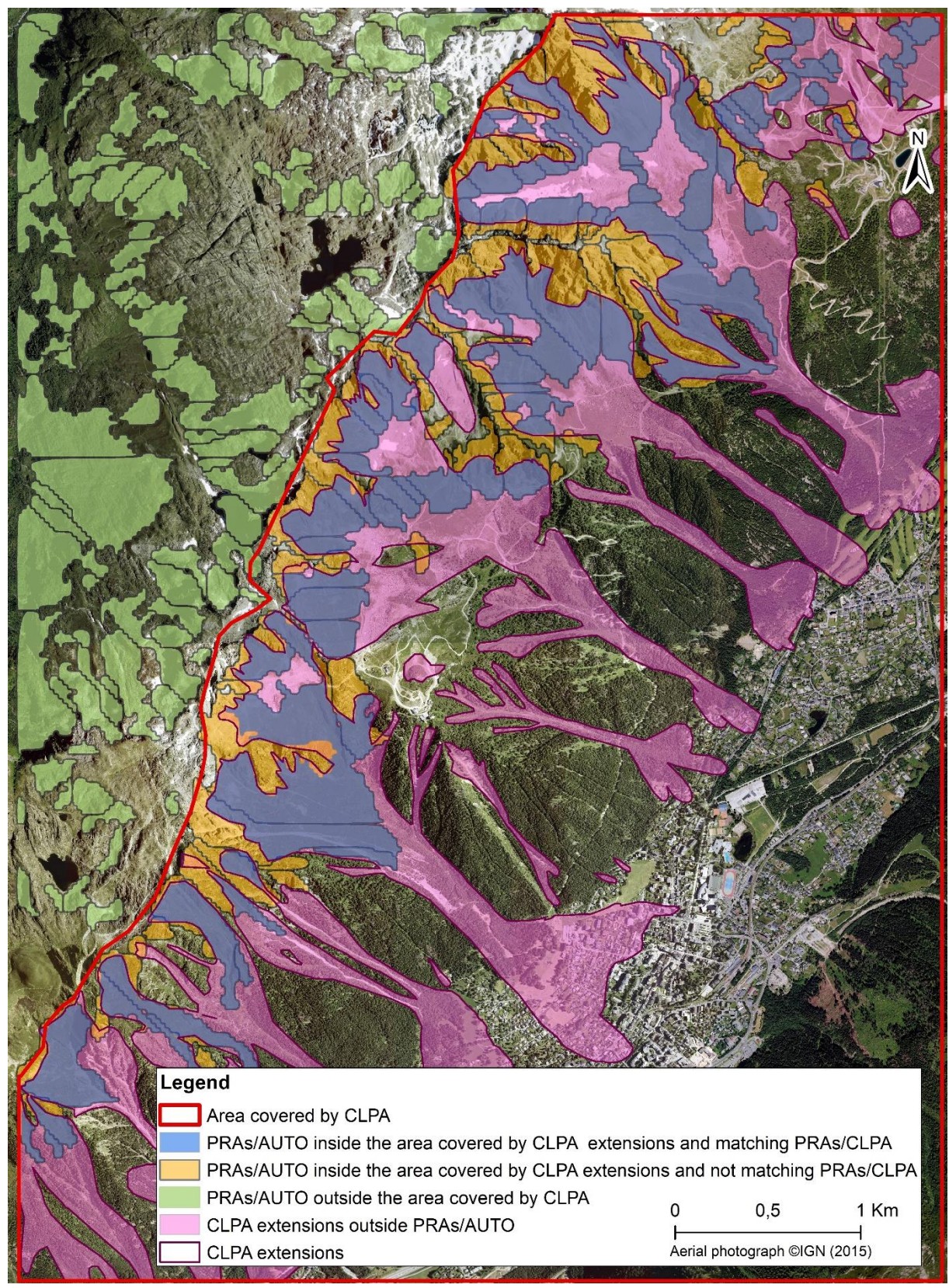

**Legend**

☐ Area covered by CLPA

☐ PRAs/AUTO inside the area covered by CLPA extensions and matching PRAs/CLPA

☐ PRAs/AUTO inside the area covered by CLPA extensions and not matching PRAs/CLPA

☐ PRAs/AUTO outside the area covered by CLPA

☐ CLPA extensions outside PRAs/AUTO

☐ CLPA extensions

0        0,5        1 Km

Aerial photograph ©IGN (2015)

**Figure 5: Result of the proposed PRA detection method for the study area of Chamonix. Agreement and mismatches with the avalanche extents from the French avalanche cadastre ("CLPA extensions") are highlighted. For the PRA detection and the determination of the validation sample, all factors and the DEM resolution are set to their default values (Figures 3-4), and forest cover data is from DB forest IGN.**

340

The two confusions matrices are computed for the areas covered by CLPA only. They evaluate the performances of the detection method in terms of number and surface of detected PRAs. In numbers, the 90 detected PRAs that intersect the processed CLPA extents represent 84.1% of the total number of detected PRAs within the areas covered by CLPA (true positive rate, Eq. 3). The false positive rate (15.9%) complements the true positive rate, and 100% of PRAs within CLPA extents are detected according to our evaluation approach, by constraint (Table 3). This leads an accuracy rate of 92.1% and an error rate of 7.9% in numbers (Eqs. 1-2). In term of areas, the total area covered by the detected PRAs in the areas covered by CLPA is 5.48 km$^2$ (Table 1), and, among them, 5.29 km$^2$ intersect CLPA extents (true positive rate of 96.5 %). This leads an accuracy rate of 98.3 %, and consequently an error rate of 1.7% in terms of areas.

| Confusion matrix in areas [km$^2$] (%) | | Confusion matrix in numbers (%) | |
|---|---|---|---|
| 5.29 km$^2$ (96.5%) | 0 (0%) | 90 (84.1%) | 0(0%) |
| 0.19 km$^2$ (3.5%) | 3.56 km$^2$ (100%) | 17 (15.9%) | 85 (100%) |

**Table 3 : Confusion matrix (Table 2) in areas (%) and numbers (%) for the study area of Chamonix. For the PRA detection and the determination of the validation sample, all factors and the DEM resolution are set to their default values (Figures 3-4), and forest cover data is from DB forest IGN.**

We now switch to massif-scale results and focus on true positive rates that, given our evaluation scheme, summarize all relevant information concerning the efficiency of the PRA detection. For the Chartreuse massif, the PRA detection method with the default factor values and data sets lead 721 individual PRAs, for a total area of 15.4 km$^2$, which represents 1.8% of the surface of the massif. Logically, detected PRAs are concentrated close to the main ridges, where the only areas in the Chartreuse massif that are both high and steep enough and forest free are located (Figure 6). The area covered by CLPA is located on the east flank of the massif along a main ridge and close to the large Gresivaudan valley, which is densely urbanized. Visually, the matching between the PRAs within processed CLPA extents and the automatically detected PRAs appears satisfactory. True positive rates reach 87% in numbers and 92.4% in areas. Due to the small fraction of the massif covered by the CLPA, the total area of detected PRAs within the area covered by CLPA is 2.3 km$^2$ only, which corresponds to 108 individual PRAs. These are 11.8% and 15.3% of detected PRA numbers and areas in the massif, respectively. Finally, as the CLPA focuses of the part of the massif where avalanche activity is more likely, the aerial fraction of detected PRAs is higher within the area covered by CLPA (5.3%) than at the massif scale (Table 1).

Figures 7-8 display massif-scale results for the Mont-Blanc and Maurienne massifs. Both massifs are more prone to avalanche activity that the Chartreuse massif due to their topography and elevation. As a consequence, total numbers and areas of detected PRAs are much higher than in the Chartreuse massif. Notably, the aerial fraction of detected PRAs is 12.5% in the Maurienne massif and peaks at 28.8% in the high elevation Mont Blanc massif (Table 1). Also, both the Mont Blanc and Maurienne massifs are much more largely covered by CLPA than the Chartreuse massif, so that around half of the PRA numbers and areas detected in these massifs are within CLPA extents (Table 1). Hence, in both massifs, evaluation scores are computed over large validation samples.

Table 4 sums up obtained true positive rates for all massifs and study areas. They are always high, especially in areas (92.4-94% for the tree massifs). The lowest true positive rate is in numbers for the massif of Mont-Blanc (80%). It can probably be explained by the fact that avalanche activity is, on average, more exhaustively documented close to release areas within the extents of CLPAs in the massifs of Chartreuse and Maurienne, making the validation sample more comprehensive in these massifs. The reason is that, in the Mont-Blanc massif, many

avalanche release areas included within the area covered by CLPA (South-East of Figure 7) are located at high elevations and in remote zones, and very far from any forest stand. Hence, related avalanche activity was missed during the establishment of CLPAs, as no insights from past snow avalanches could be retrieved, either from testimonies or from photo-interpretation (visual detection of avalanche corridors in forested slopes), which led to missing PRAs in the validation sample. By comparison, in Maurienne and Chartreuse, PRAs in areas covered by CLPA are, on average, located closer to valleys and forest stands, making arguably CLPAs, and, hence, validation samples more accurate.

|  |  | Chamonix area | Chartreuse / Dent de Crolles area | Chartreuse Massif | Mont-Blanc Massif | Maurienne Massif |
|---|---|---|---|---|---|---|
| **True positive rate (recall), Eq. 3** | **In numbers (%)** | 84.1 | 87.2 | 87 | 80 | 82.8 |
| | **In areas (%)** | 96.5 | 80.4 | 92.4 | 93.6 | 94 |

**Table 4** *: Summary of true positive rates calculated in numbers and areas for the different areas and massifs. For the PRA detection and the determination of the validation sample, all factors and the DEM resolution are set to their default values (Figures 3-4), and forest cover data is from DB forest IGN.**

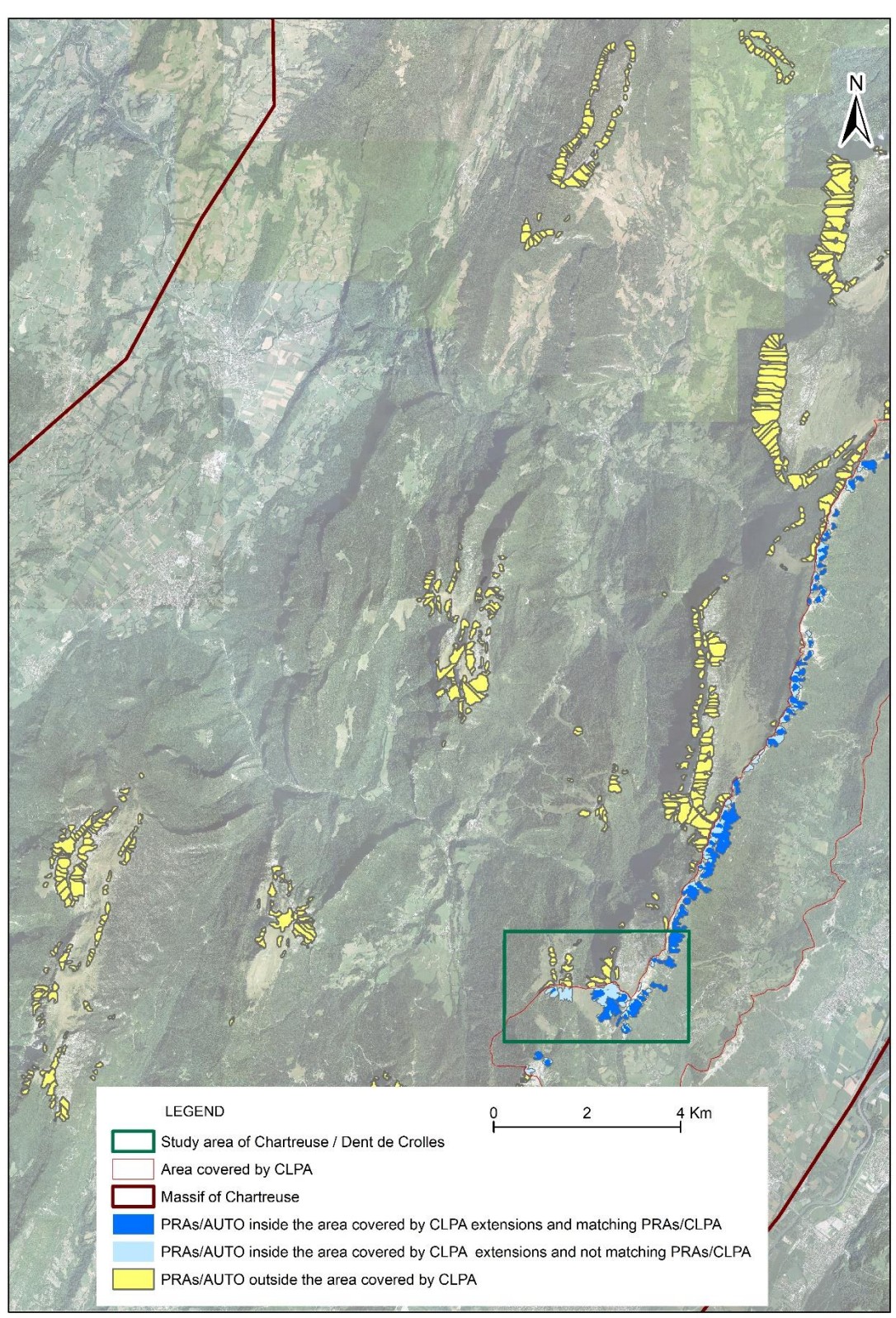

390

**Figure 6 : Result of the proposed PRA detection method for the entire Chartreuse massif. For the PRA detection and the determination of the validation sample, all factors and the DEM resolution are set to their default values (Figures 3-4), and forest cover data is from DB forest IGN. "CLPA extensions" refer to avalanche extents from the French avalanche cadastre. Aerial photograph ©IGN 2012.**

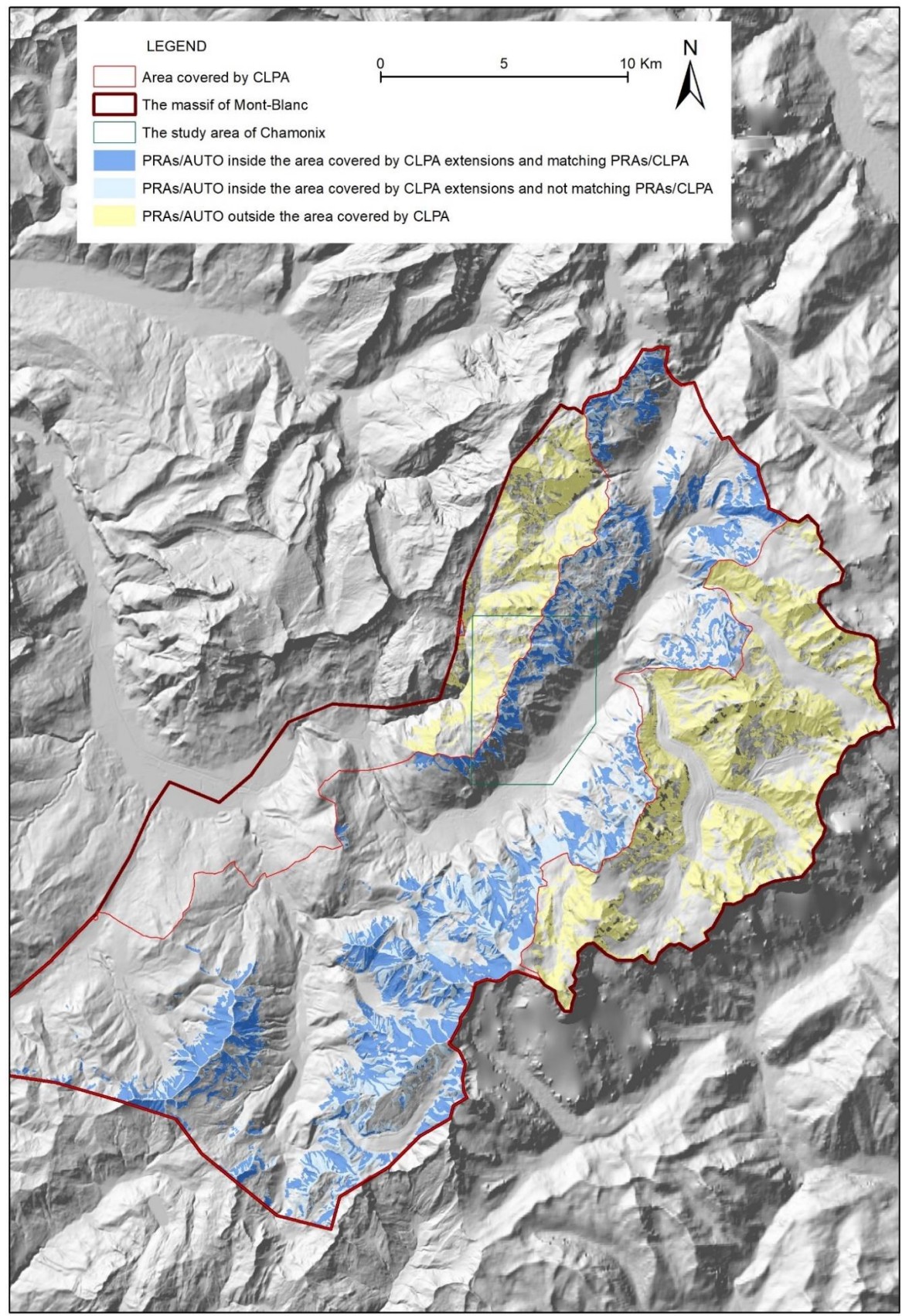

395

**Figure 7 : Result of the proposed PRA detection method for the entire Mont Blanc massif. Digital Elevation Model ©IGN. For the PRA detection and the determination of the validation sample, all factors and the DEM resolution are set to their default values (Figures 3-4), and forest cover data is from DB forest IGN. "CLPA extensions" refer to avalanche extents from the French avalanche cadastre.**

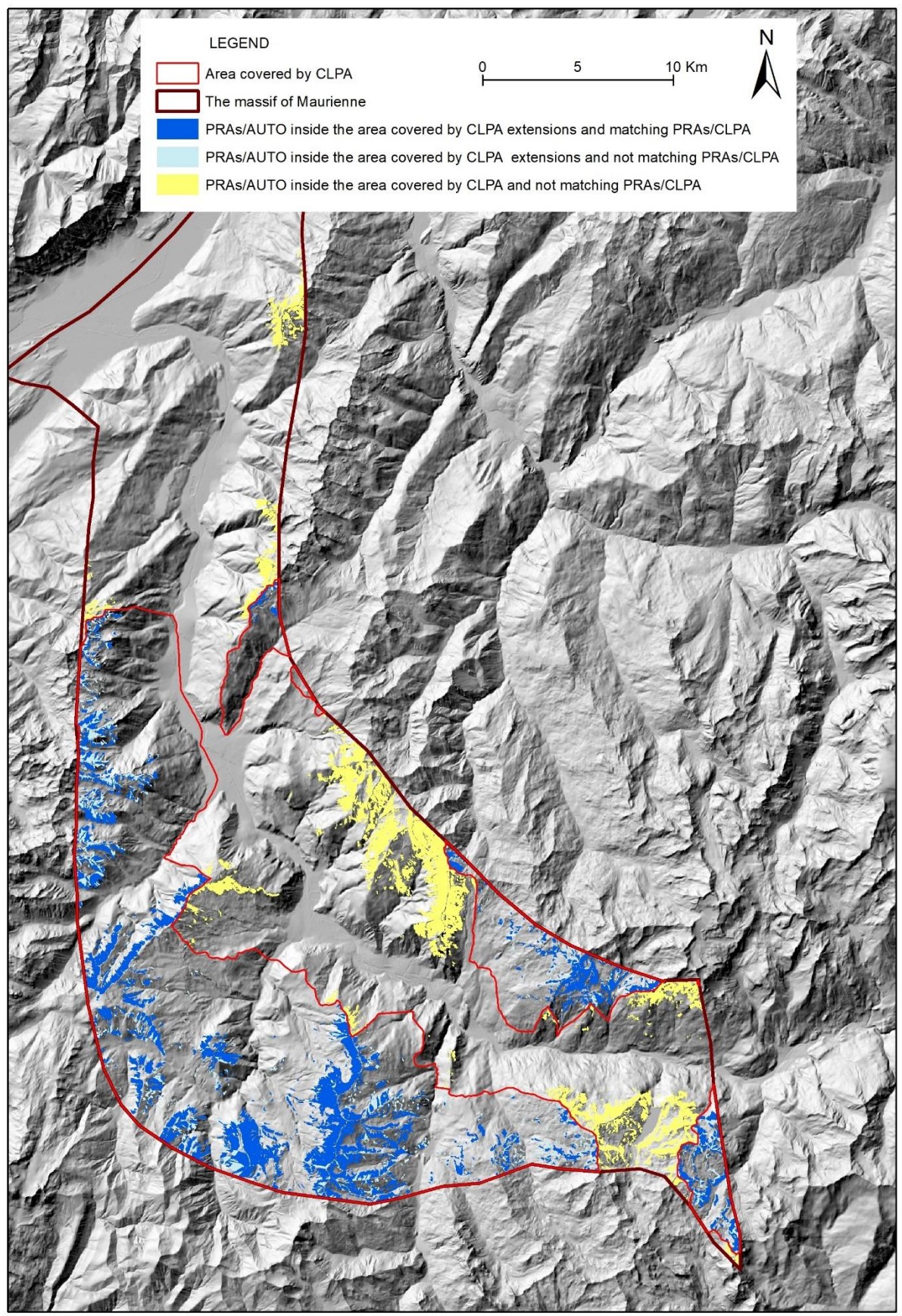

400

**Figure 8 : Result of the proposed PRA detection method for the entire Maurienne massif. Digital Elevation Model ©IGN. For the PRA detection and the determination of the validation sample, all factors and the DEM resolution are set to their default values (Figures 3-4), and forest cover data is from DB forest IGN. "CLPA extensions" refer to avalanche extents from the French avalanche cadastre.**

## 4.2 Robustness of the detection method

### 4.2.1 Marginal effect of the different factors and data sets

Table 5 shows the marginal effect on detected PRA numbers and areas and true positive rates of the different factors and data sources for the Mont Blanc massif. Corresponding results for the Chartreuse and Maurienne massif are provided in Tables S1-S2 of the SM. At this stage, the same validation sample obtained with the default setting is always considered. Main result is that, in terms of PRA numbers, the default choices lead always the best true positive rates in the three massifs. The results are almost similar for PRA areas, except that very slight increases in true positive rates are obtained for the Maurienne massif i) without the watershed delineation step (+2.4%) and ii) with forest cover extents form Corine Land Cover instead of DB forest from IGN (+0.2%). This overall result supports in a pragmatic way the usefulness of the different steps of the method, and the choices made, even if the effect of one specific filter or data set choice can be relatively minor (e.g. true positive rates are often quite similar with/without certain filters or with two competing data sources).

In more details, regarding the four different topographic filters (minimal elevation, slope range, minimal area size and maximal distance to ridge,) in all three massifs, the minimal area size has a strong impact on true positive rates, notably in terms of PRA numbers. Its influence on PRA areas is much lower, because this filter removes many small PRAs, modifying the overall PRA extent only slightly. Also, for the three massifs, the minimal elevation threshold has a limited impact, both on PRA numbers and areas, with decreases in true positive rates never exceeding 5.4%. Yet, without it, a few PRAs are detected at low elevations, notably in rather unrealistic locations (if avalanches were actually released at these locations, testimonies would definitely have been available and included in CLPAs). The two other filters have an effect which is more variable from one massif to another. The effect of the maximum distance to ridges is substantial in the Maurienne massif (with a 21.8-23.4% decrease in the true positive rate when it is not used) but moderate in the Mont Blanc massif and null in the Chartreuse massif, as in the latter massif distance to the closest ridge is usually below the default 600 threshold. Lastly, the effect of the slope range filter is particular strong on PRA numbers in Maurienne (with a 36.8% decrease in the true positive rate when it is not used), whereas it has limited effect on PRA detection in the two other massifs. The reason is probably that, in both the Mont Blanc and Chartreuse massifs, avalanche terrain is very steep, so that the filter is not very restrictive.

Comparison of the performances with/without the application of the watershed delineation algorithm shows that this step may affect the efficiency of PRA detection in a crucial way. The gain in true positive rates is very strong in the Mont Blanc massif (up to almost 76%, Table 5), more moderate in the Chartreuse massif (4.4-5.2% in numbers and areas, respectively) and contrasted in the Maurienne massif, where this step largely improves the true positive rate in PRA numbers (by 28.6%) but slightly decreases (by 2.8%) the true positive rate in PRA areas. The reason is that keeping PRAs of at least 6250m$^2$ (10 pixels) only removes very small areas when the watershed delineation is used that are kept when it is not. Yet, this effect is of limited impact, and more than largely compensated by the large increase in the true positive rate regarding PRA numbers / individualisation.

Finally, we compared to the default results the results obtained, i) with the two other nation-wide forest databases introduced in Sect. 2.2, or ii) without any forest cover filter. When forested areas are not removed at all, a much higher number of PRAs covering large areas is detected. This leads decreases in true positive rates exceeding 20%

in the Mont Blanc and Maurienne massifs. This decrease in the true positive rate reaches 33% in numbers in the Chartreuse massif, which is logical, as it is the massif where the forest is the most widespread among the massifs we study. This result shows the interest of the forest filter to exclude correctly many areas where avalanche release are virtually impossible. Comparing the different forest databases to each other's shows that the best true positive rates are obtained with the DB forest from IGN, but with drops in true positive rates not exceeding 20% both in numbers and areas when any of the two other candidates is used. A detailed analysis in the study area of Chamonix (Figure 9) confirms that, i) for some low elevation areas, Corine land cover and Theia both indicate dense forests that actually do not exist according to the aerial photograph, ii) forested areas provided by Theia are, at some locations, too segmented with regards to reality. These artefacts may explain the slightly better results obtained with the DB forest from IGN. Indeed, with the latter, detected PRAs in the area mostly only correspond to high elevations that are forest-free according to aerial photographs. Only one PRA remains detected at a low elevation near the tower of a chairlift. This corresponds to an area where a large hiking trail eroded by walkers prevents the forest development, making an avalanche release actually possible, as slope is favorable.

| | | With default values | Without minimal elevation filter | Without slope filter | Without distance to ridge filter | Without minimal area filter | Without watershed delineation | Without forest filter | With Corine Land Cover forest | With Theia forest |
|---|---|---|---|---|---|---|---|---|---|---|
| **Validation sample** | Total area of PRAs within CLPA (validation sample) [km²] | 58.3 | 58.3 | 58.3 | 58.3 | 58.3 | 58.3 | 58.3 | 58.3 | 58.3 |
| | Total number of PRAs within CLPA extents (validation sample) | 1522 | 1522 | 1522 | 1522 | 1522 | 1522 | 1522 | 1522 | 1522 |
| **Detected PRAs** | Total area of detected PRAs [km²] | 90.8 | 98.1 | 97.9 | 102.3 | 102.8 | 96.6 | 135.1 | 109.0 | 104.1 |
| | Difference in area with regards to default values [km²] | / | 7.33 | 7.15 | 11.55 | 11.98 | 5.80 | 44.31 | 18.21 | 13.34 |
| | Difference in area with regards to default values (%) | / | 8.1% | 7.9% | 12.7% | 13.2% | 6.4% | 48.8% | 20.1% | 14.7% |
| | Total number of detected PRAs | 2003 | 2085 | 2074 | 2173 | 4315 | 262 | 2803 | 2170 | 2218 |
| | Difference in numbers with regards to default values | / | 82 | 71 | 170 | 2312 | -1741 | 800 | 167 | 215 |
| | Difference in numbers with regards to default values (%) | / | 4.1% | 3.5% | 8.5% | 115.4% | -86.9% | 39.9% | 8.3% | 10.7% |
| | Total area of detected PRAs within CLPA extents [km²] | 84.9 | 86.6 | 86.6 | 89.3 | 87.1 | 17.2 | 102.3 | 93.1 | 89.8 |
| | Total number of detected PRAs within CLPA extents | 1601 | 1622 | 1623 | 1626 | 2061 | 111 | 1597 | 1528 | 1607 |
| **Evaluation** | True positive rate (recall), Eq. 3 — In numbers (%) | 80 | 77.8 | 78.2 | 74.8 | 47.8 | 42.2 | 57 | 66.6 | 71.8 |
| | True positive rate (recall), Eq. 3 — In areas (%) | 93.6 | 88.2 | 88.4 | 87.2 | 84.8 | 17.8 | 75.8 | 83.8 | 86 |
| | Difference in recall with regards to default values — In numbers (%) | / | -2.2 | -1.8 | -5.2 | -32.2 | -37.8 | -23 | -13.4 | -8.2 |
| | Difference in recall with regards to default values — In areas (%) | / | -5.4 | -5.2 | -6.4 | -8.8 | -75.8 | -17.8 | -9.8 | -7.6 |

**Table 5 : Marginal effect on true positive rates of the different factors and data sources, Mont Blanc Massif. Total area of detected PRAs and total number of detected PRAs are those of the part of the massif covered by CLPA (Table 1). For the PRA detection, in each column, only the considered factor is removed/varies according to the column label. All other factors and the DEM resolution are set to their default values (Figure 3). For the determination of the validation sample, all factors and DEM resolution are set to their default values (Figure 4), and forest cover data is from DB forest IGN.**

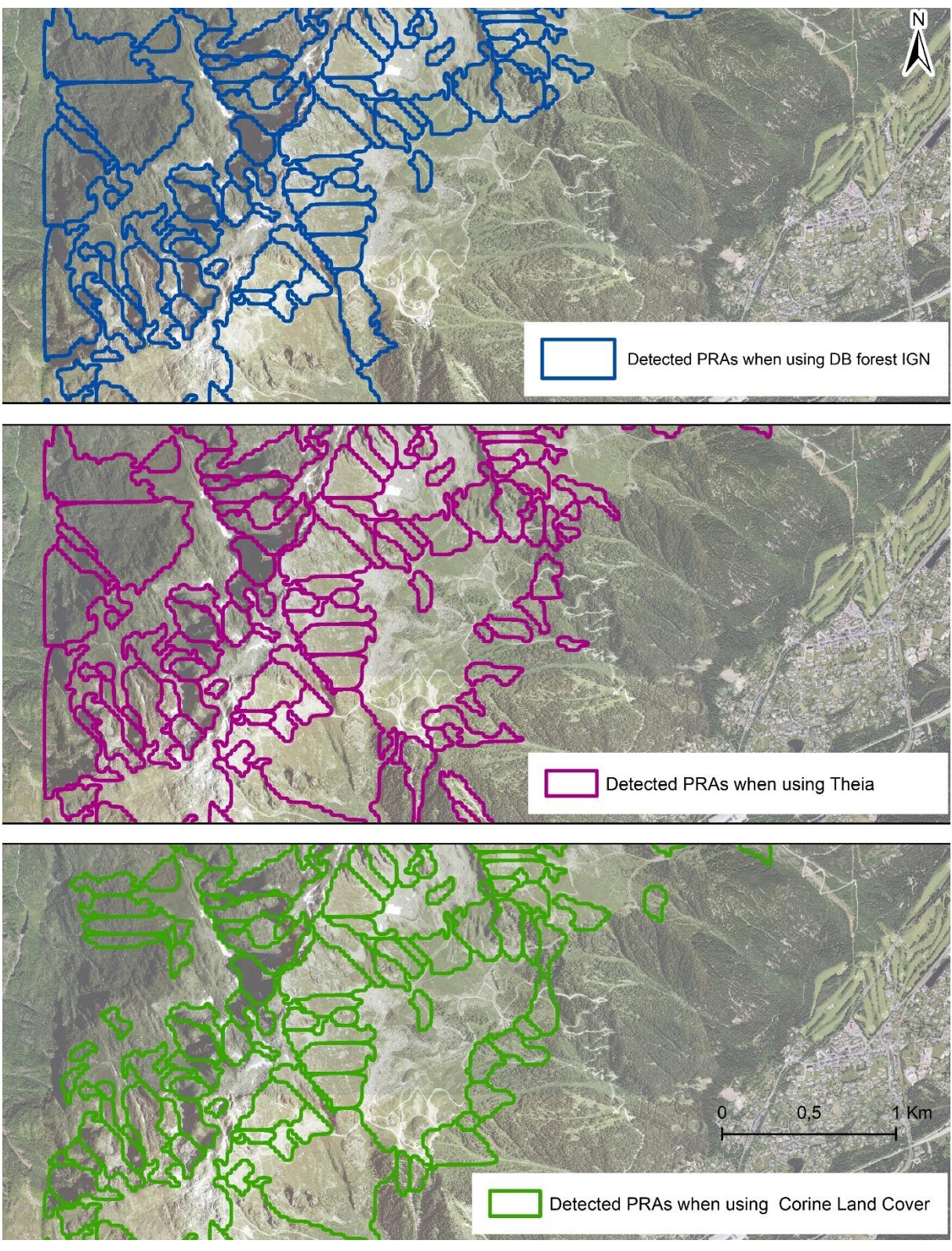

**Figure 9 : Effect of forest data source on detected PRAs, study area of Chamonix. For the PRA detection, forest data source varies, and all other factors and the DEM resolution are set to their default values (Figure 3). Aerial photograph ©IGN 2015. For the determination of the validation sample, all factors and DEM resolution are set to their default values (Figure 4), and forest cover data is from DB forest IGN.**

465

**4.2.2 In-depth parametric study of the influence of the different topographic filters**

We additionally conducted a more systematic search, varying the different topographic filters in the PRA detection over a range of values plausible according to the literature and local peculiarities of the French Alps (elevation range and corresponding snow cover characteristics), again keeping always the same validation sample. Results at the scale of the Mont Blanc massif show that, overall, true positive rates decrease with distance to the default values (Table 6). True positive rates seem nevertheless rather stable over considered ranges of parameters/thresholds, with slope range being the most influential parameter over the tested range (up to a 18% decrease in the true positive rate for numbers).

Regarding differences in the detected PRA numbers and areas, they can be larger, especially with slope range for PRA areas, and with slope range and minimal area size for PRA numbers. For instance, slope ranges more restrictive than our default choice lead to decreases in detected PRAs up to 43% in areas and up to ~25% in numbers for the Mont Blanc Massif. By contrast, the minimal area size strongly affects PRA numbers, but PRA areas much less, because increasing the minimal area size gradually discards the small PRAs below the considered threshold (Table 6). Detailed analyses of specific areas such as the Chartreuse / Dent de Crolles area confirm that a too large minimal area size (Figure 10), a too high minimal elevation (Figure S4 of the SM) or a too restricted slope range (Figure S5 of the SM) misses release areas that an expert analysis would definitely consider as suitable locations for avalanche releases. This is the case, e.g., of the "small" PRAs detected in Figure 10a with the default minimal area size, but missed in Figure 10c with a two times larger minimal area size.

| | | With default values | Minimal area (m²) | | | Minimal elevation (m) | | | Slope range (°) | | | | Maximal distance to ridge (m) | | | |
|---|---|---|---|---|---|---|---|---|---|---|---|---|---|---|---|---|
| | | | 3125 | 9375 | 12500 | 1200 | 1600 | 1800 | [26-60] | [30-60] | [32-60] | [34-60] | 400 | 500 | 700 | 800 |
| Total area of detected PRAs [km²] | | 90.8 | 93.8 | 88.2 | 85.2 | 90.7 | 88.9 | 88.7 | 90.8 | 64.5 | 58.9 | 51.8 | 81.0 | 88.5 | 94.3 | 95.0 |
| Difference in area with regards to default values [km²] | | / | 3.0 | -2.6 | -5.6 | -0.1 | -1.9 | -2.1 | 0.0 | -26.3 | -31.9 | -39.0 | -9.8 | -2.3 | 3.5 | 4.2 |
| Difference in area with regards to default values (%) | | / | 3.3% | -2.8% | -6.2% | -0.1% | -2.1% | -2.4% | 0.0% | -28.9% | -35.1% | -43.0% | -10.8% | -2.6% | 3.9% | 4.6% |
| Total number of detected PRAs | | 2003 | 2632 | 1654 | 1369 | 2000 | 1979 | 1941 | 2002 | 1598 | 1582 | 1505 | 1877 | 2008 | 2088 | 2104 |
| Difference in numbers with regards to default values | | / | 629 | -349 | -634 | -3 | -24 | -62 | -1 | -405 | -421 | -498 | -126 | 5 | 85 | 101 |
| Difference in numbers with regards to default values (%) | | / | 31.4% | -17.4% | -31.7% | -0.1% | -1.2% | -3.1% | 0.0% | -20.2% | -21.0% | -24.9% | -6.3% | 0.2% | 4.2% | 5.0% |
| Total area of detected PRAs within CLPA extents [km2] | | 84.9 | 85.7 | 83.3 | 81.2 | 84.7 | 83.1 | 76.6 | 84.8 | 61.9 | 56.3 | 49.6 | 75.3 | 82.3 | 87.5 | 87.5 |
| Total number of detected PRAs within CLPA extents | | 1601 | 1768 | 1391 | 1201 | 1590 | 1589 | 1520 | 1597 | 1409 | 1406 | 1349 | 1468 | 1576 | 1622 | 1621 |
| True positive rate (recall), Eq. 3 | In numbers (%) | 80 | 67.1 | 79.8 | 78.5 | 79.2 | 78.5 | 68.8 | 79.7 | 73.8 | 68.4 | 61.9 | 70.5 | 76.2 | 67.2 | 76.9 |
| | In areas (%) | 93.6 | 91.4 | 93.6 | 93.4 | 93.3 | 93.0 | 82.6 | 93.3 | 92.0 | 89.5 | 85.1 | 90.0 | 92.3 | 85.2 | 92.1 |
| Difference in recall with regards to default values | In numbers (%) | / | -12.9 | -0.2 | -1.5 | -0.8 | -1.5 | -11.2 | -0.3 | -6.2 | -11.6 | -18.1 | -9.5 | -3.8 | -12.8 | -3.1 |
| | In areas (%) | / | -2.2 | 0.0 | -0.2 | -0.3 | -0.6 | -11.0 | -0.3 | -1.6 | -4.1 | -8.5 | -3.6 | -1.3 | -8.4 | -1.5 |

Table 6 : Parametric study: effect of minimal area size, minimal elevation, slope range and maximal distance to ridge on PRA detection, Mont Blanc Massif. Total area of detected PRAs and total number of detected PRAs are those of the part of the massif covered by CLPA (Table 1). For the PRA detection, in each column, only the considered factor varies. All other factors and the DEM resolution are set to their default values (Figure 3), and forest cover data is from DB forest IGN. For the determination of the validation sample, all factors and DEM resolution are set to their default values (Figure 4), and forest cover data is from DB forest IGN.


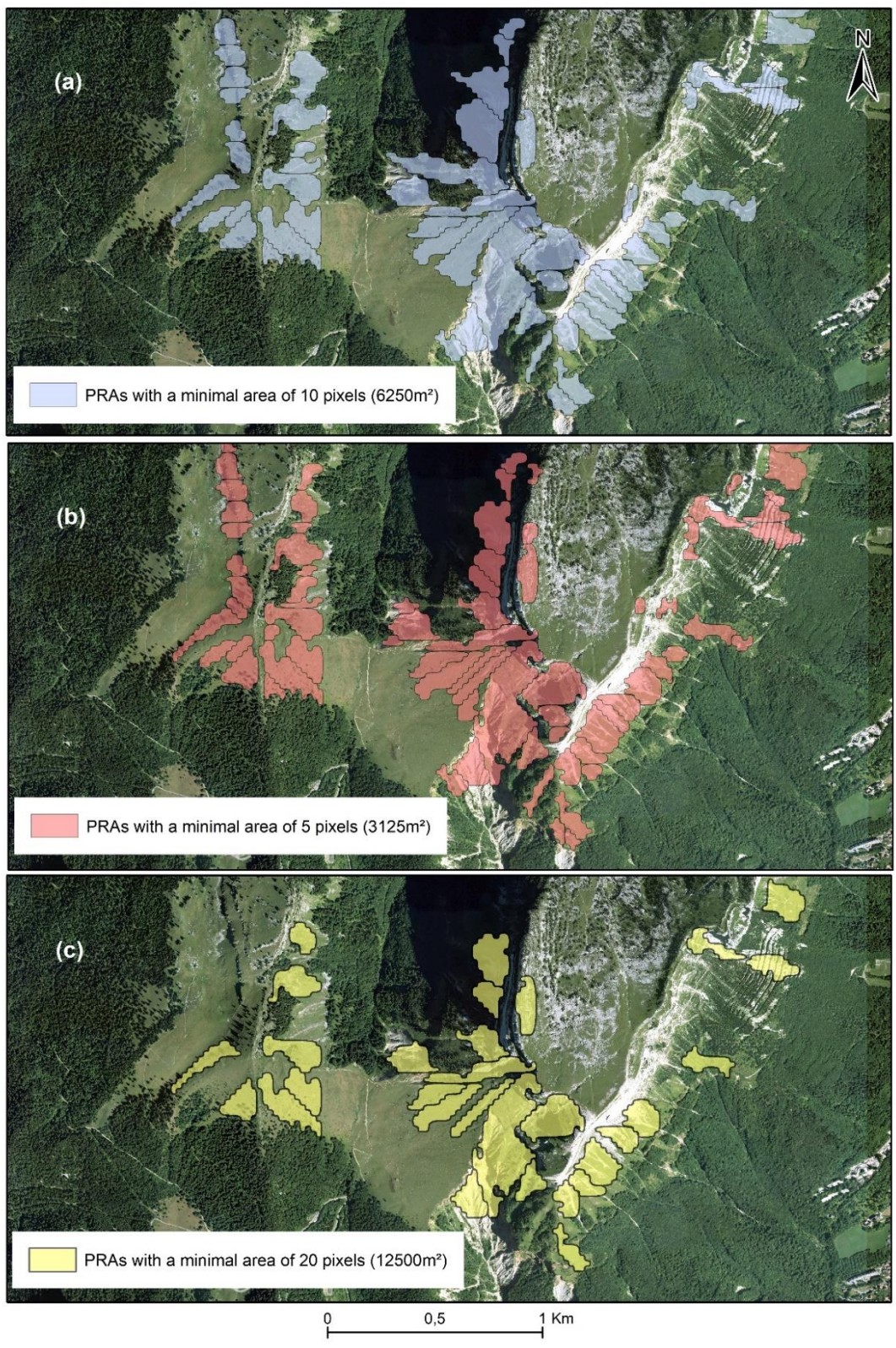

**Figure 10 : Effect on PRA detection of the minimal area size, Chartreuse/Dent de Crolles study area: a) with a 6250 m$^2$ minimal area size, b) with a 3125 m$^2$ minimal area size, c) with a 12500 m$^2$ minimal area size. For the PRA detection, minimal area size varies, all other factors and the DEM resolution are set to their default values (Figure 3) and forest cover data is from DB forest IGN. Aerial photograph ©IGN 2012. For the determination of the validation sample, all factors and the DEM resolution are set to their default values (Figure 4) and forest cover data is from DB forest IGN.**

### 4.2.3 Impact of DEM resolution

The robustness of our results to the DEM resolution was studied over the Mont Blanc massif (Table 7). With finer resolution DEMs, the number of detected PRAs significantly increases (by up to 56.3%), but the overall PRA area decreases (by ~21%). This is attributable to the fact that, with finer resolution DEMs, sharper PRA boundaries arguably more consistent with terrain characterises are evidenced (Figures 11-12). Yet, surprisingly enough, performing the PRA detection with finer resolution DEMs does not improve true positive rates when the default validation sample is considered (i.e. the one determined using Figure 4's scheme with the 25 resolution DEM), leading small 3-6% decreases for PRA numbers and areas. This holds true even if the finer resolution DEM is used also for the determination of the validation sample from CLPA avalanche extents (Table 7).

| | | DEM resolution 25 m / default values | DEM Resolution 10 m | Resolution 10m, adjusted validation sample | DEM Resolution 5m | Resolution 5m, adjusted validation sample |
|---|---|---|---|---|---|---|
| **Validation sample** | **Total area of PRAs within CLPA (validation sample) [km²]** | 58.3 | 58.3 | 53.5 | 58.3 | 45.9 |
| | **Difference in area in validation sample with regards to default values [km2]** | / | / | -4.8 | / | -12.4 |
| | **Difference in area in validation sample with regards to default values (%)** | / | / | -8.3% | / | -21.2% |
| | **Total number of PRAs within CLPA extents (validation sample)** | 1522 | 1522 | 2061 | 1522 | 2181 |
| | **Difference in numbers in validation sample with regards to default values** | / | / | 539 | / | 659 |
| | **Difference in numbers in validation sample with regards to default values (%)** | / | / | 35.4% | / | 43.3% |
| **Detected PRAs** | **Total area of detected PRAs [km²]** | 90.8 | 84 | 88.9 | 71.8 | 71.7 |
| | **Difference in area with regards to default values [km²]** | / | -6.8 | -1.9 | -19.0 | -19.1 |
| | **Difference in area with regards to default values (%)** | / | -7.5% | -2% | -20.9% | -21% |
| | **Total Number of detected PRAs** | 2003 | 2989 | 3081 | 3131 | 3130 |
| | **Difference in numbers with regards to default values** | / | 986 | 1078 | 1128 | 1127 |
| | **Difference in numbers with regards to default values (%)** | / | 49.2% | 53.8% | 56.3% | 56.3% |
| **Evaluation** | **Total area of detected PRAs within CLPA extents [km²]** | 84.9 | 76.8 | 76.6 | 66.8 | 64.1 |
| | **Total number of detected PRAs within CLPA extents** | 1601 | 2420 | 2339 | 2694 | 2457 |
| | **True positive rate (recall), Eq. 3** — **In numbers (%)** | 80 | 75.8 | 76 | 74 | 78.6 |
| | **In areas (%)** | 93.6 | 90.2 | 86.2 | 89.8 | 89.4 |
| | **Difference in recall with regards to default values** — **In numbers (%)** | / | -4.2 | -4 | -6 | -1.4 |
| | **In areas (%)** | / | -3.4 | -7.4 | -3.8 | -4.2 |

**Table 7: Effect of DEM resolution on PRA detection, Mont Blanc Massif. With both the 10 and 5m resolution DEMs, two cases are considered: either the default validation sample determined with the 25m resolution DEM is kept, or a new validation sample is determined following Figure 4. For both the PRAS detection and determination of the validation sample, all other factors are set to their default values (Figures 3-4) and forest cover data is from DB forest IGN.**


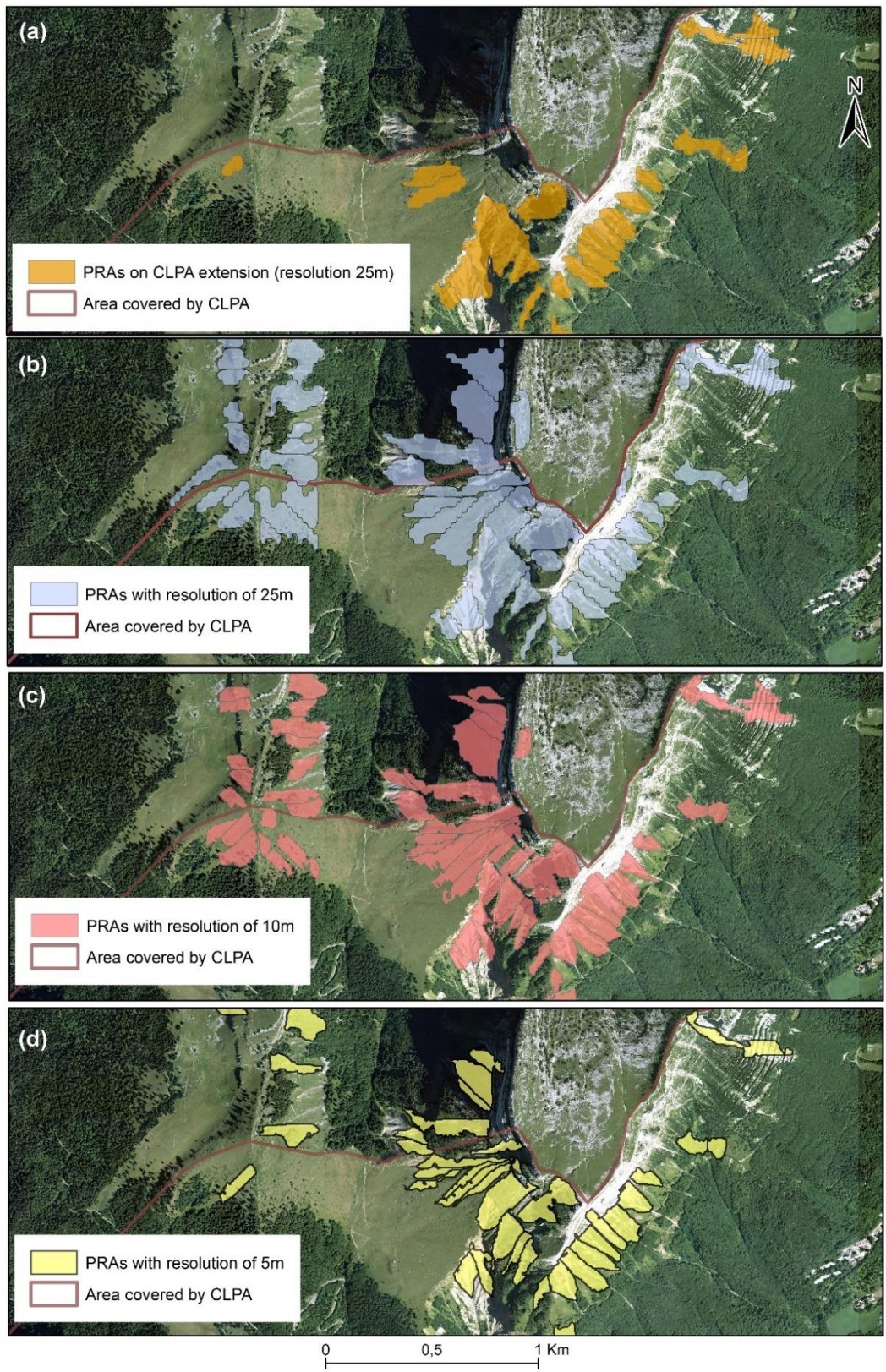


**Figure 11: Effect of DEM resolution on PRA detection, Chartreuse/Dent de Crolles study area. Aerial photograph ©IGN 2012. a) For the determination of the validation sample, all factors and the DEM resolution are set to their default values (Figure 4) and forest cover data is from DB forest IGN; the absence of CLPA in the upper left corner is clearly visible. b-d) For the PRA detection, DEM resolution varies, other factors are set to their default values (Figure 3) and**

**forest cover data is from DB forest IGN. In a), CLPA extensions" refer to avalanche extents from the French avalanche cadastre.**

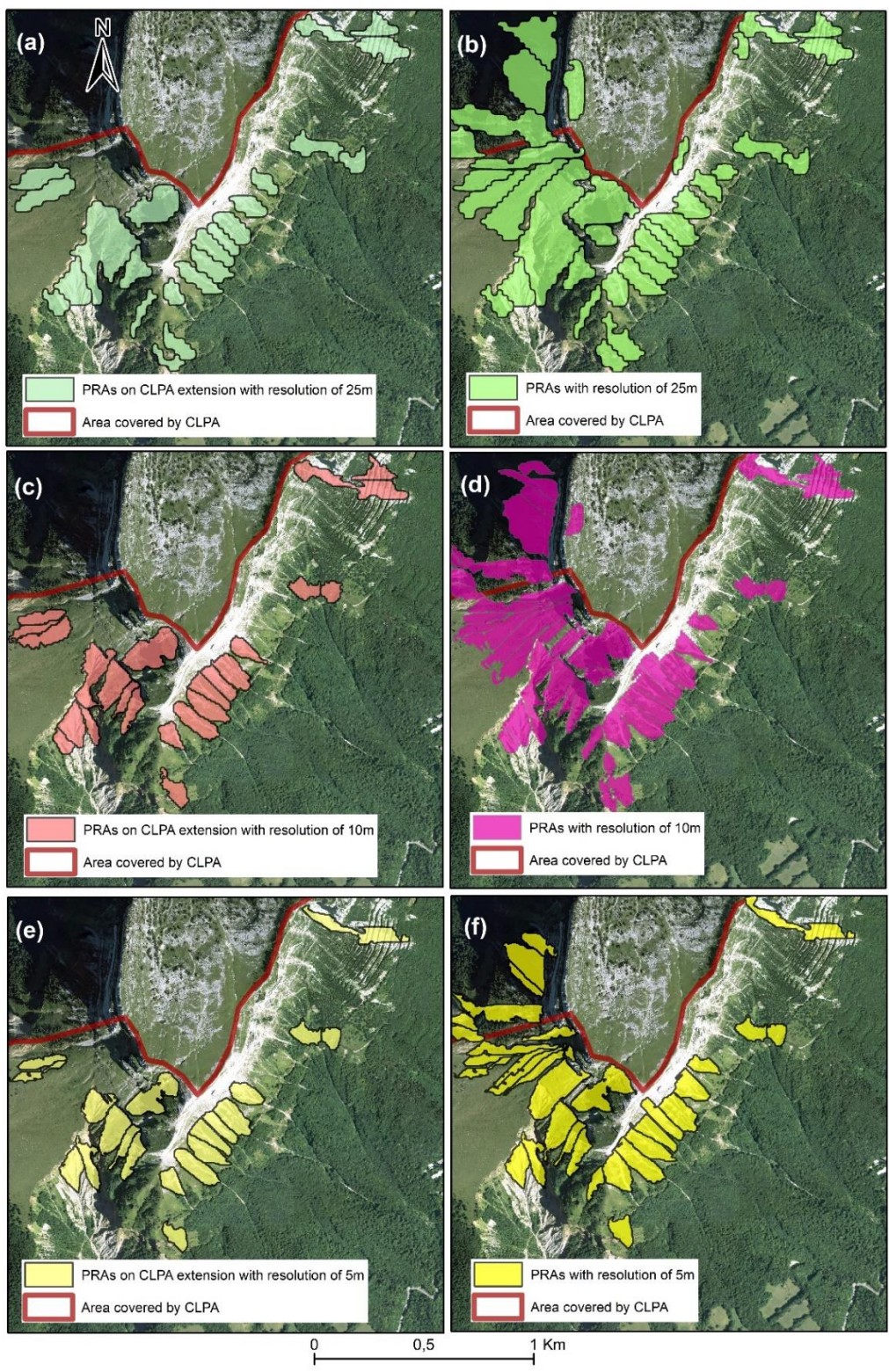

**Figure 12 : Combined effect of DEM resolution on the selection of the validation sample (left) and PRA detection (right), Chartreuse/Dent de Crolles study area. For the PRA detection and the determination of the validation sample, DEM resolution varies, all other factors are set to their default values (Figures 3-4), and forest cover data is from DB forest IGN. Aerial photograph ©IGN 2012. Left, the absence of CLPA in the upper left corner is clearly visible, and CLPA extensions" refer to avalanche extents from the French avalanche cadastre.**


**5 Discussion**

**5.1 Main outcomes of the work**

This study proposes a procedure for the automated detection of PRAs without any consideration of release frequency and applies it to three entire massifs of the French Alps, Chartreuse, Maurienne, and Mont-Blanc, demonstrating its applicability to large areas (Table 1). The approach classically uses topographical filters as follows: a minimal elevation of 1400 m, slopes between 28° and 60°, a distance to the closest ridge limited to 600

m, and a minimal area of 6250m² for the resulting PRAs. It also excludes forested areas, and uses a watershed delineation algorithm, which provides spatial entities comparable to individual avalanche paths. Retained DEM resolution is 25m and forest cover data is the one of DB forest from IGN.

Massif-scale true positive rates obtained ranged between 80-87% in PRA numbers, and between 92.4% and 94% in PRA areas (Table 4). Also the parametric study highlighted a rather high robustness of these scores to many

choices, e.g. moderate decreases in true positive rates when the best values were replaced by other realistic values. As the chosen three massifs rather well represent the diversity of altitudinal and lithological contexts of the French Alps, these results suggest that the developed method could be systematically used for the entire French Alps, a large territory where no automated PRA detection has been systematically implemented so far. This will allow in the future the detection of PRAs within the 23 massifs even in areas which are still not covered by CLPA. Yet, we

want to remember that we evaluated "true positives" only. Also we acknowledge that our evaluation approach, and notably the definition of the validation sample, favours the comparison with our detected PRAs (Sect. 5.3). As a consequence, our evaluation scores should not be directly compared with scores obtains with other approaches on other data sets, but only taken as an indication that our approach performs rather well in the French context, which was further confirmed by visual inspections in small areas.

**5.2 Selected factors and data sets and comparison to existing methods**

Default settings of the developed PRA detection method were determined/confirmed on the basis of an evaluation approach (see below), and a large parametric study (Tables 5-7 and Tables S1-S2 of the SM). Note that what we did is neither an extensive grid search testing all possible combinations, which would have been extremely numerically consuming over entire massifs, nor a proper sensitivity analysis in the strict sense of the term, which

would have required more sophisticated techniques introduced in the snow avalanche field only recently, and so far not for PRA detection (e.g. Heredia et al., RESS 2021; 2022). Yet, a large effort was conducted over large fractions of entire massifs, combining different complementary analyses: removing each filter one by one, replacing one data source by another, etc. Besides, raw massif-scale evaluation scores were supplemented by detailed visual investigations in smaller study areas (Figures 9-12 and Figures S4-S5 of the SM). Obtained results

allowed quantifying the robustness of the PRA detection method and, to a certain extent, identify optimal choices (or, at least, sets of suitable values / data), based on changes in evaluation scores with competing solutions. Most of our choices are largely in accordance with those of several past studies (e.g., Maggioni et al, 2002), except the retained resolution of the DEM that we found less important to reach high efficiency in PRA detection than in other studies (Bühler et al., 2013).

In details, a minimal elevation of 1400 m for snow avalanche release seems appropriate in the French Alps considering the limited amounts of snow precipitation expected in current and future conditions below this elevation. Yet, this threshold is dependent to the local climate and should be adapted for other regions. Our other choices regarding slope and distance to ridge are i) largely in accordance with the state of the art (Maggioni and Gruber, 2003; Bühler et al., 2013), and ii) compatible with broader knowledge regarding snow avalanche formation

(Schweizer et al., 2004). Regarding the DEM, a 25 m resolution appeared as a good compromise between high efficiency in the detection and the computational burden, even if, arguably, it leads less realistic PRA boundaries than finer DEM resolutions.

    Even if its effect was found to vary from one massif to another, our watershed delineation step, in essence rather similar to the OBIA approach of Bühler et al. (2018), was found to greatly improve true positive rates, especially

in terms of PRAs numbers. As PRA detection is mostly oriented towards avalanche simulations to evaluate hazard and risk downslope, this gain is crucial, and much more important that the slight loss in true positive rates in PRA area obtained for one of the three studied massifs. Indeed, the segmentation avoids unrealistically wide PRAs that correspond to different avalanche paths/events, which may allow performing simulations corresponding to realistic individual avalanche events.

Our parametric study showed higher true positive rates both in numbers and areas when forested terrain was excluded than when this filter was not considered. As forest cover is known to be effective to prevent avalanche release, this result is logical. We tested the three main forest data sources available at the scale of the entire French Alps. None of them is perfect, and information that is more accurate may be available in other areas/countries. However, visual analyses in different areas showed that at least the DB forest we retained is acceptable (Figure 2).

In addition, the systematic analysis performed over the three entire considered massifs showed that it leads the highest true positive rates both in numbers and areas with regard to the other available forest data sources. We yet acknowledge than even better forest data could lead to results that are even more reliable. Also, a less stringent PRA / No PRA rule as function of forest presence/absence (e.g. a higher release susceptibility with decreasing forest density) would be an interesting option for further developments.

Finally, even if we are able to compare the steps and choices involved in our approach with those involved in other PRA detection methods existing in the literature, we did not a direct comparison to existing algorithms. For this we would need i) to run all existing algorithms which for now are not available to the community as open access codes, and ii) to test them on different datasets with different characteristics (resolution, climatic and topographic context, etc.). i) is because the description provided in the scientific literature is not always sufficient to recode

the algorithms, and, even when it is, following published guidelines/equations does not guarantee that what has been done can be reproduced exactly, due to, e.g., differences introduced by different numerical implementation schemes. ii) is because different algorithms may perform better in different contexts, so that, possibly, there is no universal "best algorithm" for PRA detection.

### 5.3 Evaluation of detected PRAs based on the CLPA / avalanche cadastres

In order to evaluate the proposed method, we compared detected PRAs to the CLPA after isolating individual release areas within CLPA extents. However, evaluating a PRA detection method and even determining a suitable validation sample is by essence difficult, which contributes to explain why i) such validation data sets are rare and generally small, and ii) even proper evaluation exercises are seldom in the literature (e.g., Bühler et al., 2018).

What follows discusses the pro's and con's of the approach we chose and provides insights from the work done
that may help designing more efficient evaluation techniques for PRA detection methods.

Despite drawbacks inherent to any avalanche cadastre (uncertainty regarding some extents, or even missing avalanche extents), the CLPA is a remarkable source of information regarding past avalanches (Bourova et al., 2016) due to its old history, its rigorous protocol and because it includes various complementary sources of information (testimonies, landscape footprints, etc.). In addition, its principle makes it suitable to evaluate a method that aims at automatically identify the maximal avalanche prone terrain. Notably, as CLPA extents are concatenations of all observed avalanche extents on a given avalanche path, CLPA is more likely to provide an accurate estimate of the entire "ground truth" than any observation of single avalanche events. Yet, identifying a validation sample for a PRA detection method from CLPA extents is not that easy. First, within a given massif, some areas are covered by CLPA and some of them are not (Table 1, Figures 11-12), and the fraction of the territory covered by CLPA varies from one massif to another. As a consequence, evaluation scores can be computed only over areas which are covered by CLPA, namely much larger areas in Mont Blanc and Maurienne Massifs than in the Chartreuse massif (Table 1). Second, in areas covered by CLPA, it is known that CLPA is very complete and precise on lower slopes and in forested terrain and more likely to i) miss avalanche prone terrain (and release areas) and ii) map boundary of avalanche extents less precisely far from inhabitants and forests, notably in high elevation areas. In such latter cases, "false positives" in our evaluation scheme may be true PRAs that are simply missing in the CLPA. This may explain, e.g., why true positive rates slightly higher in numbers where obtained in Chartreuse than in the two other studied massifs (Table 4). Lastly, CLPA does not distinguish release areas from flow paths and runout zones, which implies that a pre-processing is required to isolate individual release areas within CLPA extents that can be compared with our detected PRAs. To this aim, we used the same filters than for the PRA detection. This makes that the identified PRAs and our validation data are no longer independent.

The way we define our validation sample may therefore be criticized. However, obtaining a fully independent exhaustive sample of "ground truth" PRA may simply be impossible. A first reason is that even "live", what can be observed is the full extent of an avalanche, and separating its release area from the flow path may be very difficult, so that the delimitation of any release area may always involve some uncertainty. Second, assuming that one is able to precisely map a release area, there is little chance that it corresponds to the entire PRA for the considered path. Third, even in the best avalanche cadastre, some avalanche extents corresponding to the rarest events that can be released under very specific conditions only may be missing. As a consequence:

- even with the best validation sample at hand, one will never be sure that i) all potential PRAs have been identified, ii) the maximal potential extent that can be released under the most extreme conditions has been identified for all paths. This is why only "true positives" PRAs can be trustfully validated, as it is never sure that a complete PRA or a part of a PRA automatically identified but not present in the validation sample is not simply erroneously missing in the validation sample;

- the definition of any PRA validation sample will always involve some partially subjective decisions. Our choice was to use filters explicitly in a transparent manner. This has the drawback of allowing the evaluation of true positives only. We already argued that i) focusing on true positives in PRA evaluation is sensible, ii) the raw values of our scores should be considered with care. However, it should be noted additionally that, even if focusing the PRA search and evaluation on terrain which is presumably favourable to avalanches may

indeed favour high scores (and notably increase the number of false positives), it is an interesting option to maximise the efficiency of the detection. It is indeed now a rather standard approach in machine learning (e.g., Giffard Roisin et al., 2020) that is increasingly used in susceptibility mapping approaches outside the snow avalanche field in order to focus the detection on most suitable areas and thus increase the detection power.

Finally, confusion matrices and performance criteria were seldom used so far to evaluate PRA detection methods (to our knowledge, only in Bühler et al, 2018), and with one single metric, which may not be enough to fully judge the efficiency of a PRA detection method, as, e.g. the right number of PRAs can be identified but with wrong extents, and vice-versa. As a first step towards improved evaluation schemes for PRA detection methods, we proposed to evaluate efficiency with true positive rates (recall) computed both for PRA numbers and areas, which may cover the two most critical dimensions of the problem. Yet, in the future, additional metrics should probably be considered, notably metrics that combine both information (e.g., with "success" for different thresholds defined as minimal matching areas), and/or metrics related to various characteristics of the detected PRAs (shape, elevation, etc.). This may help assessing even more precisely the strengths and weaknesses of our (or another) PRA detection method. In addition, we performed the evaluation of our detection method over unusually large areas covering significant proportions of three entire massifs with diverse characteristics. A similar approach could be further used for comparing several PRA detection methods (Sect. 6) and/or in other mountain environments with validation data having strengths and weaknesses different from those of the CLPA. This would help understanding to which extent the high scores we obtain i) are strongly influenced by our evaluation framework, ii) can be reproduced in other contexts, iii) how the most critical parameter need to be changed to fit various conditions (e.g. lower minimal elevation in colder climate).

## 6 Conclusion and outlooks

To map and mitigate avalanche risk at large scales, the automated detection of PRAs is a powerful solution, notably to provide inputs for avalanche simulation softwares (e.g., Gruber and Bartelt, 2007). In this study, a PRA detection method adapted to the characteristics of the French Alps was developed and tuned on the basis of the CLPA, a valuable large-scale data source regarding past avalanches. In addition to the potential benefit for better understanding and assessing avalanche hazard in the French Alps and in mountain environments with similar characteristics (see below), outcomes of the work include i) the determination of individual PRAs using a watershed delineation algorithm, ii) an approach to define a validation sample from a cadaster of avalanche extents, iii) an evaluation procedure based on two metrics, PRA numbers and area, and iv) a better definition of evaluation scores that should be interpreted in the context of PRA identification. These methodological developments should help progressing towards more efficient approaches for PRA detection and evaluation.

Future works may expand the effort to the 23 massifs of the French Alps and provide a statistical assessment of the detected PRAs. The latter could also be, combined with available extreme snowfall/snow depth estimates (Gaume et al., 2013; Le Roux et al., 2021), for a large-scale mapping of avalanche hazard and risk in the French Alps using simulation tools, as already done in other countries (e.g. Bühler et al., 2018; 2022). To this aim, the most direct use of our PRAs would be to consider all of them and, for each of them, their entire area, as well as "maximal" snow depths and "minimal" friction parameters for avalanche simulation. This may lead to the delineation of the entire avalanche prone terrain. To go further and evaluate hazard and risk levels downslope, additional assumptions/choices would be required regarding the magnitude and frequency of potentially triggered

avalanches. This may include several scenarios and/or the entire probability distribution for i) the frequency of trigger in each PRA, ii) the fraction of each PRA which is released, iii) the snow depth, iv) friction parameters, etc. Ultimately, results could be compared to existing hazard assessment based on interpolation of existing avalanche runout data (Lavigne et al., 2015).

A second potential outlook would be to apply the method as it is in other close territories such as the Pyrenees (Oller et al., 2021) or the Vosges Mountains (Giacona et al., 2017). Yet, this would imply further tuning of some parameters of the method such as minimal elevation and distance to ridge, to adapt these to local peculiarities. More ambitious outlooks relate to improvements of the PRA detection method: i) make it dynamic to assess PRAs conditional to snow and weather conditions (e.g., Chueca Cía et al., 2014), ii) switch from deterministic to probabilistic detection rules (e.g., Kumar et al., 2019) to, e.g., include in it the uncertainty about the best parameter values to be used, and/or iii) use CLPA extents not only as an evaluation support but as a training sample. Finally, a comparison of PRA detection methods would certainly be beneficial for the community and should be envisioned in the future. To this aim, we hope that our data, now freely available (Data availability Sect.), will foster further benchmarks between already existing or new PRA detection algorithms in order to draw firmer conclusions regarding the respective efficiency of the different proposals in different contexts.

## 7 Acknowledgements

CD holds a PhD grant from Grenoble-Alpes University. Support from the French National Research Agency through the Statistical modelling for the assessment and mitigation of mountain risks in a changing environment - SMARTEN program (ANR-20-Tremplin-ERC8-0001) is acknowledged. The authors are grateful to Margherita Maggioni, an anonymous referee and to Yves Bühler whose insightful comments helped producing a better paper. IGE/INRAE is member of Labex OSUG.

## 8 Author's contribution

NE and GE designed the research. CD performed the analyses and drafted the first version of the manuscript. CD and MD produced the illustrations. All authors discussed the results and edited the manuscript.

## 9 Data availability

The source data and results of this study are available as: Duvillier, Cecile, Eckert, Nicolas, Evin, Guillaume, & Deschâtres, Michael (2022). Development and evaluation of a method to identify potential release areas of snow avalanches based on watershed delineation - source data (Version_1) [Data set]. Zenodo, https://doi.org/10.5281/zenodo.6517730. This data can be used to reproduce all the results of the paper and for further benchmarking of snow avalanche potential release area detection methods.

## 10 Code availably

The proposed PRA detection method has been developed using the QGIS free software and environment. It combines existing algorithms from the free softwares QGIS, SAGA, GDAL and GRASS, and from ArcGIS (for watershed delineations). Scripts can be requested to CD.

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
