# Peer review of "Development and evaluation of a method to identify potential release areas of snow avalanches based on watershed delineation"

_Natural Hazards and Earth System Sciences, 2022_

## Referee Comment (RC1)

Review of the paper **Development and validation using ground truth of a method to identify potential release areas of snow avalanches based on watershed delineation** by Cécile Duvillier[1], Nicolas Eckert, Guillaume Evin and Michael Deschâtres

**General comments:**

The paper addresses an issue which is well-known in avalanche science. Potential avalanche release areas (PRAs) are one of the most important parameters to be identified for avalanche hazard assessment (for ex. hazard maps, design of defense measures, etc.).

The Introduction is very rich and refers to an exhaustive literature about the PRA definition topic. Good point! And it explains clearly which are the strong and weak points of existing methods, in particular about the validation method. At the end of the Introduction, then, it clearly states the aim of the paper, which is, beside the PRA definition method, also the development of a test and validation method (see my comments in the following).

Concerning the PRA delineation method: I think it is a good method which takes what already exists and adds a GIS watershed routine to define single PRAs. This part is more a GIS technical issue than an avalanche science issue… but it seems that the authors found a good solution to a challenging delineation problem.

Concerning the PRA validation method, I think that confusion matrix and evaluation scores are a good proposal… but the weak point is the pre-processing of the CLPA data. Finally, the authors apply the same procedure developed for the PRA identification to the CLPA database to extract the release areas from the polygons of the entire avalanche extensions… Therefore, they use the same method to produce the PRAs and, in a way, to extract a dataset to validate it. I mean, *we do what we can with what we have*, but then I would not stress so much the importance of the validation method - it is even in the title and using the word "ground truth". For me "ground truth" used in relation to potential release areas should mean observed release areas… Actually, this is the common problem of all the validation methods for PRA delineation, as not many release area datasets exist. For example, also Harvey et al. (2018) made something similar in the analysis of release areas of ski triggered avalanches. In the discussion section, this problem is actually well considered and truly presented. Though, I would not use "ground-truth"…

Stephan Harvey, Gunter Schmudlach, Yves Buhler, Lukas Durr, Andreas Stoffel, Marc Christen, AVALANCHE TERRAIN MAPS FOR BACKCOUNTRY SKIING IN SWITZERLAND, Proceedings, International Snow Science Workshop, Innsbruck, Austria, 2018 and presentation for the CSAW 2020 ( https://www.oegsl.at/automatic-high-resolution-mapping-and-classification-of-avalanche-terrain-regarding-potential-release-triggering-and-run-out-zones/?lang=en time 3:16 - 3:22)

The Results and Discussion sections might be shortened… but, actually, the reader can easily follow the *file rouge* of the whole story, therefore for me it is ok like they are (but see later my comment to lines 447-463).

**Specific comments:**

Title: maybe it could be made simpler as follows: "Development and validation of a method to identify potential release areas of snow avalanches based on watershed delineation". This title doesn't stress too much the validation method, which is not really based on ground truth data, and instead stresses the watershed delineation which is something new in the PRA definition method. Or, even, the author could give a geographical information: "Development of a method to identify potential release areas of snow avalanches based on watershed delineation and validation in the French Alps".

From the abstract (very clear!) It seems that the authors develop a validation method which might be applicable also for existing PRA methods. This gives expectations to the readers…

About the Introduction: I think it should finish with the description of the aim of the paper, without the rest, which seems a bit like an abstract… giving already information on which has been found after the analyses (but this section is only the Introduction!). For example, I would move lines 109-116 ("Following … individual PRAs.") to the Results or Discussion sections. And I think it is not necessary to explain how the paper is structured… It comes naturally while reading. Eventually the authors can move the sentences at the beginning of the corresponding sections.

Line 28: ok for the reference to Amman and Bebi (2000) which is a general overview paper, while I would not refer to Braun et al (2020) which is a very specific one. Better, maybe, to refer to another general paper, or even a book (for ex. McClung and Shaerer, The avalanche handbook, 1993).

Line 28-29: I would cancel this sentence "No countermeasure… one minute." and "therefore" at line 31. I understood the message but I think it is not necessary here.

It is very good how the authors state the reasons behind the choice of the different inputs (for ex. lines 143-169… and also make a kind of sensitivity analysis about this (Sect. 4.2).

Line 181: I do not understand the reference to Figure 3… the sentence tells about areas without CLPA but the main outputs in Figure 3 are the resulting PRAs… Moreover, here we are still in Section 2. Data, therefore results should not be presented yet. I would cancel Figure 3. Instead, the authors might put a figure showing the areas covered by CLPA within the three test areas (Mont-Blanc, Chartreuse, Maurienne).

Line 232: Here it is not clear if the identification of individual watersheds is made automatically.

Line 259: I would specify that it is the **planar** area; also at line 265.

Line 316: I would move here Figure 3, which in fact shows the results of the PRA definition for the area of Chamonix. I think it is not necessary to highlight the pink area as "CLPA extension outside PRAs/AUTO"; "CLPA extensions" is enough. The figure would result less messed up (see Figure 7, S2 and S3, which are clearer).

Line 336: I would try to find a way to add a box (or a second figure) with a zoom on the area where CLPA exists (region in the lower-right corner) in order to better show the difference between matching (blue) and not matching (light blue) PRAs. Ok to put the other two figures (Mont Blanc and Maurienne) in the supplementary materials.

Line 447-463… these lines are a repetition of the Introduction… here it is time for discussion! :o)

Line 520-21… I do not understand the sentence… a verb is missing?

**Technical suggestions:**

Figures:

Figure 1. I would use a transparency for the violet and light blue colors to show the different study areas, so that the topography from the shaded DEM can be appreciated. **This is actually a general comment valid also for the other figures**.

Figure 5: point (3) should be in *italic* and I would substitute parts with **areas**. In the second blue rectangle I would simply write **forest** instead of forest parts.

Line 43: "Wider benefits can also arise **FROM** the systematic …"

Line 327: **PRAs** instead of PRAS

Table 2: At the end I would add a reference to Table 1 as a legend for the Confusion matrix.

Concerning the English: I am not the best person to judge the quality of the English… I would probably let the paper be revised by an English native speaker.

---

## Author Comment (AC2)

**REVIEW**

This paper proposes a new method for identifying potential release areas (PRA) for snow avalanches based on terrain characteristics and validates the approach using a long-term avalanche cadaster dataset. The research is situated in France. and the study area includes three mountain massifs in the French Alps. Overall. the topic is interesting and relevant for the avalanche research community and the NHESS readership.

While I appreciate the authors' desire to create a practical. transparent. and computationally efficient algorithm for PRA identification that uses easily accessible datasets. there are several substantial weaknesses in the present study that. in my opinion. prevent this manuscript from being a meaningful contribution to the literature in its current form. Properly addressing several of my concerns would require a substantial redesign and/or expansion of the study. and I am unsure whether that can be accomplished within the current peer-review process. I hope that the following comments can help the authors to further develop their research.

**Author's response:** We deeply thank the referee for his/her meaningful suggestions and feedback. Even if some of the criticisms where for us a bit too strong, we took them as a challenge to clarify and improve our research and they certainly greatly helped us to improve the paper.
Let us just stress first that the main objective of the paper is to develop a PRA detection method that grounds on already existing developments (notably by Margherita Maggioni, Yves Bühler and their co-authors) and works reasonably well in the context of the French Alps. This is shown by performing an evaluation/validation exercise relying on an excellent data source regarding past avalanches, the CLPA in different massifs and areas of the French Alps (response Tables 1 and 2). However, in addition to the benefit for avalanche hazard assessment in the French context, there are also some slight methodological outcomes of the paper that may be of broad relevance for the topic:
- i)   The determination of individual PRAs using a watershed delineation algorithm;
- ii)  A validation approach on the basis of accuracy scores computed using two metrics, PRA numbers and area;
- iii) Broader findings and reflexions about how to validate a PRA detection method, notably how can a validation data sample be defined, and which scores can be interpreted.

None of these points are completely new in the community, but we find that they have not been fully answered so far in the literature, and we humbly hope that our paper will therefore bring some useful elements to the debate. The questions raised by both reviews however indicate that these objectives/questions were not clear enough in the first version of the paper. The paper will therefore be largely reworked to better introduce the research questions and discusses the findings and the approach with regards to these questions.
Eventually, let us note here that to make the validation and parametric study more convincing, we performed many additional analyses i) in terms of potential set of parameters, ii) by considering into the analysis the questions of the DEM resolution, iii) by performing the parametric study all over 3 entire massifs and not only over a small area, and iv) by investigating the relation between the determination of the validation sample and the accuracy scores. An additional small area within the massif of Chartreuse (Chartreuse study area / Dent de Crolles) is also considered to better highlight/illustrate some results (response Figure 1). However the matching between detected PRAs and validation sample should be checked first in terms of massif-scale scores, which provides a much more systematic assessment free of local effects and "cherry picking".
What follows provide a point-by-point answer to the referee's comments, questions and suggestions and introduces the results of these additional analyses that will be fully integrated in the reworked version of the paper.

N. Eckert. on belief of the co-authors

|  |  | Chamonix area | Chartreuse area (Dent de Crolles) | Chartreuse Massif | Mont-Blanc Massif | Maurienne Massif |
|---|---|---|---|---|---|---|
| Accuracy rate (Eq. 3) | In numbers | 92.1 | 93.6 | 93.5 | 90 | 91.4 |
|  | In areas | 98.3 | 90.2 | 96.2 | 96.8 | 97 |

*Response Table 1: Summary of accuracy scores for the different massifs and study areas (updates Table 3), including the new Chartreuse/Dent de Crolles study area (response Figure 1).*

[Figure]

*Response Figure 1: updated figure of the paper that includes the new Chartreuse/Dent de Crolles study area.*

| | Chamonix area | Chartreuse area / Dent de Crolles | Chartreuse massif | Mont Blanc massif | Maurienne massif |
|---|---|---|---|---|---|
| Total area [km2] | 34.3 | 7.6 | 847.0 | 578.0 | 917.1 |
| Total area covered by CLPA [km2] | 25.8 | 4.7 | 44.0 | 354.6 | 382.3 |
| Fraction of area covered by CLPA (%) | 75.3% | 61.7% | 5.2% | 61.3% | 41.7% |
| Total area of PRAs within CLPA extensions (validation sample) [km2] | 3.6 | 0.5 | 1.6 | 58.3 | 55.7 |
| Total number of PRAS withn CLPA extensions (validation sample) | 85 | 28 | 85 | 1522 | 1884 |
| Total area of detected PRAs [km2] | 8.1 | 1.2 | 15.4 | 166.3 | 115.1 |
| Total Number of detected PRAs | 210 | 58 | 721 | 3676 | 3638 |
| Total area of detected PRAs within area covered by CLPA [km2] | 5.5 | 0.8 | 2.3 | 90.8 | 71.6 |
| Total Number of detected PRAs within areas covered by CLPA | 107 | 39 | 108 | 2003 | 2575 |
| Aerial fraction of detected PRAs within the area | 23.7% | 15.3% | 1.8% | 28.8% | 12.5% |
| Aerial fraction of PRAs within the area covered by CLPA | 21.2% | 17.0% | 5.3% | 25.6% | 18.7% |
| Aerial fraction of PRAs within CLPA extensions with regards to total area of PRAs | 67.2% | 68.4% | 15.3% | 54.6% | 62.3% |
| Fraction of PRAs numbers within CLPA extension with regards to total number of PRAs | 40.5% | 48.3% | 11.8% | 41.4% | 51.8% |

**Response Table 2**: *Characteristics of studied massifs and areas.*

**PRIMARY ISSUES**

**Selection of terrain characteristics and threshold for PRA identification**

While the selection of terrain characteristics included in the PRA identification algorithm is based on existing literature. the reasons for their selection (or the refusal of other characteristics) are only discussed superficially. Furthermore. the selections of the parameter thresholds (e.g.. 1400 m elevation threshold. incline range) do not seem to be well grounded in evidence. I recommend that the authors conduct a proper grid search to determine the ideal parameter settings for their PRA identification algorithm. This is particularly important because they use a low-resolution DEM (25 m). which results in incline values that are biased towards lower values. This means that the thresholds described in the literature are not necessarily applicable. While the current sensitivity analysis might intend to do this. it is not done in a very rigorous and scientifically valid way. See additional comment on sensitivity analysis below.

**Author's response:** As indicated before, we conducted many additional analyses to investigate how the choice of the different parameter values and thresholds affects the PRA detection in terms of PRA areas, numbers and accuracy scores (response Table 3). This systematic search was performed over the entire massif of Mont Blanc. Results overall appear as consistent, with globally decreasing accuracy rates as one leaves the default values determined from the literature and used for the identification

of the validation sample. Overall, accuracy scores seem nevertheless rather stable over considered ranges of parameters/thresholds, with slope range being the most influential parameter over the tested range  (up to a 10% decrease in accuracy for numbers). Also, specific areas have been analysed (response Figures 2-4). They, e.g., show that a too large minimal extension, a too high minimal elevation or a too restricted slope range logically misses release areas that an expert analysis would definitely consider as suitable location for an avalanche release. These results will be more deeply analysed in the revised version of the paper. See also our responses to the next comments, notably those related to data, validation and DEM resolution and related discussion regarding the dependency of the results on the validation set-up and the interpretation of the scores.

| | With default values | Minimal area (m2) | | | Minimal elevation (m) | | | Slope range (°) | | | | Maximal distance to ridge (m) | | | |
|---|---|---|---|---|---|---|---|---|---|---|---|---|---|---|---|
| | | 3125 | 9375 | 12500 | 1200 | 1600 | 1800 | [26-60] | [30-60] | [32-60] | [34-60] | 400 | 500 | 700 | 800 |
| Total area of detected PRAs [km2] | 90.80 | 93.8 | 88.2 | 85.2 | 90.7 | 88.9 | 88.7 | 90.8 | 64.5 | 58.9 | 51.8 | 81.0 | 88.5 | 94.3 | 95.0 |
| Delta area with regards to default values [km2] | / | 2.99 | -2.57 | -5.64 | -0.05 | -1.89 | -2.14 | 0.00 | -26.25 | -31.89 | -39.03 | -9.80 | -2.33 | 3.52 | 4.20 |
| Delta area with regards to default values (%) | / | 3.3% | -2.8% | -6.2% | -0.1% | -2.1% | -2.4% | 0.0% | -28.9% | -35.1% | -43.0% | -10.8% | -2.6% | 3.9% | 4.6% |
| Total number of detected PRAs | 2003 | 2632 | 1654 | 1369 | 2000 | 1979 | 1941 | 2002 | 1598 | 1582 | 1505 | 1877 | 2008 | 2088 | 2104 |
| Delta numbers with regards to default values | / | 629 | -349 | -634 | -3 | -24 | -62 | -1 | -405 | -421 | -498 | -126 | 5 | 85 | 101 |
| Delta numbers with regards to default values (%) | / | 31.4% | -17.4% | -31.7% | -0.1% | -1.2% | -3.1% | 0.0% | -20.2% | -21.0% | -24.9% | -6.3% | 0.2% | 4.2% | 5.0% |
| Total area of  detected PRAs within CLPA extensions [km2] | 84.9 | 85.7 | 83.3 | 81.2 | 84.7 | 83.1 | 76.6 | 84.8 | 61.9 | 56.3 | 49.6 | 75.3 | 82.3 | 87.5 | 87.5 |
| Total number of detected PRAs within CLPA extensions | 1601 | 1768 | 1391 | 1201 | 1590 | 1589 | 1520 | 1597 | 1409 | 1406 | 1349 | 1468 | 1576 | 1622 | 1621 |
| Accuracy rates — In numbers | 90 | 83.6 | 89.9 | 89.3 | 89.6 | 89.3 | 84.4 | 89.9 | 86.9 | 84.2 | 81.0 | 85.2 | 88.1 | 83.6 | 88.5 |
| Accuracy rates — In areas | 96.8 | 95.7 | 96.8 | 96.7 | 96.7 | 96.5 | 91.3 | 96.7 | 96.0 | 94.7 | 92.6 | 95.0 | 96.2 | 92.6 | 96.1 |
| Delta accuracy with regards to default values — In numbers | / | -6.4 | -0.1 | -0.7 | -0.4 | -0.7 | -5.6 | -0.1 | -3.1 | -5.8 | -9.0 | -4.8 | -1.9 | -6.4 | -1.5 |
| Delta accuracy with regards to default values — In areas | / | -1.1 | 0.0 | -0.1 | -0.1 | -0.3 | -5.5 | -0.1 | -0.8 | -2.1 | -4.2 | -1.8 | -0.6 | -4.2 | -0.7 |

*Response Table 3: Parametric study performed all over the Mont Blanc Massif. Total area of detected PRAs and total number of detected PRAs are those of the part of the massif covered by CLPA. The table expands information that was provided previously in Table 5 and 7 for the small Chamonix area only.*

[Figure]

*Response Figure 2*: *Effect on PRA detection of the minimal area. Chartreuse/Dent de Crolles study area.*

[Figure]

*Response Figure 3*: Effect on PRA detection of the minimal elevation. Chartreuse/Dent de Crolles study area.

[Figure]

*Response Figure 4*: *Effect on PRA detection of the slope range. Chartreuse/Dent de Crolles study area.*

**Selection of datasets**

Several datasets used in this study seem to be of lower data quality than established best practices in the field of PRA identification suggest. For example. the forest data set seems to have considerable limitations and the DEM is of much lower resolution than suggested in the literature. While I do not have a problem with a let's-do-the-best-we-can-with-what-we-have approach (not everybody has Swiss quality datasets available!). these choices need to be clearly explained and potential shortcomings evaluated and discussed.

**Author's response:** Regarding forest cover information, there are three main data sources available at the scale of the entire French Alps and we tested all three. None of them is perfect, and, certainly, Switzerland and other countries benefit from more precise systematic forest inventories. Detailed comparison with aerial photographs shows that, logically, main difficulty arises when the forest density is low, which makes the limit between forest and non-forest difficult to set (Response Figure 5). However, visual analyses in different configurations convinced us that at least the DB forest we eventually retained is clearly not that bad. And a systematic analysis over the three entire considered massifs showed that i it lead the highest accuracy both in numbers and areas with regard to the other available forest data sources (response Table 4). Yet, better forest data could lead to results that are even more reliable. Also, a less stringent PRA/NoPra rule as function of NoForest/Forest (e.g. a higher PRA susceptibility with decreasing forest density) would for sure be an interesting option for further developments. These points will be precised in the revised version of the paper.
See our next responses about the quality of CLPA and DEM resolution.

[Figure]

*Response Figure 5: Comparison of forest extensions from Theia, Corine land cover, and DB Forest from IGN with aerial photographs (©IGN) taken in 2012 within the municipality of Le Sappey-en-Chartreuse, Chartreuse / Dent de Crolles study area (updates Figure 2).*

**Watershed delineation**

While I appreciate the simplicity of the watershed delineation approach. delineating PRAs is not new. The OBIA approach described in Bühler et al. (2018) does the same thing in a more sophisticated way. In my opinion. Section 3.1.3 and Fig. 4 explain the calculation of the flow direction. but do not actually show how the watersheds are delineated. Since the authors' method uses standard tools available in open-source GIS software. it might be more useful for the reader to get a detailed description of how these calculations are done in freely available GIS software.

**Author's response:** The watershed algorithm we use is the one from ARCGIS that follows the references in text and the workflow of Figure 4. Following this suggestion we will detail the principle of the algorithm a bit more in text and move current Figure 4 to the supplements as is does not really belong to our results.
Also we agree that in essence the idea is similar to the OBIA approach of Bühler et al. (2018), even if we have to say that we were not able to fully understand its details from the paper (we tried hard!). We will add in the revised version of the paper that both approach follow more or less the same rationale.

**Validation of PRA identification**

I see several fundamental challenges in the current validation approach that. in my opinion. provides a very biased perspective on the performance of the PRA model.

1) The authors' choice to only evaluate the performance of the model within areas of documented avalanches means that they only test whether the PRA algorithm can identify start zones in known avalanche path (true positives). It does not provide any insight about the algorithms ability to ignore terrain where avalanche do not start outside of the known avalanche paths (true negatives). While the authors explain their approach when they introduce their modified confusion matrix (L290+). this does not seem to be very meaningful to me. As explained by the authors. avalanche cadaster datasets are not widely available and have limitations in many areas. The purpose of PRA models is to identify PRA in areas where direct observations are not available. In my opinion. a more meaningful approach would be to validate the model in areas with high confidence in the avalanche mapping record and include both avalanche terrain and non-avalanche terrain so that the complete confusion matrix can be properly evaluated. As can be seen in Fig. 3 and 7. there are considerable areas outside of the avalanche path areas that the algorithm incorrectly identifies as PRAs.

2) Applying the PRA model steps (e.g.. > 1400 m. slope incline between 28 and 60°. watershed delineation. etc.) to the CLPA dataset before conducting the validation completely defeats the purpose of a validation. Obviously. the model will perform well if the validation only includes terrain with the same characteristics. In the end. the authors only evaluate the steps in the PRA algorithm that are not included in the CLPA preprocessing (slope curvature?). This is a fundamental weakness of the paper.

3) The simplified confusion matrix and calculation of the accuracy and error rates derive directly from the authors' choice of only examining true positives and false positives. As pointed out above. I do not think this is meaningful. Simply assuming that the true negative is 100% is not meaningful and leads to inflated accuracy rates. It is also unclear to me why the authors use percentages in their confusion matrix calculations. Confusion matrices are general populated with counts. which is possible for evaluating both the identification of individual PRAs and the total area of PRAs.

4) Nowhere in the manuscript is explained how the author identify a match between a PRA identified by the algorithm and the validation dataset. Is 100% overlap required or do the authors use a different rule to distinguish true positives from false positives?

5) Only focusing on the accuracy rate is a very simple evaluation of performance. Furthermore. since the error rate is simply the complement to the accuracy rate. having the error rate in all the tables does not add any value. The use of this simple validation measure is very much at odds with the content of the paragraph on model evaluation in the introduction (L87+). where the authors seem to highlight the value of more advanced evaluation approaches. This seems a missed opportunity for contributing to the literature.

6) The repeated statement that the validation in this study is done over large areas of terrain (i.e.. entire massifs) is incorrect. The validation was conducted within documented avalanche paths within these massifs. which. as highlighted in Fig. 7. are generally very small areas.

**Author's response:** We fully agree that the validation is the crucial issue Even if, with the CLPA, we have a very valuable data support, it was for us the main source of questions and concern during the work. As said in our main comment, it is also the point on which, even if clearly we do not pretend to solve the problem, we may bring some methodological/generic outcomes/thoughts for the community. This is why we choose to focus on the validation at several points in the paper, and even in the title with the "ground truth" words. But we agree that our rationale was not clear enough and the review process helped us to formalize our thoughts as follows:
- Despite drawbacks inherent to any avalanche cadastre, the CLPA is an excellent source of information regarding past avalanches, possibly one of the finest worldwide (in terms of compromise between a very large extent and a high data quality over the area), due to its old history, its extremely regular update by devoted technicians, continuous financial support from the French ministry of the environment and because it includes various complementary sources of information (testimonies, landscape footprints, etc.). This makes it over the years closer and closer to the true maximal avalanche prone terrain. From that perspective, it is perfectly suited to evaluate a method that aims at automatically identify the maximal avalanche prone terrain as we aim at. Notably, as CLPA extension polygons are concatenations/unions of all observed avalanche extensions on a given avalanche path, CLPA is more likely to provide an accurate estimate of the entire "ground truth" than any observation of single avalanche events. See the CLPA extracts below that will be inserted in the paper as an additional supplementary figure (response Figure 1).
- Yet, visual results and scores must be interpreted with care due to the peculiarities of CLPA data. Within a given massif, some areas are covered by CLPA and some of them are not (response Table 2, response Figures 2, 7, 8 and 9). Accuracy scores are evaluated only over areas which are covered by CLPA, namely much larger areas in Mont Blanc and Maurienne Massifs than in Chartreuse massif (response Table 2). By contrast, in areas covered by CLPA, it is known that CLPA is very good on lower slopes and in forested terrain and more likely to miss avalanche prone terrain (and release areas) at high elevations, far from inhabitants and forests (response Figure 7). In such latter cases, "false positives" are likely to be often avalanche extents that are missing in CLPA.
- CLPA does not distinguish release areas from flow paths and runout zones, which implies that a pre-processing is required to isolate individual release areas within CLPA extensions that can be compared with our PRAs. Hence, for us, the issue is not that the CLPA validation data is not "ground truth" but that indeed the predicted PRAs and validation data are not independent (they are initially, but the pre-processing of the CLPA with the slope, forest, etc. filters introduces some dependency). However, let us say boldly that we are almost sure that obtaining a fully independent sample of "ground truth" PRA is simply not possible. Indeed, even "live", one never really observes a release area, but the full extension of an avalanche, and delineating the release area always involves some subjectivity (except, maybe, with high speed camera and films that can

be watched in slow motion to see the avalanche at its earliest stage…). Also, assuming one is able to observe a "true" release area, there is little chance that the entire PRA is observed. Consequently, the definition of any validation sample will always involve some partially subjective and more or less explicit choices, with possible use of some filters (slope, etc.) similar to those we use. Our choice was to do it and to say it explicitly in a transparent manner.

- Even with the best validation sample at hand, one will never be sure that i) all potential PRAs have been spotted, ii) the maximal potential extension that can be released under the most extreme conditions has been spotted for all PRAs (see our previous point about place where CLPA is good / less good). As a consequence, for us, only "true positives" can be trustfully validated, as it is never sure that a complete PRA or a part of a PRA automatically identified but not present in the validation sample is not simply missing from the validation sample. This is why we focus in our validation approach on accuracy scores/ true positives only.

- Accuracy scores (or other quantities related to confusion matrixes) were seldom used so far to evaluate PRA detection methods, to our knowledge, only in Bühler et al (2018) and with one single metric. Even if this is far from nothing, this is not much. Especially, we stress that one metric is not enough to judge the accuracy of a PRA detection method, as, e.g. the right number of PRAs can be identified but with wrong extents, and vice-versa. As a first step, we propose to evaluate accuracy scores both for PRA numbers and areas, which may cover the two most critical dimensions of the problem, but additional complementary metrics should probably be used as well in the future (focusing e.g. on the shape of PRAs, their elevation, etc.).

- Eventually let us state that focusing the PRA search and validation on terrain which are presumably favourable to avalanches is arguably not a bad idea. It is a rather standard approach in machine learning that is increasingly used in susceptibility mapping approaches outside the snow avalanche field in order to focus the detection on most suitable areas and thus increase the detection power. But we agree that this should be considered when interpreting the obtained scores.

In the reworked paper, we will reinforce these points in the discussion and justification of the research objectives and approach to highlight the limits and outcomes of our work on PRA validation for the community. Yet, we will remove "ground truth" from the title in order to avoid any misinterpretation. And we will even better explain the peculiarities of the CLPA data (we tried to do it already but we understand that this may be hard to understand from different countries with different ways of collecting, presenting and using avalanche data), and how they affect the results.

We will also remove unnecessary numbers in confusion matrixes and Tables, focusing only on accuracy scores on numbers and areas. These sum up all information related to true positives that we can decently evaluate (see response Tables).

We agree that how the matching between the detected PRA and the validation sample is done was not sufficiently clear. Detected PRAs and the validation sample are considered as polygons. In terms of numbers, a detected PRA and a validation polygon match as soon as their intersection is non-zero. The confusion matrix in area is computed by evaluating intersected areas. This will be added to the revised manuscript.

We will eventually further discuss the obtained scores and how they should be interpreted (see our response below about "sensitivity study"), with the support of the additional in-depth large scale parametric/sensitivity study that includes the effect of change on the validation sample.

[Figure]

*Response Figure 6:* *Extracts of the Official French avalanche cadastre "CLPA" (March 2022 edition). Magenta end orange polygons correspond to the extent of past avalanches from i) testimonies and documentary sources and ii) photo-interpretation of landscape footprints, respectively. Full legend at ;https://www.avalanches.fr/static/1public/epaclpa/CLPA_feuilles_carte/CLPA_legende_carte.pdf). Small study areas of Chamonix and Chartreuse/Dent de Crolles are located, as well as the limits of the areas covered by CLPA in both massif.*

[Figure]

***Response Figure 7:*** *Result of the proposed PRA detection method (Fig. 5) for the entire Mont Blanc massif. Digital Elevation Model ©IGN.Updates Figure S2.*

**Challenges in sensitivity analysis**

As mentioned above. the sensitivity analysis does not seem sufficiently rigorous to provide meaningful insight. For example. the authors only compared the benefit of the 1400 m elevation threshold to not having an elevation threshold at all. Why not test other threshold values (1300 m. 1500 m. etc.)? The sensitivity analysis also only examines certain parameters and leaves out others without explanation. In my opinion. properly deriving the parameter selection and thresholds from the available data is a critical piece that needs to be included in this paper.

Given that the authors use a low-resolution DEM that is far below the quality recommended in the literature. it seems critical that this choice is justified with a proper sensitivity analysis. While the performance of the lower-resolution DEM will likely be lower. its use can still be justified based on a cost-benefit argument. but it will be important to have the comparison to better understand the consequences of this choice.

Some of the main results of the sensitivity analysis do not seem to match common sense. The fact that including slope incline increases the accuracy rate by less than 2% seems wrong as incline is one of the primary determining characteristics of avalanche terrain. In my understanding. this odd result is the direct consequence of only looking at terrain within documented avalanche paths and processing the validation dataset with PRA algorithm rules. which clearly highlights the limitations of the validation approach taken in this study.

Author's response: Let us first state that we do not really perform a sensitivity analysis, which would require a much more sophisticated approach that what we do. Such approaches have been introduced in the snow avalanche field only recently (e.g. Heredia et al., RESS 2021) and this remains probably to be done for the specific issue of PRA detection. So we prefer speak of a parametric study.

Regarding the different parameters / thresholds, we now provide systematic results all over the Mont Blanc Massif (response Tables 4 and 5). Table 4 is not a complete grid search, which is something numerically intensive and, to our opinion, not mandatory given our objectives and the ways our scores and results should be interpreted, see below. However, it is a systematic search moving each parameter one by one. Also, we expanded the results that were already in the paper by evaluating how accuracy scores evolve when applying the different steps/filters of the method successively, and with Theia and Corine land cover forest data instead of the IGN DB forest data. This could be done all over the Mont Blanc, Maurienne and Chartreuse Massif (response Table 4). We also investigated the effect of DEM resolution on i) scores (response Table 5) and ii) the visual aspect of detected PRAs (Response Figure 8). As already stated, results overall highlight i) decreasing accuracy rates as one leaves the default values determined from the literature and used for the identification of the validation sample, ii) rather stable accuracy. Yet, it can be noted that, when removed one by one, the watershed delineation step and the forest and minimal area filters appear as important, at least in terms of PRA numbers. Regarding the resolution issue, performing the PRA detection with higher resolution DEMs does not improve the results when the same validation sample is considered (i.e. determined with the 25 resolution DEM). Yet, more numerous PRAs of smaller areas are detected.

The effect of the DEM resolution was further studied by investigating how it affects the determination of the validation sample and the subsequent accuracy scores after the PRA detection (response Table 5, response Figure 9) highlighting again limited added value (and even a slight decrease in accuracy) with higher DEM accuracy.

Eventually we understand that our scores may appear as erroneously high. We want to remember that they concern the "true positives" only, and that we are fully aware that they are not independent of

the way the validation set up is designed. As a consequence, they should not be directly compared with scores obtains with other approaches on other data sets. We simply consider them as probative enough to suggest that our approach performs rather well in the French context. Visual inspections on well-known areas confirmed that the PRA detection was indeed able to perform rather decently. In the same spirit, our parametric search should not be seen as a way to determine a truly optimal combination. We do not think we have the data that would allow this. Simply, we are able with it to determine a set of values, or different ranges of values, that perform rather well, and it is probably a good thing that the method is not too sensitive to very slight changes over reasonable ranges of most important parameters/thresholds.

These results will be inserted and more deeply analysed in the revised version of the paper. Also the discussion section will stress even more than in the current version how our scores and results should be interpreted.

| | | With default values | Without Minimal elevation filter | Without slope filter | Without distance to ridge filter | Without minimal area filter | Without watershed delineation | Without forest filter | With Corine Land Cover forest | With Theia forest |
|---|---|---|---|---|---|---|---|---|---|---|
| Validation sample | Total area of PRAs within CLPA (validation sample) | 58.3 | 58.3 | 58.3 | 58.3 | 58.3 | 58.3 | 58.3 | 58.3 | 58.3 |
| | Total number of PRAs within CLPA extensions (validation sample) | 1522 | 1522 | 1522 | 1522 | 1522 | 1522 | 1522 | 1522 | 1522 |
| Detected PRAs | Total area of detected PRAs | 90.80 | 98.13 | 97.95 | 102.35 | 102.78 | 96.60 | 135.11 | 109.01 | 104.14 |
| | Delta area with regards to default values [km2] | / | 7.33 | 7.15 | 11.55 | 11.98 | 5.80 | 44.31 | 18.21 | 13.34 |
| | Delta area with regards to default values (%) | / | 8.1% | 7.9% | 12.7% | 13.2% | 6.4% | 48.8% | 20.1% | 14.7% |
| | Total number of detected PRAs | 2003 | 2085 | 2074 | 2173 | 4315 | 262 | 2803 | 2170 | 2218 |
| | Delta numbers with regards to default values | / | 82 | 71 | 170 | 2312 | -1741 | 800 | 167 | 215 |
| | Delta numbers with regards to default values (%) | / | 4.1% | 3.5% | 8.5% | 115.4% | -86.9% | 39.9% | 8.3% | 10.7% |
| | Total area of detected PRAs within CLPA extensions | 84.89 | 86.63 | 86.64 | 89.28 | 87.12 | 17.16 | 102.29 | 93.08 | 89.76 |
| | Total number of detected PRAs within CLPA extensions | 1601 | 1622 | 1623 | 1626 | 2061 | 111 | 1597 | 1528 | 1607 |
| Accuracy | Accuracy rate (Eq. 3) — In numbers | 90 | 88.9 | 89.1 | 87.4 | 73.9 | 71.1 | 78.5 | 83.3 | 85.9 |
| | Accuracy rate (Eq. 3) — In areas | 96.8 | 94.1 | 94.2 | 93.6 | 92.4 | 58.9 | 87.9 | 91.9 | 93 |

| | | | | | | | | | | |
|---|---|---|---|---|---|---|---|---|---|---|
| Delta Accuracy with regards to default values | In numbers | / | -1.1 | -0.9 | -2.6 | -16.1 | -18.9 | -11.5 | -6.7 | -4.1 |
| | In areas | / | -2.7 | -2.6 | -3.2 | -4.4 | -37.9 | -8.9 | -4.9 | -3.8 |

**Response Table 4**: *Evolution of accuracy rates when applying the different steps/filters of the method successively, and with Theia and Corine land cover forest data instead of the IGN DB forest data all over the Mont Blanc Massif. Total area of detected PRAs and total number of detected PRAs are those of the part of the massif covered by CLPA. The table expands information that was provided previously in Table 4 and 6 for the small Chamonix area only. For the revised paper, similar Tables will be produced for the entire Maurienne and Chartreuse massif.*

| | | With default values | Resolution 10 m | Resolution 10m. adjusted validation sample | Resolution 5m | Resolution 5m. adjusted validation sample |
|---|---|---|---|---|---|---|
| Validation sample | Total area of PRAs within CLPA (validation sample) | 58.3 | 58.3 | 53.5 | 58.3 | 45.9 |
| | Delta area in validation sample with regards to default values | / | / | -4.8 | / | -12.4 |
| | Delta area in validation sample with regards to default values (%) | / | / | -8.3% | / | -21.2% |
| | Total number of PRAs within CLPA extensions (validation sample) | 1522 | 1522 | 2061 | 1522 | 2181 |
| | Delta numbers in validation sample with regards to default values | / | / | 539 | / | 659 |
| | Delta numbers in validation sample with regards to default values (%) | / | / | 35.4% | / | 43.3% |
| Detected PRAs | Total area of detected PRAs | 90.8 | 84.0 | 88.9 | 71.8 | 71.7 |
| | Delta area with regards to default values [km2] | / | -6.8 | -1.9 | -19.0 | -19.1 |
| | Delta area with regards to default values (%) | / | -7.5% | -2.0% | -20.9% | -21.0% |
| | Total Number of detected PRAs | 2003 | 2989 | 3081 | 3131 | 3130 |
| | Delta numbers with regards to default values | / | 986 | 1078 | 1128 | 1127 |
| | Delta numbers with regards to default values (%) | / | 49.2% | 53.8% | 56.3% | 56.3% |
| Accuracy | Total area of detected PRAs within CLPA extensions | 84.9 | 76.8 | 76.6 | 66.8 | 64.1 |
| | Total number of detected PRAs within CLPA extensions | 1601 | 2420 | 2339 | 2694 | 2457 |
| | Accuracy rates — In numbers | 90 | 87.9 | 88 | 87 | 89.3 |
| | Accuracy rates — In areas | 96.8 | 95.1 | 93.1 | 94.9 | 94.7 |
| | Delta Accuracy with regards to default values — In numbers | / | -2.1 | -2 | -3 | -0.7 |
| | Delta Accuracy with regards to default values — In areas | / | -1.7 | -3.7 | -1.9 | -2.1 |

**Response Table 5**: *Evolution of accuracy rates with DEM resolution on PRA detection and selection of the validation sample for the entire Mont Blanc Massif.*

[Figure]

*Response Figure 8: (b-d) Effect of DEM resolution on PRA detection. Chartreuse/Dent de Crolles study area. a) shows the validation sample obtained with the default setting. The absence of CLPA in the upper left corner is clearly visible.*

[Figure]

*Response Figure 9: Effect of DEM resolution on PRA detection (right) and selection of the validation sample (left). Chartreuse/Dent de Crolles study area. The absence of CLPA in the upper left corner is clearly visible.*

**Comparison with other algorithms**

In my opinion. comparing the algorithm introduced in this paper with some of the established methods would be very important for highlighting the value of the new approach. I think this should be included in this manuscript.

**Author's response:** We are sorry but this is where we simply cannot go at this stage. Doing so would imply having access to existing algorithms as open access codes/routines. which is currently not the case. Obviously, the principle of existing algorithm is published, but the description is not always very precise and easy to follow. And even when it is, following published guidelines/equations do not guarantee that what the different authors have done can be reproduced exactly, due toe, e.g.. differences introduced by different numerical implementation schemes. Therefore, the only comparison that can be done at this stage is the one regarding the choice of the different parameter values and/or steps of the approach. This was already done in the previous version of the paper but will be reinforced in the reworked discussion.
However, we really think that such an inter-comparison on different data sets (as results may be to some extent case-study dependent) would be beneficial for the community and we would be very happy to contribute. This is a reason why, as a first step, we provide the full data set corresponding to the paper in open access.

**Limited discussion**

In its current form. the discussion primarily repeats information from earlier sections of the manuscript (i.e.. introduction. methods and results) without adding much value. This is partially due to the fundamental limitations mentioned above. The tone of the discussion is also quite casual (e.g.. L529: "All in all. the validation data we use is certainly not perfect and our validation approach may potentially favour the comparison with our detected PRAs. …"). which does not seem appropriate for a scientific publication. See the technical comments section for additional comments. The outlook section does not seem to offer any novel ideas as it primarily discusses already existing application cases for PRA maps (e.g.. large scale mapping of avalanche hazard and risk) and existing research extensions (e.g.. PRAs conditional on snow and weather conditions. probabilistic detection rules).

**Author's response:** The whole discussion will be deeply reworked in the revised version of the paper to remove the unnecessary material, and, by contrast, to expand the discussion with respect to our answers to the different comments. See also our responses concerning the tone.

**SECONDARY ISSUES**

**Set up of the research objective and expectations**

To motivate their study. the authors provide a fairly comprehensive. even though not completely up-to-date. summary of the existing literature on PRA identification in the introduction. In this overview. they identify several limitations of the existing approaches (e.g.. disagreement about relevant terrain factors. delineation of individual PRAs. validation of PRA algorithms) to set the stage for their research objective on L103. This setup creates the expectation (explicitly or implicitly) that the algorithm introduced in this manuscript will address these issues. There are multiple issues with this. First. I do not agree with all the claims that are made in the introduction. The terrain parameters included in the various PRA algorithms do not differ from each other that much. Bühler et al. (2018) have presented an approach for meaningfully delineating PRAs. and the most cited algorithms have been validated

with mapped avalanche datsets. Second. the paper does not deliver on these expectations due to the methodological issues mentioned above. This results in disappointment and sets the paper up for failure.

I think the manuscript would benefit from a more focused introduction that describes the research objective more honestly and positions the study within the existing literature more accurately. I have no problem with a study that aims to create a simple approach for PRA identification based on easily accessible datasets. but this objective should be clearly stated at the beginning of the paper to create meaningful expectations.

**Author's response:** We will rework the end of the introduction to better state the different research objectives of our work (see our previous answers, notably our first answer concerning the overall objective of the study). We will also delate the material unnecessary at this stage in the revised manuscript, but we think it is good for the reader to have a hint already at this stage of the paper of the workflow and the main outcomes with regards to these objectives.
Eventually, we fully agree that the approach and parameters we eventually retain is largely in accordance with the existing literature (except maybe for the resolution that we found less important than in other studies with regards to our chosen validation metrics and data), but we never intended to say something different. We will take extra-care in the revised manuscript to clarify this.

**Language and tone**

Overall. the quality of the English in this manuscript is not very high. and the text includes many terms that are not meaningful in this context (see partial list in technical comments). While the use of these terms might the result of a poor translation from French. it is important to check the manuscript for proper use of terminology before publication.

Furthermore. the tone of the writing is rather ambitious and glowing. Examples include "In this paper. a method that well identifies …" (L16 and L103). "… the CLPA is a very valuable source of information. … and. arguably. among the rare existing ones …" (L197). and "… the CLPA is almost surely a true avalanche extent." (L188). The further exasperates the reader's sense of unfulfilled expectations mentioned above.

**Author's response:** We agree that the English of the paper was largely improvable. In addition to changes in structures and of some sentences, the revised version of the paper will be proofread by a native English speaker.

Regarding the overall tone, we are really sorry if the referee has been exasperated by our limited skills in English, this was obviously really not our objective. We will try to improve with the help of the professional proofreader.

However, regarding the specific points mentioned by the referee, we accept that we said it badly or at least too casually, but we stick on the idea of the exceptional quality of the CLPA data (at least in terms of combination of a very large extent over which the data is of high quality). See our previous specific response about CLPA.

**Figures and tables**

The figures included in this paper are not of high quality. Several are hard to read (e.g.. Fig. 2). and the figure layout and formatting of the legends seem different in every figure. In my opinion. Fig. 5 and 6 do not add any value beyond what is already explained in the text. It is also unclear to me why the

validation maps for the Mont-Blanc and Maurienne massifs are currently in the supplementary material and not included in the main manuscript.

Captions of tables are typically presented above tables. In all confusion matrix tables. the different components of the confusion matrix should be properly labelled.

**Author's response:** We produced high quality Figures for the first version of the paper already, and a loss of resolution simply occurred because the submission as a single pdf file was mandatory in the system (maybe we missed something). We are sorry but the resulting bad resolution of some of the Figures is not our fault. However, we worked hard to improve the quality of the figures even more in terms of Layout, resolution, scope, etc. Examples have been provided before and the new very high quality figures will be provided in the revision.
We also agree that the Figures corresponding to the Mont Blanc and Maurienne Massif should be moved to the paper core, the overall number of Figures is not that high.
And we will check the position and content of all Table labels.

**TECHNICAL COMMENTS**

**Author's response:** We again thank the referee for his/her careful reading of our paper. All edits will be corrected and all suggestions, when not specifically answered below, will also be fully integrated in the revised version of the manuscript.

**Abstract**

L10: 'lacunar' is not a meaningful word in this context.

L18: 'Confrontation' should be 'Comparison'.

**Introduction**

L30: Extremely convoluted sentence.

L64: 'Retained' should probably be 'included'.

L83: 'on the field' should probably be 'in reality'.

L98: I am not sure whether the existing PRA algorithms are 'competing'.

**Author's response:** Formulation will be amended.

L104: Should be '… where avalanches can occur'.

L108-120: This preview of the methods and results is not necessary in the introduction. It is best to finish the introduction with the statement of the research objective.

**Author's response:** We will rework the end of the introduction to even better state the different research objectives of our work (see our first answer). We will also delete the material unnecessary at

this stage in the revised manuscript. However. we think it is good for the reader to have a hint already at this stage of the paper of the workflow and main outcomes with regards to these objectives.

**Data**

L134: 'reputed' should be 'well known'.

L176: All abbreviation need to be properly introduced the first time they are used in the manuscript. There are additional abbreviations that have not been introduced.

L176: It should be 'It consists of…'.

L179: I don't not understand what is meant with '… is mainly produced at the destination of …'.

**Author's response:** simply "the target audience is". This will be corrected.

L192: '… near human stakes …' should be '… near human assets or settlements.'

L225: 'In order to …' can be simplified to 'To …'. There are several cases of this in the manuscript.

L241: The sentence that describes how the thresholds and parameters are chosen (reference to Sect 4.2) does not seem to belong here.

**Author's response:** We will highlight in a specific place of the default value for the different thresholds and parameters have been fixed.

L245+: The description of the algorithm provided in this section does not seem consistent with the information presented in Fig. 5.

**Author's response:** We will slightly rework the description to improve its clarity.

L261: The statement that PRA identification target primarily large avalanches needs to be stated much earlier in the manuscript as it is a fundamental assumption of the study.

**Author's response:** It will be moved earlier in the paper (introduction).

L271+: The description of the CLPA seems repetitive as it discusses information that was mentioned previously already.

**Author's response:** We will delete the unnecessary material.

L280: Typo: PRAS should be PRAs. There are several instances of this typo in the manuscript.

**Results**

L327+: The description of the confusion matrix provides exactly the same information that is shown in the table. Hence. it does not add any additional value.

**Author's response:** We will delete the unnecessary material.

L349: I do not know what you mean with 'probative'. There are several uses of this work in the manuscript.

**Author's response:** see our responses about validation and scores.

L350+: The explanation provided here seems rather speculative and not well grounded.

**Author's response:** We will ground our explanations on the characteristics of the massifs and of the CLPA in them.

L371: Not sure what you mean with 'parametric study'.

**Author's response:** see our responses about sensitivity analysis.

L380: In academic writing. the term 'significant' should only be used in the context of statistical significance. Use 'considerable' or substantial' instead.

L382: 'Use 'more substantially' instead of 'more largely'.

L406: The last sentence in this paragraph is too hand-wavy and not grounded in evidence.

**Author's response:** We will ground our conclusions on the results of the parametric studies.

L410: Table 6 does NOT show that not including the forest layer results in the worst performance. The accuracy rates without forest are higher than with the theia dataset.

**Author's response:** see Response Table 4 that confirm our results all over the Mont Blanc Massif.

L421: Delete 'eventually'.

L425: Use 'tower' instead of 'pylon'.

L435: Delete 'eventually'.

**Discussion**

L447+: There is no need to repeat the information from the intro at the beginning of the discussion section.

**Author's response:** The whole discussion will be deeply reworked in the revised version of the paper. Regarding this specific point, we will delete unnecessary repetitions, but we think it is important before starting the discussion to remember briefly the objectives of the work and what has been achieved.

L455+: I do not think these objectives have been achieved by this study.

**Author's response:** see our responses about paper objectives, validation and scores.

L469: I am not sure what is meant with 'probative'.

**Author's response:** see our responses about validation and scores.

L475+: This discussion primarily repeats information from the methods and results section without adding much value.

**Author's response:** The whole discussion will be deeply reworked in the revised version of the paper to remove the unnecessary material, and, by contrast expand the discussion with respect to our answers to the different comments. See also our responses concerning the tone.

L499: Reword this sentence. It should be '… by comparing X to the processes CLPA dataset.'.

L503+: This sentence seems like an excuse and is not very convincing. The readers' expectations should be managed by properly describing the research objective in the introduction.

**Author's response:** see our responses about paper objectives, validation, scores and comparison of existing algorithms.

L507: 'Envisaged' should be 'envisioned'.

L510: 'Confronted' is the wrong word here.

L529: Similar to the sentence on L503. this sounds like an excuse. Limitations should be discussed more seriously.

**Author's response:** see our responses about paper objectives, validation, scores and comparison of existing algorithms.

---

## Author Response (AR1)

Nicolas Eckert
Grenoble Alpes University, INRAE
UR ETNA, France

Grenoble, December 17th, 2022

Submission of a revised version of our article to *NHESS* now entitled "Development and evaluation of a method to identify potential release areas of snow avalanches based on watershed delineation

Dear Yves Bühler, NHESS scientific editor.

We deeply thank both referees for their feedback on our article and their insightful comments. We also thank you for your editorial revue and your suggestions. All points have been addressed in the revised version attached to this submission, except that we did not make recourse to a professional English proofreader, as this would have delayed the resubmission too much. We did ourselves our best to improve the language and tone, but if this is found insufficient, we will do it.

In what follows, we further provide a point-by-point answer to all comments and questions and detail the changes made in the revised manuscript.  Thank you for your consideration of our work.

Sincerely,

Nicolas Eckert, on behalf of the authors

Review of the paper **Development and validation using ground truth of a method to identify potential release areas of snow avalanches based on watershed delineation** by Cécile Duvillier, Nicolas Eckert, Guillaume Evin and Michael Deschâtres

**General comments:**
The paper addresses an issue which is well-known in avalanche science. Potential avalanche release areas (PRAs) are one of the most important parameters to be identified for avalanche hazard assessment (for ex. hazard maps, design of defense measures, etc.).
The Introduction is very rich and refers to an exhaustive literature about the PRA definition topic. Good point! And it explains clearly which are the strong and weak points of existing methods, in particular about the validation method. At the end of the Introduction, then, it clearly states the aim of the paper, which is, beside the PRA definition method, also the development of a test and validation method (see my comments in the following).
**Author's response:** We thank Margherita Maggioni for her positive feedback of our work. Let us just stress here that the main objective of the paper is to develop a PRA detection method that grounds on already existing developments (notably by herself, Yves Buhler and their co-authors) and works reasonably well in the context of the French Alps. This is shown by performing an evaluation/validation exercise relying on an excellent data source regarding past avalanches, the CLPA. However, in addition to the benefit for avalanche hazard assessment in the French context, there are also some slight methodological outcomes of the paper that may be of broad relevance for the topic:

      i)        The determination of individual PRAs using a watershed delineation algorithm;
      ii)      A validation approach on the basis of accuracy scores computed using two metrics, PRA numbers and area;
      iii)     Broader findings and reflexions about how to validate a PRA detection method, notably how can a validation data sample be defined, and which scores can be interpreted.

None of these points are completely new in the community, but we find that they have not been fully answered so far in the literature, and we humbly hope that our paper will therefore bring some new useful elements to the debate. The questions raised by both reviews (and notably by referee two) however indicate that these objectives were not clear enough in the first version of the paper. The paper has therefore been largely reworked to better introduce the research questions and discusses the findings and the approach with regards to these questions.
See below our specific response about the validation.
Eventually, let us note here that to make the validation and parametric study more convincing, they have been largely expanded in the revised version of the paper i) in terms of potential set of parameters by conducting a much more comprehensive parametric study, ii) by considering into the analysis the questions of the DEM resolution, iii) by performing the study all over 3 entire massifs and not only over a small area, and iv) by investigating the relation between the determination of the validation sample and the accuracy scores (see additional results / figures below and in our response to referee 2). An additional small area within the massif of Chartreuse (Chartreuse study area / Dent de Crolles) is also now considered to better highlight/illustrate some results.

Concerning the PRA delineation method: I think it is a good method which takes what already exists and adds a GIS watershed routine to define single PRAs. This part is more a GIS technical issue than an avalanche science issue… but it seems that the authors found a good solution to a challenging delineation problem.
**Author's response:** We thank again Margherita Maggioni for her positive judgment on our work.

Concerning the PRA validation method, I think that confusion matrix and evaluation scores are a good proposal… but the weak point is the pre-processing of the CLPA data. Finally, the authors apply the same procedure developed for the PRA identification to the CLPA database to extract the release areas from the polygons of the entire avalanche extensions… Therefore, they use the same method to produce the PRAs and, in a way, to extract a dataset to validate it. I mean, *we do what we can with what we have*, but then I would not stress so much the importance of the validation method - it is even in the title and using the word "ground truth". For me "ground truth" used in relation to potential release areas should mean observed release areas… Actually, this is the common problem of all the validation methods for PRA delineation, as not many release area datasets exist. For example, also Harvey et al. (2018) made something similar in the analysis of release areas of ski triggered

avalanches. In the discussion section, this problem is actually well considered and truly presented. Though, I would not use "ground-truth"...

Stephan Harvey, Gunter Schmudlach, Yves Buhler, Lukas Durr, Andreas Stoffel, Marc Christen, AVALANCHE TERRAIN MAPS FOR BACKCOUNTRY SKIING IN SWITZERLAND, Proceedings, International Snow Science Workshop, Innsbruck, Austria, 2018 and presentation for the CSAW 2020 ( https://www.oegsl.at/automatic-high-resolution-mapping-and-classification-of-avalanche-terrain-regarding-potential-release-triggering-and-run-out-zones/?lang=en time 3:16 - 3:22)

**Author's response:** We fully agree that the validation is the crucial issue. Even if, with the CLPA, we have a very valuable data support, it was for us the main source of questions and concern during the work. As said in our main comment, it is also the point on which, even if clearly we do not pretend to solve the problem, we may bring some methodological/generic outcomes/thoughts for the community.  This is why we choose to focus on the validation at several points in the paper, and even in the title with the "ground truth" words. But we agree that our rationale was not clear enough and the review process helped us to formalize our thoughts as follows:

- Despite drawbacks inherent to any avalanche cadastre, the CLPA is an excellent source of information regarding past avalanches, possibly one of the finest worldwide, due to its old history, its extremely regular update by devoted technicians, continuous financial support from the French ministry of the environment and because it includes various complementary sources of information (testimonies, landscape footprints, etc.). This makes it over the years closer and closer to the true maximal avalanche prone terrain. From that perspective, it is perfectly suited to evaluate a method that aims at automatically identify the maximal avalanche prone terrain as we aim at. Notably, as CLPA extension polygons are concatenations/unions of all observed avalanche extensions on a given avalanche path, CLPA is more likely to provide an accurate estimate of the entire "ground truth" than any observation of single avalanche events. See the CLPA extracts below that is now included in the revised SM of the paper.

- CLPA does not distinguish release areas from flow paths and runout zones, which implies that a pre-processing is required to isolate individual release areas within CLPA extensions that can be compared with our PRAs. Hence, for us, the issue is not that the CLPA validation data is not "ground truth" but that indeed the predicted PRAs and validation data are not independent (they are initially, but the pre-processing of the CLPA with the slope, forest, etc. filters introduces some dependency). However, let us say boldly that we are almost sure that obtaining a fully independent sample of "ground truth" PRA is simply not possible. Indeed, even "live", one never really observes a release area, but the full extension of an avalanche, and delineating the release area always involves some subjectivity (except, maybe, with high speed camera and films that can be watched in slow motion to see the avalanche at its earliest stage…). Also, assuming one is able to observe a "true" release area, there is little chance that the entire PRA is observed. Consequently, the definition of any validation sample will always involve some partially subjective and more or less explicit choices, with possible use of some filters (slope, etc.) similar to those we use. Our choice was to do it and to say it explicitly in a transparent manner.

- Even with the best validation sample at hand, one will never be sure that i) all potential PRAs have been spotted, ii) the maximal potential extension that can be released under the most extreme conditions has been spotted for all PRAs. As a consequence, only "true positives" can be validated, as it is never sure that a complete PRA or a part of a PRA automatically identified but not present in the validation sample is not simply missing from the validation sample. This is why we focus in our validation approach on accuracy scores/ true positives only.

- Accuracy scores (or other quantities related to confusion matrixes) were seldom used so far to evaluate PRA detection methods, to our knowledge, only in Bühler et al (2018) and with one single metric. Even if this is far from nothing, this is not much. Especially, we stress that one metric is not enough to judge the accuracy of a PRA detection method, as, e.g. the right number of PRAs can be identified but with wrong extents, and vice-versa. As a first step, we propose to evaluate accuracy scores both for PRA numbers and areas, which may cover the two most critical dimensions of the problem, but additional complementary metrics should probably be used as well in the future (focusing e.g. on the shape of PRAs, their elevation, etc.).

In the reworked paper, we have considerably reinforced these points to highlight the limits and outcomes of our work on PRA validation for the community (Sect 5.3). Yet, we have  remove "ground truth" from the title in order to avoid any misinterpretation, and replaced "validation" by "evaluation" in the tittle which is more neutral.

We also removed unnecessary numbers in confusion matrixes and Tables, focusing only on accuracy scores on numbers and areas. These sum up all information related to true positives that we can decently evaluate.

Reworked discussion also further discusses the obtained scores and how they should be interpreted (see response to referee 2), with the support of an additional in-depth large scale parametric/sensitivity study that

includes the effect of change on the validation sample). Eventually, results/discussion now better insist on how the CLPA peculiarities affect the results (supplementary Figure 3).

The Results and Discussion sections might be shortened… but, actually, the reader can easily follow the *file rouge* of the whole story, therefore for me it is ok like they are (but see later my comment to lines 447-463).
**Author's response:** The results and discussion sections of the revised paper have been reorganised in order to avoid any redundancies and incorporate the new results and discussion regarding the validation, parametric study and main findings (see previous responses).

**Specific comments:**
Title: maybe it could be made simpler as follows: "Development and validation of a method to identify potential release areas of snow avalanches based on watershed delineation". This title doesn't stress too much the validation method, which is not really based on ground truth data, and instead stresses the watershed delineation which is something new in the PRA definition method. Or, even, the author could give a geographical information: "Development of a method to identify potential release areas of snow avalanches based on watershed delineation and validation in the French Alps". From the abstract (very clear!) It seems that the authors develop a validation method which might be applicable also for existing PRA methods. This gives expectations to the readers…
**Author's response:** We agree that "**Development and evaluation of a method to identify potential release areas of snow avalanches based on watershed delineation**" may be a fair title and we go for it in the revised version of the manuscript. See also our previous responses concerning the overall paper scope/objectives of the work and the specific issue of the methodological contribution of the paper in terms of validation of detected PRAs.

About the Introduction: I think it should finish with the description of the aim of the paper, without the rest, which seems a bit like an abstract… giving already information on which has been found after the analyses (but this section is only the Introduction!). For example, I would move lines 109-116 ("Following … individual PRAs.") to the Results or Discussion sections. And I think it is not necessary to explain how the paper is structured… It comes naturally while reading. Eventually the authors can move the sentences at the beginning of the corresponding sections.
**Author's response:** We have reworked the end of the introduction to even better state the different research objectives of our work (see our first answers). We also delated the material unnecessary at this stage in the revised manuscript, but we think it is good for the reader to have a hint already at this stage of the paper of the workflow and the main outcomes with regards to these objectives.

Line 28: ok for the reference to Amman and Bebi (2000) which is a general overview paper, while I would not refer to Braun et al (2020) which is a very specific one. Better, maybe, to refer to another general paper, or even a book (for ex. McClung and Shaerer, The avalanche handbook, 1993).
**Author's response**: We replaced the reference to Braun et al. by the reference to McClung and Shearer in the revised version of the paper.

Line 28-29: I would cancel this sentence "No countermeasure… one minute." and "therefore" at line 31. I understood the message but I think it is not necessary here.
**Author's response:** This sentence has been removed from the revised version of the manuscript.

It is very good how the authors state the reasons behind the choice of the different inputs (for ex. lines 143-169… and also make a kind of sensitivity analysis about this (Sect. 4.2).
**Author's response:** We thank Margherita Maggioni for this encouraging comment. However, following our first responses and our response to referee 2, we stress that in the revised version of the paper we largely expanded the parametric study (see our previous responses).

Line 181: I do not understand the reference to Figure 3… the sentence tells about areas without CLPA but the main outputs in Figure 3 are the resulting PRAs… Moreover, here we are still in Section 2. Data, therefore results should not be presented yet. I would cancel Figure 3. Instead, the authors might put a figure showing the areas covered by CLPA within the three test areas (Mont-Blanc, Chartreuse, Maurienne).
**Author's response:** We have reorganised the figures and text as suggested in the revised version of the manuscript. Also integrated in the paper core the figures providing the full results over the three massifs and added several new figures in the paper core and SM to better convey our message.

Line 232: Here it is not clear if the identification of individual watersheds is made automatically.
**Author's response:** It is indeed, using the existing algorithm from ARCGIS that follows the references in text and the workflow of Figure 4. This has been precised in the revised version of the paper. Note also that in the revised version of the paper, following suggestions of referee 2, we detail the principle of the algorithm a bit more in text and we moved the flow direction Figure to the supplements as is does not really belong to our results.

Line 259: I would specify that it is the **planar** area; also at line 265.
**Author's response:** This was added in the revised version of the manuscript

Line 316: I would move here Figure 3, which in fact shows the results of the PRA definition for the area of Chamonix. I think it is not necessary to highlight the pink area as "CLPA extension outside PRAs/AUTO"; "CLPA extensions" is enough. The figure would result less messed up (see Figure 7, S2 and S3, which are clearer).
**Author's response:** We have reorganised and reworked the figures as suggested.

Line 336: I would try to find a way to add a box (or a second figure) with a zoom on the area where CLPA exists (region in the lower-right corner) in order to better show the difference between matching (blue) and not matching (light blue) PRAs. Ok to put the other two figures (Mont Blanc and Maurienne) in the supplementary materials.
**Author's response:** We tried to improve the presentation of our results by adding the Chartreuse/Dent de Crolles study area. Also, the figure related to Chamonix study area has been improved. However we want to stress that the matching between detected PRAs and validation sample should be checked first in terms of massif-scale scores, which we will provide in a much more systematic way in the revised paper. This provides a much more fair assessment free of local effect and "cherry picking".

Line 447-463… these lines are a repetition of the Introduction… here it is time for discussion! :o)
**Author's response:** The whole discussion was deeply reworked in the revised version of the paper. Regarding this specific point, we deleted unnecessary repetitions, but we think it is important before starting the discussion to remember briefly the objectives of the work and what has been achieved.

Line 520-21… I do not understand the sentence… a verb is missing?
**Author's response:** Indeed the sentence was "The fact that we apply the same filters to the CLPA extension and to the whole terrain on which PRAs are detected also plays a role, sorry. This was corrected within the reworked discussion section.

**Technical suggestions:**
Figure 1. I would use a transparency for the violet and light blue colors to show the different study areas, so that the topography from the shaded DEM can be appreciated. **This is actually a general comment valid also for the other figures**.
**Author's response:** We have worked to improve the readability of the figures following the suggestions.

Figure 5: point (3) should be in *italic* and I would substitute parts with **areas**. In the second blue rectangle I would simply write **forest** instead of forest parts.
Line 43: "Wider benefits can also arise **FROM** the systematic …"
Line 327: **PRAs** instead of PRAS
**Author's response:** These edits have been corrected in the revised version of the manuscript.

Table 2: At the end I would add a reference to Table 1 as a legend for the Confusion matrix.
**Author's response:** This  has been done in the revised version of the manuscript.

Concerning the English: I am not the best person to judge the quality of the English… I would probably let the paper be revised by an English native speaker.
**Author's response:** We agree that the English of the paper was largely improvable. In addition to changes in structures and tone of some sentences (see answer to referee 2), will did our best to proof-read the paper.

This paper proposes a new method for identifying potential release areas (PRA) for snow avalanches based on terrain characteristics and validates the approach using a long-term avalanche cadaster dataset. The research is situated in France. and the study area includes three mountain massifs in the French Alps. Overall. the topic is interesting and relevant for the avalanche research community and the NHESS readership.

While I appreciate the authors' desire to create a practical. transparent. and computationally efficient algorithm for PRA identification that uses easily accessible datasets. there are several substantial weaknesses in the present study that. in my opinion. prevent this manuscript from being a meaningful contribution to the literature in its current form. Properly addressing several of my concerns would require a substantial redesign and/or expansion of the study. and I am unsure whether that can be accomplished within the current peer-review process. I hope that the following comments can help the authors to further develop their research.

**Author's response:** We deeply thank the referee for his/her meaningful suggestions and feedback. Even if some of the criticisms where for us a bit too strong, we took them as a challenge to clarify and improve our research and they certainly greatly helped us to improve the paper.

Let us just stress first that the main objective of the paper is to develop a PRA detection method that grounds on already existing developments (notably by Margherita Maggioni, Yves Bühler and their co-authors) and works reasonably well in the context of the French Alps. This is shown by performing an evaluation/validation exercise relying on an excellent data source regarding past avalanches, the CLPA in different massifs and areas of the French Alps. However, in addition to the benefit for avalanche hazard assessment in the French context, there are also some slight methodological outcomes of the paper that may be of broad relevance for the topic:

    iv)    The determination of individual PRAs using a watershed delineation algorithm;

    v)    A validation approach on the basis of accuracy scores computed using two metrics, PRA numbers and area;

    vi)    Broader findings and reflexions about how to validate a PRA detection method, notably how can a validation data sample be defined, and which scores can be interpreted.

None of these points are completely new in the community, but we find that they have not been fully answered so far in the literature, and we humbly hope that our paper will therefore bring some useful elements to the debate. The questions raised by both reviews however indicate that these objectives/questions were not clear enough in the first version of the paper. The paper will therefore be largely reworked to better introduce the research questions and discusses the findings and the approach with regards to these questions.

Eventually, let us note here that to make the validation and parametric study more convincing, we performed many additional analyses i) in terms of potential set of parameters, ii) by considering into the analysis the questions of the DEM resolution, iii) by performing the parametric study all over 3 entire massifs and not only over a small area, and iv) by investigating the relation between the determination of the validation sample and the accuracy scores. An additional small area within the massif of Chartreuse (Chartreuse study area / Dent de Crolles) is also considered to better highlight/illustrate some results. However the matching between detected PRAs and validation sample should be checked first in terms of massif-scale scores, which provides a much more systematic assessment free of local effects and "cherry picking".

What follows provide a point-by-point answer to the referee's comments, questions and suggestions and introduces the results of these additional analyses that have been fully integrated in the reworked version of the paper.

**PRIMARY ISSUES**

**Selection of terrain characteristics and threshold for PRA identification**

While the selection of terrain characteristics included in the PRA identification algorithm is based on existing literature. the reasons for their selection (or the refusal of other characteristics) are only discussed superficially. Furthermore. the selections of the parameter thresholds (e.g.. 1400 m elevation threshold. incline range) do not seem to be well grounded in evidence. I recommend that the authors conduct a proper grid search to determine the ideal parameter settings for their PRA identification algorithm. This is particularly important because they use a low-resolution DEM (25 m). which results in incline values that are biased towards lower values. This means

that the thresholds described in the literature are not necessarily applicable. While the current sensitivity analysis might intend to do this. it is not done in a very rigorous and scientifically valid way. See additional comment on sensitivity analysis below.

**Author's response:** As indicated before, we conducted many additional analyses to investigate how the choice of the different parameter values and thresholds affects the PRA detection in terms of PRA areas, numbers and accuracy scores. This systematic search was performed over the entire massif of Mont Blanc. Results overall appear as consistent, with globally decreasing accuracy rates as one leaves the default values determined from the literature and used for the identification of the validation sample. Overall, accuracy scores seem nevertheless rather stable over considered ranges of parameters/thresholds, with slope range being the most influential parameter over the tested range (up to a 10% decrease in accuracy for numbers). Also, specific areas have been analysed. They, e.g., show that a too large minimal extension, a too high minimal elevation or a too restricted slope range logically misses release areas that an expert analysis would definitely consider as suitable location for an avalanche release. These results are more deeply analysed in the revised version of the paper. See also our responses to the next comments, notably those related to data, validation and DEM resolution and related discussion regarding the dependency of the results on the validation set-up and the interpretation of the scores.

**Selection of datasets**

Several datasets used in this study seem to be of lower data quality than established best practices in the field of PRA identification suggest. For example. the forest data set seems to have considerable limitations and the DEM is of much lower resolution than suggested in the literature. While I do not have a problem with a let's-do-the-best-we-can-with-what-we-have approach (not everybody has Swiss quality datasets available!). these choices need to be clearly explained and potential shortcomings evaluated and discussed.

**Author's response:** Regarding forest cover information, there are three main data sources available at the scale of the entire French Alps and we tested all three. None of them is perfect, and, certainly, Switzerland and other countries benefit from more precise systematic forest inventories. Detailed comparison with aerial photographs shows that, logically, main difficulty arises when the forest density is low, which makes the limit between forest and non-forest difficult to set. However, visual analyses in different configurations convinced us that at least the DB forest we eventually retained is clearly not that bad. And a systematic analysis over the three entire considered massifs showed that i it lead the highest accuracy both in numbers and areas with regard to the other available forest data sources (response Table 4). Yet, better forest data could lead to results that are even more reliable. Also, a less stringent PRA/NoPra rule as function of NoForest/Forest (e.g. a higher PRA susceptibility with decreasing forest density) would for sure be an interesting option for further developments. These points have been precised in the revised version of the paper. See our next responses about the quality of CLPA and DEM resolution.

**Watershed delineation**

While I appreciate the simplicity of the watershed delineation approach. delineating PRAs is not new. The OBIA approach described in Bühler et al. (2018) does the same thing in a more sophisticated way. In my opinion. Section 3.1.3 and Fig. 4 explain the calculation of the flow direction. but do not actually show how the watersheds are delineated. Since the authors' method uses standard tools available in open-source GIS software. it might be more useful for the reader to get a detailed description of how these calculations are done in freely available GIS software.

**Author's response:** The watershed algorithm we use is the one from ARCGIS that follows the references in text and the workflow of Figure 4. Following this suggestion we detailed the principle of the algorithm a bit more in text and moved the flow direction figure to the supplements as is does not really belong to our results.
Also we agree that in essence the idea is similar to the OBIA approach of Bühler et al. (2018), even if we have to say that we were not able to fully understand its details from the paper (we tried hard!). We added in the revised version of the paper that both approach follow more or less the same rationale.

**Validation of PRA identification**

I see several fundamental challenges in the current validation approach that. in my opinion. provides a very biased perspective on the performance of the PRA model.

1) The authors' choice to only evaluate the performance of the model within areas of documented avalanches means that they only test whether the PRA algorithm can identify start zones in known avalanche path (true positives). It does not provide any insight about the algorithms ability to ignore terrain where avalanche do not start outside of the known avalanche paths (true negatives). While the authors explain their approach when they introduce their modified confusion matrix (L290+). this does not seem to be very meaningful to me. As explained by the authors. avalanche cadaster datasets are not widely available and have limitations in many areas. The purpose of PRA models is to identify PRA in areas where direct observations are not available. In my opinion. a more meaningful approach would be to validate the model in areas with high confidence in the avalanche mapping record and include both avalanche terrain and non-avalanche terrain so that the complete confusion matrix can be properly evaluated. As can be seen in Fig. 3 and 7. there are considerable areas outside of the avalanche path areas that the algorithm incorrectly identifies as PRAs.

2) Applying the PRA model steps (e.g.. > 1400 m. slope incline between 28 and 60°. watershed delineation. etc.) to the CLPA dataset before conducting the validation completely defeats the purpose of a validation. Obviously. the model will perform well if the validation only includes terrain with the same characteristics. In the end. the authors only evaluate the steps in the PRA algorithm that are not included in the CLPA preprocessing (slope curvature?). This is a fundamental weakness of the paper.

3) The simplified confusion matrix and calculation of the accuracy and error rates derive directly from the authors' choice of only examining true positives and false positives. As pointed out above. I do not think this is meaningful. Simply assuming that the true negative is 100% is not meaningful and leads to inflated accuracy rates. It is also unclear to me why the authors use percentages in their confusion matrix calculations. Confusion matrices are general populated with counts. which is possible for evaluating both the identification of individual PRAs and the total area of PRAs.

4) Nowhere in the manuscript is explained how the author identify a match between a PRA identified by the algorithm and the validation dataset. Is 100% overlap required or do the authors use a different rule to distinguish true positives from false positives?

5) Only focusing on the accuracy rate is a very simple evaluation of performance. Furthermore. since the error rate is simply the complement to the accuracy rate. having the error rate in all the tables does not add any value. The use of this simple validation measure is very much at odds with the content of the paragraph on model evaluation in the introduction (L87+). where the authors seem to highlight the value of more advanced evaluation approaches. This seems a missed opportunity for contributing to the literature.

6) The repeated statement that the validation in this study is done over large areas of terrain (i.e.. entire massifs) is incorrect. The validation was conducted within documented avalanche paths within these massifs. which. as highlighted in Fig. 7. are generally very small areas.

**Author's response:** We fully agree that the validation is the crucial issue Even if, with the CLPA, we have a very valuable data support, it was for us the main source of questions and concern during the work. As said in our main comment, it is also the point on which, even if clearly we do not pretend to solve the problem, we may bring some methodological/generic outcomes/thoughts for the community.  This is why we chose to focus on the validation at several points in the paper, and even in the title with the "ground truth" words. But we agree that our rationale was not clear enough and the review process helped us to formalize our thoughts as follows:
- Despite drawbacks inherent to any avalanche cadastre, the CLPA is an excellent source of information regarding past avalanches, possibly one of the finest worldwide (in terms of compromise between a very large extent and a high data quality over the area), due to its old history, its extremely regular update by devoted technicians, continuous financial support from the French ministry of the environment and because it includes various complementary sources of information (testimonies, landscape footprints, etc.). This makes it over the years closer and closer to the true maximal avalanche prone terrain. From that

perspective, it is perfectly suited to evaluate a method that aims at automatically identify the maximal avalanche prone terrain as we aim at. Notably, as CLPA extension polygons are concatenations/unions of all observed avalanche extensions on a given avalanche path, CLPA is more likely to provide an accurate estimate of the entire "ground truth" than any observation of single avalanche events. See the CLPA extracts below that will be inserted in the paper as an additional supplementary figure (response Figure 1).

- Yet, visual results and scores must be interpreted with care due to the peculiarities of CLPA data. Within a given massif, some areas are covered by CLPA and some of them are not (response Table 2, response Figures 2, 7, 8 and 9). Accuracy scores are evaluated only over areas which are covered by CLPA, namely much larger areas in Mont Blanc and Maurienne Massifs than in Chartreuse massif (response Table 2). By contrast, in areas covered by CLPA, it is known that CLPA is very good on lower slopes and in forested terrain and more likely to miss avalanche prone terrain (and release areas) at high elevations, far from inhabitants and forests (response Figure 7). In such latter cases, "false positives" are likely to be often avalanche extents that are missing in CLPA.

- CLPA does not distinguish release areas from flow paths and runout zones, which implies that a pre-processing is required to isolate individual release areas within CLPA extensions that can be compared with our PRAs. Hence, for us, the issue is not that the CLPA validation data is not "ground truth" but that indeed the predicted PRAs and validation data are not independent (they are initially, but the pre-processing of the CLPA with the slope, forest, etc. filters introduces some dependency). However, let us say boldly that we are almost sure that obtaining a fully independent sample of "ground truth" PRA is simply not possible. Indeed, even "live", one never really observes a release area, but the full extension of an avalanche, and delineating the release area always involves some subjectivity (except, maybe, with high speed camera and films that can be watched in slow motion to see the avalanche at its earliest stage…). Also, assuming one is able to observe a "true" release area, there is little chance that the entire PRA is observed. Consequently, the definition of any validation sample will always involve some partially subjective and more or less explicit choices, with possible use of some filters (slope, etc.) similar to those we use. Our choice was to do it and to say it explicitly in a transparent manner.

- Even with the best validation sample at hand, one will never be sure that i) all potential PRAs have been spotted, ii) the maximal potential extension that can be released under the most extreme conditions has been spotted for all PRAs (see our previous point about place where CLPA is good / less good). As a consequence, for us, only "true positives" can be trustfully validated, as it is never sure that a complete PRA or a part of a PRA automatically identified but not present in the validation sample is not simply missing from the validation sample. This is why we focus in our validation approach on accuracy scores/ true positives only.

- Accuracy scores (or other quantities related to confusion matrixes) were seldom used so far to evaluate PRA detection methods, to our knowledge, only in Bühler et al (2018) and with one single metric. Even if this is far from nothing, this is not much. Especially, we stress that one metric is not enough to judge the accuracy of a PRA detection method, as, e.g. the right number of PRAs can be identified but with wrong extents, and vice-versa. As a first step, we propose to evaluate accuracy scores both for PRA numbers and areas, which may cover the two most critical dimensions of the problem, but additional complementary metrics should probably be used as well in the future (focusing e.g. on the shape of PRAs, their elevation, etc.).

- Eventually let us state that focusing the PRA search and validation on terrain which are presumably favourable to avalanches is arguably not a bad idea. It is a rather standard approach in machine learning that is increasingly used in susceptibility mapping approaches outside the snow avalanche field in order to focus the detection on most suitable areas and thus increase the detection power. But we agree that this should be considered when interpreting the obtained scores.

In the reworked paper, we reinforced these points considerably in the discussion to highlight the limits and outcomes of our work on PRA validation for the community. Also, we removed "ground truth" from the title in order to avoid any misinterpretation. And we now better explain the peculiarities of the CLPA data (we tried to do it already but we understand that this may be hard to understand from different countries with different ways of collecting, presenting and using avalanche data), and how they affect the results.

We also removed unnecessary numbers in confusion matrixes and Tables, focusing only on accuracy scores on numbers and areas. These sum up all information related to true positives that we can decently evaluate.

We agree that how the matching between the detected PRA and the validation sample is done was not sufficiently clear. Detected PRAs and the validation sample are considered as polygons. In terms of numbers, a

detected PRA and a validation polygon match as soon as their intersection is non-zero. The confusion matrix in area is computed by evaluating intersected areas. This has been added to the revised manuscript.

We eventually further discuss the obtained scores and how they should be interpreted (see our response below about "sensitivity study"), with the support of the additional in-depth large scale parametric/sensitivity study that includes the effect of change on the validation sample.

**Challenges in sensitivity analysis**

As mentioned above. the sensitivity analysis does not seem sufficiently rigorous to provide meaningful insight. For example. the authors only compared the benefit of the 1400 m elevation threshold to not having an elevation threshold at all. Why not test other threshold values (1300 m. 1500 m. etc.)? The sensitivity analysis also only examines certain parameters and leaves out others without explanation. In my opinion. properly deriving the parameter selection and thresholds from the available data is a critical piece that needs to be included in this paper.

Given that the authors use a low-resolution DEM that is far below the quality recommended in the literature. it seems critical that this choice is justified with a proper sensitivity analysis. While the performance of the lower-resolution DEM will likely be lower. its use can still be justified based on a cost-benefit argument. but it will be important to have the comparison to better understand the consequences of this choice.

Some of the main results of the sensitivity analysis do not seem to match common sense. The fact that including slope incline increases the accuracy rate by less than 2% seems wrong as incline is one of the primary determining characteristics of avalanche terrain. In my understanding. this odd result is the direct consequence of only looking at terrain within documented avalanche paths and processing the validation dataset with PRA algorithm rules. which clearly highlights the limitations of the validation approach taken in this study.

**Author's response:** Let us first state that we do not really perform a sensitivity analysis, which would require a much more sophisticated approach that what we do. Such approaches have been introduced in the snow avalanche field only recently (e.g. Heredia et al., RESS 2021) and this remains probably to be done for the specific issue of PRA detection. So we prefer speak of a parametric study.

Regarding the different parameters / thresholds, we now provide systematic results all over the Mont Blanc Massif. New Table 6 is not a complete grid search, which is something numerically intensive and, to our opinion, not mandatory given our objectives and the ways our scores and results should be interpreted, see below. However, it is a systematic search moving each parameter one by one. Also, we expanded the results that were already in the paper by evaluating how accuracy scores evolve when applying the different steps/filters of the method successively, and with Theia and Corine land cover forest data instead of the IGN DB forest data. This could be done all over the Mont Blanc, Maurienne and Chartreuse Massif. We also investigated the effect of DEM resolution on i) scores and ii) the visual aspect of detected PRAs and included results in the paper. As already stated, results overall highlight i) decreasing accuracy rates as one leaves the default values determined from the literature and used for the identification of the validation sample, ii) rather stable accuracy. Yet, it can be noted that, when removed one by one, the watershed delineation step and the forest and minimal area filters appear as important, at least in terms of PRA numbers. Regarding the resolution issue, performing the PRA detection with higher resolution DEMs does not improve the results when the same validation sample is considered (i.e. determined with the 25 resolution DEM). Yet, more numerous PRAs of smaller areas are detected.

The effect of the DEM resolution was further studied by investigating how it affects the determination of the validation sample and the subsequent accuracy scores after the PRA detection highlighting again limited added value (and even a slight decrease in accuracy) with higher DEM accuracy.

Eventually we understand that our scores may appear as erroneously high. We want to remember that they concern the "true positives" only, and that we are fully aware that they are not independent of the way the validation set up is designed. As a consequence, they should not be directly compared with scores obtains with other approaches on other data sets. We simply consider them as probative enough to suggest that our approach performs rather well in the French context. Visual inspections on well-known areas confirmed that the PRA

detection was indeed able to perform rather decently. In the same spirit, our parametric search should not be seen as a way to determine a truly optimal combination. We do not think we have the data that would allow this. Simply, we are able with it to determine a set of values, or different ranges of values, that perform rather well, and it is probably a good thing that the method is not too sensitive to very slight changes over reasonable ranges of most important parameters/thresholds.

These results have been inserted and more deeply analysed in the revised version of the paper. Also the discussion section now stresses even more than in the previous version how our scores and results should be interpreted.

**Comparison with other algorithms**

In my opinion. comparing the algorithm introduced in this paper with some of the established methods would be very important for highlighting the value of the new approach. I think this should be included in this manuscript.

**Author's response:** We are sorry but this is where we simply cannot go at this stage. Doing so would imply having access to existing algorithms as open access codes/routines. which is currently not the case. Obviously, the principle of existing algorithm is published, but the description is not always very precise and easy to follow. And even when it is, following published guidelines/equations do not guarantee that what the different authors have done can be reproduced exactly, due toe, e.g.. differences introduced by different numerical implementation schemes. Therefore, the only comparison that can be done at this stage is the one regarding the choice of the different parameter values and/or steps of the approach. This was already done in the previous version of the paper but was reinforced in the reworked discussion.
However, we really think that such an inter-comparison on different data sets (as results may be to some extent case-study dependent) would be beneficial for the community and we would be very happy to contribute. This is a reason why, as a first step, we provide the full data set corresponding to the paper in open access.

**Limited discussion**

In its current form. the discussion primarily repeats information from earlier sections of the manuscript (i.e.. introduction. methods and results) without adding much value. This is partially due to the fundamental limitations mentioned above. The tone of the discussion is also quite casual (e.g.. L529: "All in all. the validation data we use is certainly not perfect and our validation approach may potentially favour the comparison with our detected PRAs. …"). which does not seem appropriate for a scientific publication. See the technical comments section for additional comments. The outlook section does not seem to offer any novel ideas as it primarily discusses already existing application cases for PRA maps (e.g.. large scale mapping of avalanche hazard and risk) and existing research extensions (e.g.. PRAs conditional on snow and weather conditions. probabilistic detection rules).

**Author's response:** The whole discussion was deeply reworked in the revised version of the paper to remove the unnecessary material, and, by contrast, to expand the discussion with respect to our answers to the different comments. See also our responses concerning the tone.

**SECONDARY ISSUES**

**Set up of the research objective and expectations**

To motivate their study. the authors provide a fairly comprehensive. even though not completely up-to-date. summary of the existing literature on PRA identification in the introduction. In this overview. they identify several limitations of the existing approaches (e.g.. disagreement about relevant terrain factors. delineation of individual PRAs. validation of PRA algorithms) to set the stage for their research objective on L103. This setup creates the expectation (explicitly or implicitly) that the algorithm introduced in this manuscript will address these issues. There are multiple issues with this. First. I do not agree with all the claims that are made in the introduction. The terrain parameters included in the various PRA algorithms do not differ from each other that much. Bühler et al.

(2018) have presented an approach for meaningfully delineating PRAs. and the most cited algorithms have been validated with mapped avalanche datsets. Second. the paper does not deliver on these expectations due to the methodological issues mentioned above. This results in disappointment and sets the paper up for failure.

I think the manuscript would benefit from a more focused introduction that describes the research objective more honestly and positions the study within the existing literature more accurately. I have no problem with a study that aims to create a simple approach for PRA identification based on easily accessible datasets. but this objective should be clearly stated at the beginning of the paper to create meaningful expectations.

**Author's response:** We reworked the end of the introduction to better state the different research objectives of our work (see our previous answers, notably our first answer concerning the overall objective of the study). We also delated the material unnecessary at this stage in the revised manuscript, but we think it is good for the reader to have a hint already at this stage of the paper of the workflow and the main outcomes with regards to these objectives.
Eventually, we fully agree that the approach and parameters we eventually retain is largely in accordance with the existing literature (except maybe for the resolution that we found less important than in other studies with regards to our chosen validation metrics and data), but we never intended to say something different. We took extra-care in the revised manuscript to clarify this.

**Language and tone**

Overall. the quality of the English in this manuscript is not very high. and the text includes many terms that are not meaningful in this context (see partial list in technical comments). While the use of these terms might the result of a poor translation from French. it is important to check the manuscript for proper use of terminology before publication.

Furthermore. the tone of the writing is rather ambitious and glowing. Examples include "In this paper. a method that well identifies …" (L16 and L103). "… the CLPA is a very valuable source of information. … and. arguably. among the rare existing ones …" (L197). and "… the CLPA is almost surely a true avalanche extent." (L188). The further exasperates the reader's sense of unfulfilled expectations mentioned above.

**Author's response:** We agree that the English of the paper was largely improvable. In addition to changes in structures and of some sentences, we did our best to proofread the paper.

Regarding the overall tone, we are really sorry if the referee has been exasperated by our limited skills in English, this was obviously really not our objective. We tried to improve our formulations as much as we could.

However, regarding the specific points mentioned by the referee, we accept that we said it badly or at least too casually, but we stick on the idea of the exceptional quality of the CLPA data (at least in terms of combination of a very large extent over which the data is of high quality). See our previous specific response about CLPA.

**Figures and tables**

The figures included in this paper are not of high quality. Several are hard to read (e.g.. Fig. 2). and the figure layout and formatting of the legends seem different in every figure. In my opinion. Fig. 5 and 6 do not add any value beyond what is already explained in the text. It is also unclear to me why the validation maps for the Mont-Blanc and Maurienne massifs are currently in the supplementary material and not included in the main manuscript.

Captions of tables are typically presented above tables. In all confusion matrix tables. the different components of the confusion matrix should be properly labelled.

**Author's response:** We produced high quality Figures for the first version of the paper already, and a loss of resolution simply occurred because the submission as a single pdf file was mandatory in the system (maybe we

missed something). We are sorry but the resulting bad resolution of some of the Figures is not our fault. However, we worked hard to improve the quality of the figures even more in terms of Layout, resolution, scope, etc.
We also agree that the Figures corresponding to the Mont Blanc and Maurienne Massif should be moved to the paper core, the overall number of Figures is not that high.
And we checked the position and content of all Table labels.

**TECHNICAL COMMENTS**

**Author's response:** We again thank the referee for his/her careful reading of our paper. All edits were be corrected and all suggestions, when not specifically answered below, were fully integrated in the revised version of the manuscript.

**Abstract**

L10: 'lacunar' is not a meaningful word in this context.

L18: 'Confrontation' should be 'Comparison'.

**Introduction**

L30: Extremely convoluted sentence.

L64: 'Retained' should probably be 'included'.

L83: 'on the field' should probably be 'in reality'.

L98: I am not sure whether the existing PRA algorithms are 'competing'.

**Author's response:** Formulation was amended.

L104: Should be '… where avalanches can occur'.

L108-120: This preview of the methods and results is not necessary in the introduction. It is best to finish the introduction with the statement of the research objective.

**Author's response:** We reworked the end of the introduction to even better state the different research objectives of our work (see our first answer). We also deleted the material unnecessary at this stage in the revised manuscript. However, we think it is good for the reader to have a hint already at this stage of the paper of the workflow and main outcomes with regards to these objectives.

**Data**

L134: 'reputed' should be 'well known'.

L176: All abbreviation need to be properly introduced the first time they are used in the manuscript. There are additional abbreviations that have not been introduced.

L176: It should be 'It consists of…'.

L179: I don't not understand what is meant with '… is mainly produced at the destination of …'.

**Author's response:** simply "the target audience is". This was corrected.

L192: '… near human stakes …' should be '… near human assets or settlements.'

L225: 'In order to …' can be simplified to 'To …'. There are several cases of this in the manuscript.

L241: The sentence that describes how the thresholds and parameters are chosen (reference to Sect 4.2) does not seem to belong here.

**Author's response:** We now highlight more specifically how the default values for the different thresholds and parameters have been fixed.

L245+: The description of the algorithm provided in this section does not seem consistent with the information presented in Fig. 5.

**Author's response:** We slightly reworked the description to improve its clarity.

L261: The statement that PRA identification target primarily large avalanches needs to be stated much earlier in the manuscript as it is a fundamental assumption of the study.

**Author's response:** It has been moved to introduction.

L271+: The description of the CLPA seems repetitive as it discusses information that was mentioned previously already.

**Author's response:** We deleted the unnecessary material.

L280: Typo: PRAS should be PRAs. There are several instances of this typo in the manuscript.

**Results**

L327+: The description of the confusion matrix provides exactly the same information that is shown in the table. Hence. it does not add any additional value.

**Author's response:** We deleted the unnecessary material.

L349: I do not know what you mean with 'probative'. There are several uses of this work in the manuscript.

**Author's response:** see our responses about validation and scores.

L350+: The explanation provided here seems rather speculative and not well grounded.

**Author's response:** We now better ground our explanations on the characteristics of the massifs and of the CLPA in them (new Table 1).

L371: Not sure what you mean with 'parametric study'.

**Author's response:** see our responses about sensitivity analysis.

L380: In academic writing. the term 'significant' should only be used in the context of statistical significance. Use 'considerable' or substantial' instead.

L382: 'Use 'more substantially' instead of 'more largely'.

L406: The last sentence in this paragraph is too hand-wavy and not grounded in evidence.

**Author's response:** We now better ground our conclusions on the results of the parametric studies.

L410: Table 6 does NOT show that not including the forest layer results in the worst performance. The accuracy rates without forest are higher than with the theia dataset.

**Author's response:** see new Table 5 for the Mont Blanc Massif and related text.

L421: Delete 'eventually'.

L425: Use 'tower' instead of 'pylon'.

L435: Delete 'eventually'.

**Discussion**

L447+: There is no need to repeat the information from the intro at the beginning of the discussion section.

**Author's response:** The whole discussion was deeply reworked in the revised version of the paper. Regarding this specific point, we deleted unnecessary repetitions, but we think it is important before starting the discussion to remember briefly the objectives of the work and what has been achieved.

L455+: I do not think these objectives have been achieved by this study.

**Author's response:** see our responses about paper objectives, validation and scores.

L469: I am not sure what is meant with 'probative'.

**Author's response:** see our responses about validation and scores.

L475+: This discussion primarily repeats information from the methods and results section without adding much value.

**Author's response:** The whole discussion was deeply reworked in the revised version of the paper to remove the unnecessary material, and, by contrast expand the discussion with respect to our answers to the different comments. See also our responses concerning the tone.

L499: Reword this sentence. It should be '… by comparing X to the processes CLPA dataset.'.

L503+: This sentence seems like an excuse and is not very convincing. The readers' expectations should be managed by properly describing the research objective in the introduction.

**Author's response:** see our responses about paper objectives, validation, scores and comparison of existing algorithms.

L507: 'Envisaged' should be 'envisioned'.

L510: 'Confronted' is the wrong word here.

L529: Similar to the sentence on L503. this sounds like an excuse. Limitations should be discussed more seriously.

**Author's response:** see our responses about paper objectives, validation, scores and comparison of existing algorithms.

---

## Author Response (AR2)

Nicolas Eckert
IGE, Grenoble Alpes University, INRAE, France

Grenoble, February 18th 2023

Submission of a revised version of our article to *NHESS* entitled "Development and evaluation of a method to identify potential release areas of snow avalanches based on watershed delineation

Dear Yves Bühler, NHESS scientific editor.

We deeply thank the referee for his/her feedback on our article and their insightful comments. We also thank you for your editorial revue and your suggestions. All points have been addressed in the revised version attached to this submission. In what follows, we further provide a point-by-point answer to all comments and questions and detail the changes made in the revised manuscript. We hope this revised version will be found suitable for publication in NHESS. Thank you for your consideration of our work.

Sincerely,

Nicolas Eckert, on behalf of the authors

Dear authors:

I appreciate the opportunity to review the revised version of your manuscript. It is obvious that you have put a lot of effort into the revisions, and the quality of the manuscript has improved substantially. I really appreciate the addition of the more in-depth discussion of the strengths and weaknesses of the various datasets and the parametric analyses. Despite these substantial improvements, I feel that there are several remaining issues that should be addressed before the manuscript can be published.

**Authors' Response (A.R.):** We deeply thank the referee for his/her positive judgement about our work and his/her meaningful additional suggestions and feedback. We hope our revised version will be found suitable for publication in NHESS.

Major comments

Description of steps of PRA decision method and CLPA procession

I think that the description of the steps of the PRA detection method could be improved by better aligning the description in the text with the graphic presented in Figure 3. Right now, I find the description rather confusing because it talks about three main steps that do not seem obvious in Fig. 3, and while I understand the reason for the split over the two columns, they do not obviously line up with the description in the text. Furthermore, the presentation of the CLPA processing steps in Fig. 4 is visually very different even though some of the steps are the same as in the PRA detection method. Given these similarities and the fact that the PRA detection and CLPA procession steps are closely tied (as explained in the text several times), I think that a more consistent graphic presentation that highlight these connections more obviously (either in a single or two figures) would allow the reader to understand the approach of the analysis more easily.

A.R.: Regarding the description of the method, we tried to rework it once more to make it more explicit. Notably, we removed the reference to "three main steps", which was indeed confusing, as these do not appear on Figure 3.

Regarding Figure 4, it includes both i) a very brief presentation of the data included within the CLPA, ii) how this data is processed to generate a validation sample for our PRA detection method. We reworked the caption of the figure to better underline this (previous caption that mentioned only the processing of the CLPA data was indeed confusing).

Confusion matrix

I am still concerned about the fact that you use the accuracy rate in your study. In reality, you are only looking at the precision/positive predictive value (= true positives/(true positives + false positive)), and your assumption that the true negative rate is 100% artificially produces an accuracy rate value that is halfway between the precision value and 100% without adding any value to the analysis. Similarly, this assumption also creates error rate values that are halfway between the false discovery rate (= false positives/(true positives + false positive)) and 0%. Given your new description of the strengths and weaknesses of the CLPA dataset (which is much appreciated), the assumption of a 100% negative predictive value and 0% false omission rate seems somewhat bold.

In my opinion, it would be more accurate and more transparent to base your evaluation on precision/positive predictive value instead of the accuracy rate. This will not affect the results of your analysis at all, but it will describe the focus of your evaluation more honestly and prevents possible confusion with accuracy rates presented in other studies that actually work with the full confusion matrix. Note that you explicitly point out this limitation yourself on L633. I think it would be very useful for you to highlight in the conclusion section that future studies should aim to assess PRA algorithms with the full confusion matrix.

A.R.: We agree with this comment and have reworked the full paper (including tables and supplements) to provide all results in terms of true positive rates (also known as recall) instead of accuracy rates. Only exception is the description of the confusion matrix for the area of Chamonix,

which is used to introduce the different terms of the matrix and the different scores in a pedagogic way.

By contrast, we did not mention that further research should focus on the full confusion matrix as we think that this is not feasible. As explained in text, we indeed believe that, even with the "best" data set of observed avalanche release areas at hand, one will never be sure that a false positive is simply not a release area or a fraction of a release area that has never been observed so far but could be triggered one day under very specific conditions, see our discussion section for further details.

Comparisons in parametric studies

I appreciate that you now explore the robustness of your approach with a parametric studies. However, I am a bit confused about the fact that parameter values and ranges were only changed in the PRA algorithm and not the validation dataset even though most of them are used the same way in both. It seems obvious that the PRA algorithm that uses the same parameter values as the CLPA processing will naturally perform the best! Applying a different slope or elevation filter in the PRA algorithm but keeping the default one for the CLPA processing obvious decreases the performance. I understand that this relates to the challenging task of defining the "ground truth" (which requires some assumptions), but it seems to me that potential insight from the current approach is limited.

Would it make more sense to also change the parameter values in the CLPA procession like you did for the DEM resolution analysis (L505). I have not completely thought this through, but it would keep the assumptions consistent and allow you to compare apples with apples and not apples with oranges.

**A.R.:** This is a tough question that we had in mind during the whole work, and we either do not have a definite answer to it. We chose not to recompute the validation sample at each time as, indeed, ground truth should be fixed, but we agree that this favours our "default setting" in the parametric study. For the DEM resolution, we performed both computations as it was a particularly critical point of the analysis for which our findings slightly differ from the state of the art. We could have done this also for all other analyses, but this would have largely increased the number of tables and scores to be analysed (which are already quite numerous). In addition, we do not think that this would have had a large benefit. Indeed, as discussed, our parametric search should not be seen as a way to determine a truly optimal combination of parameters. We do not think we have the data that would allow this. More modestly, our parametric study, as it is conducted, shows that our PRA detection method is, to a certain extent, rather robust over a certain range of parameters which is consistent with the state of the art. Also the DEM analysis example shows that, even when the validation sample is recomputed, the default setting may still be favoured (Table 7). These elements were somehow already present in the discussion of the previous version of the paper but we reworked it slightly to try to be even clearer.

Minor comments

L153: If the optimum DEM resolution is examined in the study, shouldn't all DEM datasets be described in the data section and not just the 25 m one?

**A.R.:** Indeed, we added in text the precision that the DEMs of finer resolution were also provided by IGN.

L192: I think it would be useful to explicitly explain why you trust the CLPA dataset so much instead of just stating it as a fact.

**A.R.:** As stated in our discussion, the trust comes from the CLPA long history, with regular updates by skilled and devoted technicians, continuous support by the French ministry of the environment and the inclusion of a large amount of different data sources, so as to be as close as possible to reality. The paragraph has been reformulated as: "Due to its long history, its regular update by devoted technicians, the continuous financial support of the French ministry of the environment and the consideration in the determination of avalanche terrain of a large amount of different data sources, CLPA is very reliable, meaning that an avalanche extent which is within the CLPA is almost surely a true avalanche extent".

L237: It is still a bit unclear how the watersheds are actually delineated. Figure S2 shows how the flow direction and accumulation are calculated but does not show how the actual boundaries are drawn. A slightly bigger example with the actual boundaries drawn would be more informative.

A.R.: To delineate watersheds, we used a standard algorithm well described in the literature. However, the procedure it is not very easy to explain in a few words and to represent within a single figure. As this is not the heart of our paper, we prefer providing the idea only and referring to the source papers. We have added the reference to Djokic and Ye (2000) for a seminal description of the watershed delineation procedure (which includes several illustrations).

L312: The fact that only one pixel of a validation PRA must be identified for a successful match seems a very low bar and a critical assumption of the analysis. It might be worthwhile to justify this choice in more detail and/or explore the effect of different thresholds.

A.R.: We agree that one pixel for a successful match can actually be seen as a "lower bound" (related accuracy is measured by our recall defined on PRA numbers). This is exactly why we also provide a kind of "upper bound" with the recall measured on PRA areas. Our discussion already mentioned that one single metric is certainly not enough to truly assess the efficiency of a detection method, so that we proposed two metrics that cover the most critical dimensions of the problem (PRA numbers and areas). We added to the discussion that, in the future, additional metrics should be considered, notably metrics that combine both information (e.g. successful match for different thresholds defined as minimal matching areas), and/or metrics related to various other characteristics of the detected PRAs (shape, elevation, etc.). This may help assessing even more precisely the strengths and weaknesses of our (or another) PRA detection method.

L540: I appreciate the honest discussion of the limitations of the performance measure here, but I think this could be addressed/avoided by using a more appropriate performance measure that takes the limitations of the dataset into account more honestly earlier (see earlier comment).

A.R.: See our response to the main comment about the choice of the performance measure. The whole text has been reworked accordingly.

L585: It would be better to include the suggestion for a full comparison of different PRA algorithms in the conclusion section where you make other suggestions about future research.

A.R.: This has been done.

L641: I did not read the paper by Giffard-Roisin et al. (2020) in detail, but I think it would be important to briefly mention that while there are benefits to increasing detection power, increasing false positives also has its cost/challenges.

A.R.: We added a note saying that doing so may indeed increase the number of false positives.

L651: It is not completely correct that you validated your PRA algorithm over entire massifs, because your performance measures are only based on the areas where CLPA data is available, which are fractions of the entire massifs.

A.R.: We reformulated as "covering significant proportions of three entire massifs with diverse characteristics"

L656: Are these suggestions meaningful/realistic given the inherent limitations of the CLPA dataset?

A.R.: Probably not all for the CLPA data, the reason why we wrote "and/or with different validation data". Indeed different validation data with different strengths and weaknesses (we doubt that any "perfect' data set may exist) may allow investigating these different issues. We precised as "A similar approach could be further used for comparing different PRA detection methods and/or in other contexts with different validation data having strengths and weaknesses different from those of the CLPA."

The manuscript will require detailed editing as the English is still of limited quality. Below are some comments for improving the writing, but there are likely more issues. I assume that the NHESS editorial team will take care of this before the manuscript is published.

**A.R.:** We agree that the English of the paper was still improvable. In addition to suggested changes, we did our best to proofread the paper once more.

Abstract
L17-20: I think the performance measures and values used in this study need to be described more accurately in the abstract. See earlier comment on the performance measures.

**A.R.:** We have reformulated the sentence as: "Comparison to an extensive cadastre of past avalanche limits from different massifs of the French Alps used as ground truth leads to true positive rates (recall) between 80-87% in PRA numbers and 92.4% and 94% in PRA areas,…". See also our response to referee one about the choice of the performance measure.

Introduction
L 62: "Eventually" is not the right term here. You could say "finally" instead. There are many incorrect uses of "eventually" throughout the manuscript. Please replace throughout.

**A.R.:** This has been done.

L63: The last sentence in the paragraph (As a consequence, …), does not seem properly connected to the rest of the paragraph. Please expand and explain in more detail.

**A.R.:** We have reformulated the sentence as "Finally, PRA detection methods are primarily oriented towards large avalanches, which are of interest to assess long-term risk for people and settlements downslope, so that a minimal size is generally considered (e.g. Maggioni et al. 2002).

L70: Missing "and" before ii).

**A.R.:** This has been done.

L 75: Replace "is very dependent" with "depends".

**A.R.:** This has been done.

L 93: "confront" should be "compare".

**A.R.:** This has been done.

L101: Replace "summed-up as" with "summarized in". "Summed-up" is used in several locations of the manuscript and should be replaced everywhere.

**A.R.:** This has been done.

L103: Replace "remain little used so far" with "have only seem limited use so far."

**A.R.:** This has been done.

L115: Replace "ground" with "build".

**A.R.:** This has been done.

Data
L150 – Table 1: First, this table seems to include results already. This is rather unusual for a table in the methods/data section. Second, the areas are not explicitly introduced in the text. Their purpose is mentioned on L 139 in general, but the actual areas are not described.

**A.R.:** For us a Table (or a figure) does not belong to a specific section, it is just located at the place where it is called first in text. Actually, this table is called at several places in text, including in the data,

methods and results sections, so it is logical that it includes information relevant for the case study presentation and for the application of our method. Another solution would have been to split the table in several tables possibly located closer to their use in text, but this would have enlarged the number of tables, which is already high, so that we think it that our solution is sensible.

Regarding the small study areas, we have added the following sentences to introduce them in text (with reference to Figure S1 in the SM where they are mapped): "The Chamonix area is a 34.3 km2 area, which is part of the Mont Blanc massif and includes the municipality of Chamonix Mont Blanc. The Chartreuse / Dent de Crolles area is an even smaller area (7.6 km2) located within the Chartreuse massif and with the Dent de Crolles (2,062 m a.s.l.) in its center (Figure S1 in the SM).

L197: "Avalanche extensions", which is used extensively throughout the manuscript is not the right term. In this particular case, "avalanche records" would work, but most often it refers to the "accuracy of the recorded extent of observed avalanches." Please correct this throughout the manuscript.

**A.R.:** CLPA really represents avalanche maximal extents and not avalanche records. Notably, it does not include, e.g., the dates and the characteristics of single avalanche events, and even not the contours/extents of individual avalanche events. This is written in the paper, but we understand that it may be confusing for people from countries where habits regarding avalanche data are different. Also, the term "avalanche extension" is the one officially used in the official CLPA caption, and it is very important for us to be precise and consistent from this perspective. Yet, we understand that our formulation was not fully correct from the point of view of the English language. We reworked the paper in order to reach an acceptable compromise, namely we now use "avalanche extent" throughout the text but we kept "avalanche extension" in the captions within the figures when it is necessary (Figs. 4-8), with a note of explanation in the expanded Figure description (the expanded caption below the same figures).

PRA detection

L 236: Delete "(where flow accumulation is equal to zero)" as it is repetitive.
**A.R.:** This has been done.

L 249: Replace "few" with "too little".
**A.R.:** This has been done.

L 260: "e.g.," should probably be "i.e.,"
**A.R.:** This has been done.

Results

L 340 – Caption of Fig. 5: Replace "concordance" with "agreement".
**A.R.:** This has been done.

Discussion

L646: It seems inaccurate to mention the confusion matrix here since you did not use the full confusion matrix. Instead, you should more strongly highlight that you examined the performance with respect to area and number of" PRAs, which is more novel.
**A.R.:** The sentence "Finally, confusion matrices and performance criteria were seldom used so far to evaluate PRA detection methods" does not refer to our work and is for us fine. Instead, we specified more clearly what we did in the next sentence as "As a first step towards improved evaluation schemes for PRA detection methods, we proposed to evaluate efficiency with true positive rates (recall) computed both for PRA numbers and areas, which may cover the two most critical dimensions of the problem".

Conclusion

L664: It is unclear to me what you mean with "and close contexts (see below)."

**A.R.:** We reformulated as "mountain environments with similar characteristics".

L668: You should explicitly explain how your results contribute to the field and not leave this up to the reader. They might not see it themselves.
**A.R.:** Our sentence was devoted to refer to the findings that where listed just before. We reformulated as "outcomes of the work include i) the determination of individual PRAs using a watershed delineation algorithm, ii) an approach to define a validation sample from a cadaster of avalanche extents, iii) an evaluation procedure based on two metrics, PRA numbers and area, and iv) a better definition of accuracy scores that should be interpreted in the context of PRA identification. These methodological developments should help progressing towards more efficient approaches for PRA detection and evaluation". We hope it is clear like this.

L680: Replace "confronted" with "compared" or "contrasted".
**A.R.:** We replaced by "compared".